# Towards non-invertible anomalies from generalized Ising models

Shang Liu[1] and Wenjie Ji[2]

**1** Kavli Institute for Theoretical Physics, University of California,
Santa Barbara, California 93106, USA
**2** Department of Physics, University of California, Santa Barbara, California 93106, USA

## Abstract

We present a general approach to the bulk-boundary correspondence of noninvertible topological phases, including both topological and fracton orders. This is achieved by a novel bulk construction protocol where solvable $(d + 1)$-dimensional bulk models with noninvertible topology are constructed from the so-called generalized Ising (GI) models in $d$ dimensions. The GI models can then terminate on the boundaries of the bulk models. The construction generates abundant examples, including not only prototype ones such as $\mathbb{Z}_2$ toric code models in any dimensions no less than two, and the X-cube fracton model, but also more diverse ones such as the $\mathbb{Z}_2 \times \mathbb{Z}_2$ topological order, the 4d $\mathbb{Z}_2$ topological order with pure-loop excitations, etc. The boundary of the solvable model is potentially anomalous and corresponds to precisely only sectors of the GI model that host certain total symmetry charges and/or satisfy certain boundary conditions. We derive a concrete condition for such bulk-boundary correspondence. The condition is violated only when the bulk model is either trivial or fracton ordered. A generalized notion of Kramers-Wannier duality plays an important role in the construction. Also, utilizing the duality, we find an example where a single anomalous theory can be realized on the boundaries of two distinct bulk fracton models, a phenomenon not expected in the case of topological orders. More generally, topological orders may also be generated starting with lattice models beyond the GI models, such as those with symmetry protected topological orders, through a variant bulk construction, which we provide in an appendix.



# 1  Introduction

Bulk-boundary correspondence has been a key concept for understanding topological phases of matter since the discovery of quantum Hall effect. Transport responses in quantum Hall bars are fundamentally contributed by the chiral gapless edge modes on the boundaries. [1–6] More generally, the nontrivial boundary properties of various topological phases in $d + 1$ spacial dimensions can be understood as a consequence of anomalies of the boundary theory in $d$ dimensions [7–14] – obstructions in realizing a theory in a *local lattice model* in the $d$ spatial dimensions with a tensor product of local Hilbert spaces, a local Hamiltonian, and an onsite symmetry action if any. The edge theory of an integer quantum Hall insulator has an invertible gravitational anomaly characterized by the imbalanced left and right moving modes. The boundary low energy effective theories for onsite-symmetry protected topological phases have 't Hooft anomalies, which prevent an onsite symmetry realization on a lattice without a topological bulk, as well as the theory to be coupled to gauge fields. Each of the above anomalies is invertible, as the anomaly can be matched by an invertible phase in one dimension

higher, the outcome model in one higher dimension with a boundary is a local lattice model. This connection between the bulk topological phases of matter and the boundary anomaly has significantly deepened our understanding of both sides.

In recent years, the bulk-boundary correspondence for noninvertible topological phases, and the notion of noninvertible anomaly have attracted much interest [6, 15–44]. Particularly, the boundary of two dimensional non-invertible topological phases, even when gappable, does not admit a local lattice model. As the simplest example, the one-dimensional transverse-field Ising model, *restricted to the $\mathbb{Z}_2$ symmetric sector*, is not realizable as a one-dimensional model with a tensor product Hilbert space and a local Hamiltonian. Nevertheless, it can be realized as a boundary theory of the two-dimensional $\mathbb{Z}_2$ toric code model [36, 37, 45], and is termed to have a noninvertible anomaly. A modular covariance condition satisfied by the Ising model with restricted Hilbert space follows from this bulk-boundary correspondence: Threading different anyon fluxes in the bulk changes the total symmetry charge and the boundary condition of the boundary Ising model. This leads to a vector of parition functions for the boundary model. Then, under modular transformations (certain large diffeomorphisms on the underlying spacetime manifold), this partition function vector transforms covariantly, according to the topological $S$ and $T$ matrices of the bulk topological order, which capture the statistics of anyons. Such correspondence between the $d$-dimensional model subject to global constraints and the $(d+1)$-dimensional topological order has been termed as the matching of *non-invertible anomaly*. The vector of partition functions in this example can also be implied from the categorical symmetry[1] of the Ising model [47–49]. The modular covariance condition also holds in various one-dimensional critical systems on the boundary of two dimensional systems with topological excitations and topological defects [37, 50]. Related findings with more mathematical oriented discussions are in [51–53]. More examples of generalized symmetries, whose generators under multiplication form a fusion category, have been uncovered in models with either restricted [47, 54–56] or non-restricted Hilbert spaces [57] in recent years.

Along one direction to generalize the above example, in this paper, we consider a wide class of qubit lattice models in arbitrary spatial dimensions, which can have sets of $\mathbb{Z}_2$ symmetries that may be global or within subsystems and are dubbed generalized Ising (GI) models. We provide a generic construction, which, when applied to each GI model, produces at least one exactly solvable lattice model in one dimension higher, dubbed a bulk model. The ground state subspace of each bulk model is stable against local perturbations.[2] The construction generates abundant topological or fracton ordered models: not only prototype ones such as the $\mathbb{Z}_2$ toric code models in two spatial dimensions or higher, and the X-cube fracton model [59]; but also more diverse types such as $\mathbb{Z}_2 \times \mathbb{Z}_2$ topological order, four-dimensional $\mathbb{Z}_2$ topological order with pure-loop topological excitation, etc.

A main result that follows from the construction, is a concrete demonstration that the lattice model with (discrete) global symmetries terminates on the boundary of the bulk model with topological or fracton order in generic dimensions. The boundary-bulk correspondence is explicit in UV. That is, there exists an isomorphism between the GI models subject to global constraints which are either symmetry charge projections or boundary conditions, and the boundary of the topological order and/or fracton order. The isomorphism is between the Hilbert spaces, as well as between the local operator algebras generated by Hamiltonian local terms. The latter means that any Hamiltonian local terms allowed on the boundary of the topological and/or fracton order must be a product of local terms in the GI model Hamiltonian. In this sense, the most general Hamiltonian allowed on the boundary of the topological order is the GI model.

---

[1]Field theories of categorical symmetries are in development. [46] In essence, $n+1$-dimensional topological field theory can act on $n$-dimensional quantum field theories.

[2]That is, the model is locally topologically ordered [58].

Such bulk boundary correspondence can be regarded as examples of non-invertible anomalies.[3] It happens in the constructed bulk models under a specific condition (Claim 1): colloquially speaking, either there is a non-local symmetry that can be dualized to a generalized boundary condition, or a generalized boundary condition that can be dualized to a non-local symmetry. When the condition is violated, the constructed bulk model is either trivial or has a fracton order. The condition, which highlights the equal roles of non-local symmetry charges and boundary conditions and the necessity of duality shows up explicitly through the generic bulk construction.

Up to date, commuting projector Hamiltonians realizing topological ordered phases in three or more dimensions are far from exhaustive. There are a few constructions that generalize naturally in any spacial dimensions $d > 2$. Examples include the higher dimensional (generalized) toric codes [60,61], Dijkgraaf-Witten models [62], Walker-Wang models (with a non-modular category as an input) [63], and generalized double semion models [64].

Our construction adds to this list, and yet, in some sense, is simpler. The construction generates a stabilizer Hamiltonian in $d+1$ spatial dimensions, from a $d$-dimensional model on a qubit lattice with a set of $\mathbb{Z}_2$ symmetries. The construction does not start with the categorical data of the underlying TQFTs, but is based on observations on the commutation relations of Hamiltonian local terms in the $d$-dimensional model. Ground state degeneracy (GSD) that signals a topological and/or fracton order can also be computed with the stabilizer formalism, say using the standard polynomial representation [58,65].

The simplicity of such stabilizer codes in generic dimensions is inviting for an explicit analysis of the bulk-boundary correspondence, which is summarized above. This result of boundary-bulk correspondence is in complementary to many existing boundary analysis of commuting projector models of TQFTs: The boundary of the ground state of a discrete gauge theory has a global symmetry and is constrained to the charge-neutral sector [47, 49]; for discrete gauge theories in $2+1$ spacetime dimensions for a few Abelian groups, the local operators on the boundary have been matched with topological operators in the bulk, and share the same set of $F$-symbols and $R$-symbols [66,67]; the boundary of a Levin-Wen model has generalized symmetries generated by topological operators restricted to the boundary, which are found either through a lattice analysis [68], or at an abstract level [46,69,70]; the gapped surface of (confined) Walker-Wang models can be topologically ordered and protected by symmetries that are anomalous [14,71].

As far as long range orders in the bulk is concerned, one distinction of our construction is that it generates fracton ordered models as well. This is particularly interesting given that the bulk-boundary correspondence for fracton orders is yet barely explored [33,43].[4] One intriguing result we obtain is that, with some appropriate boundary condition, a single anomalous theory can live on the boundaries of two distinct bulk fracton models, a phenomenon not expected in the case of conventional topological orders.

As a heads-up, let us give an outline of the construction. We define a large class of GI models whose Hamiltonian local terms (HLTs) are either products of Pauli-$Z$ operators or products of Pauli-$X$ operators. The HLTs and symmetries of the GI model satisfy a couple of conditions. Being so, a dual model can always be obtained through a generalized Kramers-Wannier duality. A bulk model – a model of one dimension higher, can be constructed on alternating layers of the GI model lattice and the dual lattice. The HLTs of the bulk model are within the stabilizer

---

[3]This is in a weaker sense, referring to that a model, due to global constraints, is not a local lattice model on its own (thus has a *global gravitational anomaly*), yet is isomorphic to the boundary of a long range entangled phase. More completely, the model subject to distinct global constraints should be captured by a vector of partition functions. And each distinct sector of the Hilbert space of a lattice model can be the boundary of a $(d + 1)$-dimensional model with topological orders, where the topological charge in the bulk determines the boundary sector.

[4]See also Ref. [44] that appeared soon after the first version of our arXiv preprint.

formalism. Each term is a product of local terms of the GI model and its dual. By virtue of the properties of the GI models, we show the bulk model has several nice properties:

- Any ground state is robust against local perturbations.

- When the GI model has non-local $X$-type symmetries, or when the dual model has non-local $Z$-type symmetries, the bulk model is either topologically ordered or fracton-ordered.

- When the bulk model has a pure topological order, its boundary has non-invertible anomaly. The symmetric sector of the GI model that satisfies certain (generalized) boundary conditions is a boundary termination for the bulk model.

- When the bulk model has fracton orders, it can have an anomaly-free boundary, such that when discrete global symmetries appear on the boundary, the boundary Hilbert space includes all charged sectors, rather than only the symmetric sector.

The rest of this paper is organized as follows. In Section 2, we define GI models and give a few examples. In Section 3, we introduce the generalized Kramers-Wannier duality which plays an important role in our construction of bulk models. In Section 4, we construct the bulk model, and together describe a few prototype examples. Then we prove that it has a stable spectral gap and a robust GSD on a topologically nontrivial space manifold. Hence it has a topological or fracton order. The bulk-boundary correspondence is analyzed in Section 5. In Section 6, we study a collection of examples of topological and fracton orders generated from the construction. Particularly, an interesting example demonstrates that two distinct fracton models can host the same anomalous boundary theory. In the end, we summarize and discuss future questions. In Appendix G, we also give a variant bulk construction that generates topological and/or fracton orders from qubit lattice models beyond the GI model and is applicable to some models with symmetry protected topological orders. Further technical details are also summarized in appendixes.

## 2 Generalized Ising models

A GI model is referred to a model on a lattice of qubits in arbitrary spatial dimensions, whose Hamiltonian and $\mathbb{Z}_2$ symmetries has the following properties. The Hamiltonian consists of two types of terms: GI terms and generalized transverse field terms. A generalized Ising (transverse field) term is a product of Pauli-$Z$ (Pauli-$X$) operators acting on a local subset of qubits, and is denoted by $\mathcal{O}_\alpha^Z$ ($\mathcal{O}_i^X$) with some index $\alpha$ ($i$) referring the subset. Generically, $\alpha$ and $i$ are from different index sets. Written explicitly, the Hamiltonian is then

$$H = -\sum_\alpha J_\alpha \mathcal{O}_\alpha^Z - \sum_i h_i \mathcal{O}_i^X \,, \tag{1}$$

where $J_\alpha$ and $h_i$ are real coefficients. We suppose the model lives on a $d$-dimensional parallelogram with either periodic or open boundary condition along any direction. We will impose some additional assumptions on the model later in this section.

The model may have many $\mathbb{Z}_2$ symmetries. For our purpose, we only consider one group of $\mathbb{Z}_2$ symmetries: all $Z$-type symmetries generated by products of Pauli-$Z$ operators (minus sign factor excluded), and all those $X$-type symmetries generated by Pauli-$X$ operators (minus sign factor excluded) that *commute with all $Z$-type symmetries*. We refer this selected group of symmetries the *compatible* symmetries in the GI model. Compatibility is to emphasize that the generator of each $X$-type $\mathbb{Z}_2$ symmetry commutes with not only the Hamiltonian but also all

the $Z$-type symmetry generators. In fact, this implies that each $X$-type symmetry generator is a product of several $\mathcal{O}_i^X$ operators, analogous to the standard transverse-field Ising model. For a proof, see Corollary 5 in Appendix B. From now on, $X$-type $\mathbb{Z}_2$ symmetries in the GI model only refer to those compatible ones.

The many $\mathbb{Z}_2$ symmetries may either be local or nonlocal, and it is useful to distinguish them for our purpose. Let $\{G_r^Z\}$ with some index $r$ be a complete but not necessarily independent set of generators of the *local $Z$-type symmetries*. We say there are $n_Z$ number of *nonlocal $Z$-type symmetries* if one can find a maximal set of symmetry generators $\{U_1^Z, U_2^Z, \cdots, U_{n_Z}^Z\}$ satisfying that each $U_k^Z$ is not a product of the remaining ones and $G_r^Z$. Formally, this $\mathbb{Z}_2^{n_Z}$ group is nothing but the quotient of the full $Z$-type symmetry group over its local symmetry subgroup. In practice, the set $\{U_k^Z\}$ can be obtained by repeatedly adding new $U_k^Z$ that is independent from $G_r^Z$ and the existing $U_1^Z, \cdots, U_{k-1}^Z$ until the list is maximal. Similarly, for the $X$-type symmetries, we have the local generators $\{G_s^X\}$ with some index $s$ which belongs to a generically different index set from that for $r$, and the independent nonlocal generators $\{U_1^X, U_2^X, \cdots, U_{n_X}^X\}$ with some $n_X$.

## 2.1 Assumptions

Before stating our assumptions, let us introduce some useful terminologies. Consider a set of *commuting local* operators $\{M_i\}$ where each $M_i$ is a tensor product of $I, X, Y, Z$. We say that a local operator $\mathcal{A}$ is **locally generated** by $\{M_i\}$ if $\mathcal{A}$ is generated by a few $M_i$'s in a neighborhood of $\mathcal{A}$'s support, such that the linear size of this neighborhood exceeds that of $\mathcal{A}$ by an $O(1)$ constant. We say that $\{M_i\}$ is a **complete set of local observables** (CSLO) if any local operator $\mathcal{A}$ that is a tensor product of $I, X, Y, Z$ and commutes with all $M_i$ can be locally generated by $\{M_i\}$. In fact, one can show that if $\{M_i\}$ forms a CSLO, then any local operator $\mathcal{A}$, not necessarily a product of Paulis, that commutes with all $M_i$ can be locally generated.

We assume the GI models to have the following properties.

- $\{\mathcal{O}_i^X\} \cup \{G_r^Z\}$ is a CSLO.

- Any local $X$-type symmetry generator is *locally generated* by $\{G_s^X\}$.

The first assumption physically means that when $J_\alpha = 0$, $h_i \neq 0$, after restricting to the gauge invariant sector $G^Z = G^X = 1$, the system has a spectral gap stable to local perturbations, together with either a unique ground state or a robust GSD [72]. This is analogous to the Ising disordered phase.

Independent of these assumptions, the bulk model to be constructed has a Hamiltonian whose local terms all commute. These two assumptions ensure that the ground states of the bulk model to be constructed are robust against local perturbations. The above assumptions may seem technical, but in many cases, are not hard to verify, as we will see. Also note that the choices of $G^Z$ and $G^X$ operators are not unique. It suffices to make one choice that satisfies the assumptions.

## 2.2 Examples

Let us now introduce some examples. Periodic boundary condition will be taken for convenience. A particularly simple situation is when $\mathcal{O}_i^X$ are just the traditional transverse field terms $X_i$ with $i$ labeling the qubits on the lattice. In this case, there is no $Z$-type symmetry at all, and our first assumption is trivially satisfied. The simplest example of this class is of course the standard one-dimensional Ising model: $H = -J \sum_i Z_i Z_{i+1} - h \sum_i X_i$, which has an $X$-type $\mathbb{Z}_2$ symmetry generated by $\prod_i X_i$. The plaquette Ising models (see Refs. [59, 73–75] and the

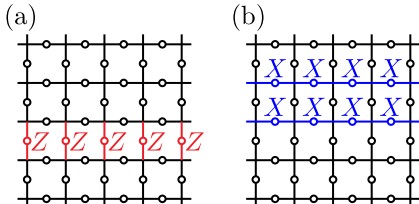

Figure 1: Nonlocal symmetry generators of the model in Eq. 3. $U_1^Z$ is illustrated in (a). $U_2^Z$ is similar but extended along the vertical direction. (b) is an example of an $X$-type symmetry generator.

references therein), and the quantum Newman-Moore model [76–80] are other examples of this class.

The two-dimensional plaquette Ising model has the Hamiltonian

$$H = -J \begin{array}{c} Z\!\!-\!\!Z \\ Z\!\!-\!\!Z \end{array} \quad - h \,\, \overset{X}{\bigcirc} \,\, , \tag{2}$$

defined for qubits on the vertices of a 2D square lattice. Here and throughout, we sometimes suppress the summation for simplicity when there is little confusion. The product of Pauli $X$ operators along each row and column generates a $\mathbb{Z}_2$ symmetry of the model, known as a subsystem symmetry. Point excitations of this model in the ordered phase ($J > h$) has restricted mobility. The quantum Newman-Moore model has subsystem $\mathbb{Z}_2$ symmetries acting on Sierpinski triangles, but is otherwise similar to the two-dimensional plaquette Ising model, so let us not write it down explicitly.

Our next example contains nontrivial $\mathcal{O}^X$ terms in its Hamiltonian. Consider the following model whose qubits live on the links of a 2D square lattice,

$$H = -J \,\, \overset{Z \quad Z}{-\!\!\bigcirc\!\!-\!\!\bigcirc\!\!-} \,\, - h \,\, X\overset{X}{\underset{X}{\bigcirc\!\!\bigcirc}}X \,\, . \tag{3}$$

Here, the nearest neighboring two-body $Z$-type terms along the vertical links are not included as they are not independent: they are equivalent to the two-body $Z$-type terms along the horizontal directions up to a local $Z$-type symmetry of the Hamiltonian.

The local $Z$-type symmetries of this model are generated by the product of four $Z$ operators around each vertex. Take the local $Z$-type symmetries as gauge constraints, the model is a quantum $\mathbb{Z}_2$ gauge theory, and found to arise in the system of Josephson arrays of superconductor and ferromagnet when deposited on top of a quantum spin Hall insulator [81,82].

There are two nonlocal $Z$-type symmetries, and we may take $U_1^Z$ ($U_2^Z$) to be the product of all vertical-link (horizontal-link) $Z$ operators along some horizontal (vertical) line, see Fig. 1a. There are no local $X$-type symmetries. Each $X$-type non-local symmetry generator of the model is a product of all vertical-link $X$ operators along even number of vertical lines, and all horizontal-link $X$ operators along even number of horizontal lines. See Fig. 1b for an example.

Later, we will construct a bulk theory for this model, which has the X-cube fracton order [59].

## 3 Generalized Kramers-Wannier duality

In this section, we define generalized Kramers-Wannier dual theories for each GI model. Such dual theories play an important role in our construction of the bulk models. A dual theory

lives on a generically different lattice which we dub the dual lattice. The operator map of the duality can be written as

$$\mathcal{O}_\alpha^Z \mapsto \Delta_\alpha^Z, \qquad \mathcal{O}_i^X \mapsto \Delta_i^X, \tag{4}$$

where $\Delta_\alpha^Z$ ($\Delta_i^X$) is a local product of Pauli $Z$ (Pauli $X$) operators on the dual lattice, such that the commuting or anticommuting relations between the operators are preserved. Moreover, the above operator map should be local: if we place the original and the (generically different) dual lattices together, then each $\mathcal{O}_\alpha^Z$ ($\mathcal{O}_i^X$) operator should be closed to the corresponding $\Delta_\alpha^Z$ ($\Delta_i^X$) operator. The dual model Hamiltonian then reads

$$H' = -\sum_\alpha J_\alpha \Delta_\alpha^Z - \sum_i h_i \Delta_i^X. \tag{5}$$

Such a duality exists for any GI model, because we can always let the dual lattice consist of qubits labeled by $\alpha$, and then let $\Delta_\alpha^Z = Z_\alpha$, $\Delta_i^X = \prod_{\alpha \in I_i} X_\alpha$ where $I_i$ is the set of $\mathcal{O}_\alpha^Z$ terms that anticommute with $\mathcal{O}_i^X$. This is dubbed the *standard dual theory*. We may treat the dual theory as a GI model as well, but with the roles of $X$ and $Z$ exchanged, which means we first include all $X$-type $\mathbb{Z}_2$ symmetries, and then include all *compatible* $Z$-type symmetries. Similarly as in the GI model, here, all $Z$-type symmetries are generated by products of Hamiltonian local terms $\Delta_i^Z$. We denote the local symmetry generators in the dual theory by $\{\Gamma_\rho^X\}$ and $\{\Gamma_\sigma^Z\}$. The independent nonlocal symmetry generators are denoted as $\{\Omega_1^X, \cdots, \Omega_{m_X}^X\}$ and $\{\Omega_1^Z, \cdots, \Omega_{m_Z}^Z\}$. In Appendix B, we prove that if we restrict to the symmetric sectors on both sides of the duality, then the operator map (4) follows from a Hilbert space isomorphism, i.e. an exact duality.

Similar to the original theory, we make the following assumptions for the dual theory:

- $\{\Delta_\alpha^Z\} \cup \{\Gamma_\rho^X\}$ is a CSLO.

- Any local $Z$-type symmetry generator is locally generated by $\{\Gamma_\sigma^Z\}$.

In addition, we assume that

- $n_X + m_Z \geq 1$.

In other words, either there exist compatible nonlocal $X$-type symmetries in the original model, *i.e.* $n_X \geq 1$, or there exist compatible nonlocal $Z$-type symmetries in the dual model, *i.e.* $m_Z \geq 1$. This will help ensure our bulk model to have a topological and/or fracton order. Later in the Section 5, we discuss the further conditions on the GI model (and its dual) so that the GI model has non-invertible anomaly that can be matched with the bulk model to be constructed.

For example, the standard dual theory of the standard one-dimensional Ising model is $H' = -J \sum_i Z_{i+1/2} - h \sum_i X_{i-1/2} X_{i+1/2}$, where we place the dual lattice qubits in between the original ones, reflected by the $1/2$ shifts in the indices. Another example is that the standard dual theory of *both* the two-dimensional plaquette Ising model in Eq. 2 and the model in Eq. 3 is

$$H' = -J \; \overset{Z}{\bullet} \; -h \; \begin{matrix} X\bullet\!\!-\!\!-\!\!\bullet X \\ \phantom{X}\,\big|\qquad\big|\phantom{X} \\ X\bullet\!\!-\!\!-\!\!\bullet X \end{matrix} \;, \tag{6}$$

which is nothing but the two-dimensional plaquette Ising model with the substitutions $X \leftrightarrow Z$ and $J \leftrightarrow h$. This dual theory has no local symmetry.

We emphasize that a single GI model may have multiple dual models. Just from the example above, another possible dual theory of the two-dimensional plaquette Ising model is Eq. 3 with the substitutions $X \leftrightarrow Z$ and $J \leftrightarrow h$. As a consequence, multiple bulk models may be constructed from a single GI model, as we will see.

Table 1: Summary of notations.

| | |
|---|---|
| **GI Model** $d$-dim | • $H = -\sum_\alpha J_\alpha \mathcal{O}_\alpha^Z - \sum_i h_i \mathcal{O}_i^X$<br><br>• Local Symmetries: $\{G_r^Z\}$, $\{G_s^X\}$<br><br>• Nonlocal Symmetries: $\{U_k^Z \mid 1 \le k \le n_Z\}$, $\{U_k^X \mid 1 \le k \le n_X\}$ |
| **Dual Model** $d$-dim | • $H' = -\sum_\alpha J_\alpha \Delta_\alpha^Z - \sum_i h_i \Delta_i^X$<br><br>• Local Symmetries: $\{\Gamma_\rho^X\}$, $\{\Gamma_\sigma^Z\}$<br><br>• Nonlocal Symmetries: $\{\Omega_k^X \mid 1 \le k \le m_X\}$, $\{\Omega_k^Z \mid 1 \le k \le m_Z\}$ |
| **Bulk Model** $(d+1)$-dim | • Odd Layers: Original Lattice ($\circ$)<br><br>• Even Layers: Dual Lattice ($\bullet$)<br><br>• $H_{\text{bulk}}$: Eq. 7 or Fig. 2. |

## 4  Bulk theory

Given some GI model in $d$ spatial dimensions and a dual model of it, we will now construct a bulk theory in one higher dimensions such that certain charge and boundary condition sector(s) of the GI model can live on its boundary. We will explain later what this precisely means.

### 4.1  Construction and prototype examples

The lattice on which the bulk theory lives is an alternating stack of the original and dual $d$-dimensional lattices; see Fig. 2. As an example, we also show the bulk lattice thus constructed from the standard one-dimensional Ising model and its standard dual model in Fig. 3. Here and throughout, we often use empty circles (solid dots) to represent qubits in layers of the original (dual) lattice. We label the original and dual lattice layers by odd and even indices, then our bulk theory is defined by the following Hamiltonian; see Table 1 for a recap of the many notations.

$$
\begin{aligned}
H_{\text{bulk}} = &-\sum_{\alpha,l} \Delta_{\alpha,2l}^Z \mathcal{O}_{\alpha,2l+1}^Z \Delta_{\alpha,2l+2}^Z - \sum_{i,l} \mathcal{O}_{i,2l-1}^X \Delta_{i,2l}^X \mathcal{O}_{i,2l+1}^X \\
&-\sum_{r,l} G_{r,2l+1}^Z - \sum_{s,l} G_{s,2l+1}^X - \sum_{\rho,l} \Gamma_{\rho,2l}^X - \sum_{\sigma,l} \Gamma_{\sigma,2l}^Z \,,
\end{aligned}
\tag{7}
$$

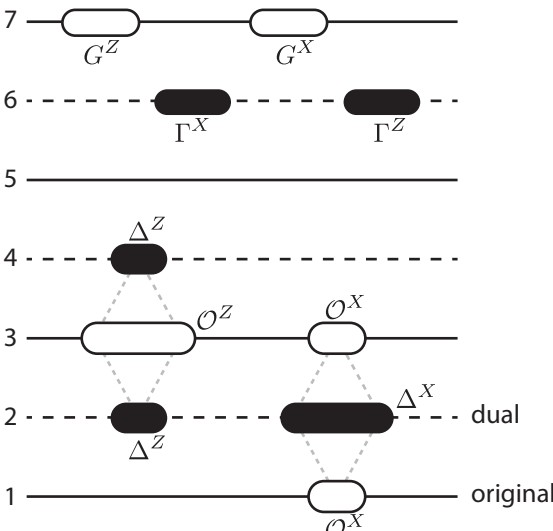

Figure 2: An illustration of the bulk model. Horizontal solid (dashed) lines represent the original (dual) lattice layers. The various operators in $H_{\text{bulk}}$ are schematically plotted.

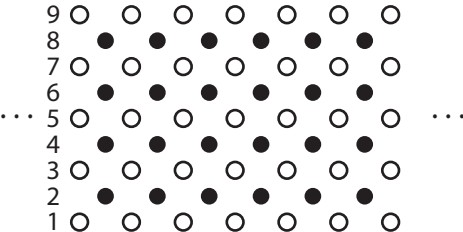

Figure 3: The 2D bulk lattice constructed from the standard one-dimensional Ising model and its standard dual model.

where the second subscript of each operator is the layer index. These various operators are schematically plotted in Fig. 2. The $\Delta^Z \mathcal{O}^Z \Delta^Z$ ($\mathcal{O}^X \Delta^X \mathcal{O}^X$) terms will be called the $Z$-suspension ($X$-suspension) terms; this name is from the special case where $\Delta^Z = Z$ ($\mathcal{O}^X = X$). The $G$ and $\Gamma$ terms will be called the gauge symmetry terms. By construction, all local operators in $H_{\text{bulk}}$ commute with each other, thus the model is exactly solvable. In other words, $H_{\text{bulk}}$ is a stabilizer Hamiltonian.

The simplest example comes out starting from the standard one-dimensional Ising model and its standard Kramers-Wannier dual. We obtain the following bulk Hamiltonian,

$$
H_{\text{bulk}} = -Z \bigcirc \overset{\overset{Z}{\bullet}}{\underset{\bullet}{\bigcirc}} \bigcirc Z - X \bullet \overset{\overset{X}{\bigcirc}}{\underset{\bigcirc}{\bigcirc}} \bullet X \,, \tag{8}
$$

which lives on the lattice shown in Fig. 3. There are no gauge symmetry terms in 8. The Hamiltonian represents nothing but the $\mathbb{Z}_2$ toric code model [45]; it can be cast to the standard form by replacing empty circles and solid dots by vertical and horizontal links, respectively. Similarly, the three-dimensional toric code model can be generated from the standard two-dimensional Ising model and its standard dual, the $\mathbb{Z}_2$ lattice gauge model without matter.

From the two-dimensional plaquette Ising model (2) and its standard dual (6), we obtain

$$H_{\text{bulk}} = -Z \cdots Z - X \cdots X \quad \left( \uparrow^z \searrow^y \rightarrow_x \right),$$ (9)

where the 3D Cartesian frame is indicated in the bracket with $z$ being the out-of-layer direction. The model is also obtainable via other constructions [83, 84]. Point excitations in this bulk model are free to move along $z$ direction, but have restricted mobility along $x$ and $y$ directions like the original two-dimensional model. In other words, the model is fractonic along $x$ and $y$, but behaves like a topologically ordered system along $z$.

Another example, from the two-dimensional model in Eq. 3 and its standard dual theory in Eq. 6, we obtain

$$H_{\text{bulk}} = - X \cdots X - Z \cdots Z - Z \cdots Z,$$ (10)

where we have associated qubits in the dual lattice layers with $z$-direction links, and the Cartesian frame is the same as that in Eq. 9. This three-dimensional model is topologically equivalent[5] to the X-cube fracton model [59]. Recall that the two-dimensional plaquette Ising model has an alternative dual theory: Eq. 3 with the substitutions $X \leftrightarrow Z$ and $J \leftrightarrow h$. The bulk theory constructed from this pair of models is the same as Eq. 10 but with $X$ and $Z$ exchanged. We have thus found that multiple bulk models may be constructed from a single GI model by choosing different dual models.

## 4.2 Robust ground state degeneracy

We now show that the general bulk theory $H_{\text{bulk}}$ has a stable spectral gap, and any possible GSD of it is robust. Therefore any degenerate ground states, say on the lattice in $d \geq 2$ dimensions with periodic boundary condition, would imply topological and/or fracton orders. Then we compute the GSD.

Regarding $H_{\text{bulk}}$ as the negative sum over a set of stabilizers, then in the ground subspace, all these stabilizers equal to $+1$, i.e. there is no frustration.[6] According to Ref. [72], the following lemma implies that the model has a spectral gap stable to local perturbations, together with either a unique ground state or a robust GSD.[7] In particular, the lemma shows that the stabilizer Hamiltonian constructed is a quantum code with macroscopic code distance. The logical operators, if any, are all non-local operators that commutes with the stabilizers.

---

[5]Comparing to the X-cube Hamiltonian, the model constructed here lacks the X-shape terms in one of the three orientations, but those absent terms can be generated by the existing X-shape terms, thus the two models are topologically equivalent.

[6]This is possible because the group generated by these stabilizers does not contain $-1$.

[7]To meet the conditions in Ref. [72], we also make the following assumptions that are usually fulfilled, and in particular are satisfied when the bulk model has translation symmetries along all directions: (1) There is an $O(1)$ upper bound on the geometric sizes of all bulk stabilizers. This leads to certain precise locality requirements on the $d$-dimensional models. (2) There is a natural way of taking thermodynamic limit for the GI model and its dual such that the lattice structure, Hamiltonian local terms, and local symmetry generators ($G$ and $\Gamma$ operators) of both models within distance $R$ from any site do not depend on the total system size, as long as the latter is not too closed to $R$. This guarantees a similar property for the lattice structure and Hamiltonian local terms of the bulk model.

**Theorem 1.** *The stabilizers in $H_{bulk}$ form a CSLO. In other words, any local operator $\mathcal{A}$ that is a tensor product of $I, X, Y, Z$ and commutes with all the stabilizers in $H_{bulk}$ can be locally generated by those stabilizers.*

*Proof.* Up to an unimportant phase factor, we can write $\mathcal{A} = \mathcal{A}_Z \mathcal{A}_X$ where $\mathcal{A}_Z$ ($\mathcal{A}_X$) is a product of Pauli $Z$ (Pauli $X$) operators on different sites. $\mathcal{A}_Z$ and $\mathcal{A}_X$ must themselves be local and commute with all the stabilizers in $H_{\text{bulk}}$, because all the stabilizers in $H_{\text{bulk}}$ are either $X$-type or $Z$-type. We will show that $\mathcal{A}_Z$ and $\mathcal{A}_X$ are both local products of the operators appearing in $H_{\text{bulk}}$.

Each $Z$ operator in $\mathcal{A}_Z$ has some integer layer index $l$. Let the maximal and minimal ones of those layer indices be $l_{\max}$ and $l_{\min}$, respectively. Suppose $l_{\max}$ is odd, i.e. coinciding with an original lattice layer. In order to commute with all the $X$-suspension terms that span the three layers $l_{\max}$, $l_{\max+1}$ and $l_{\max+2}$, and also by our assumptions on the original $d$-dimensional model, the top layer of $\mathcal{A}_Z$ must be a local product of some $G^Z$ terms. Thus we may multiply $\mathcal{A}_Z$ by those $G^Z$ terms and reduce $l_{\max}$ by at least 1. Now suppose that $l_{\max}$ is even. The top layer of $\mathcal{A}_Z$ coincides with a dual lattice layer, and commutes with all the $\Gamma^X$ operators on that layer. Given our assumptions on the dual $d$-dimensional model, it follows that the top layer of $\mathcal{A}_Z$ must be a local product of some $\Delta^Z$ operators. Let $h = l_{\max} - l_{\min}$ be the height of the $\mathcal{A}_Z$ operator. Whenever $h \geq 2$, we may multiply $\mathcal{A}_Z$ with some $Z$-suspension operators and reduce its height by at least 1. Repeat the above two operations to decrease $l_{\max}$, and the similar operations to increase $l_{\min}$. Eventually, the reduced $\mathcal{A}_Z$ operator acts on a single dual lattice layer (even layer index), if it is not yet fully reduced to the identity. This single-layer operator is a product of some $\Delta^Z$ operators, and must commute with all the $\Delta^X$ operators acting on that layer (due to the $X$-suspension operators), thus it is a local $Z$-type symmetry generator of the dual theory and is a local product of some $\Gamma^Z$ operators by our assumptions on the dual model. Given the locality of the $\mathcal{O}^X$, $\mathcal{O}^Z$, $\Delta^Z$, and $\Delta^X$ operators, as well as the locality of the generalized Kramers-Wannier operator map, the above reduction procedure implies that $\mathcal{A}_Z$ is locally generated by the stabilizers in $H_{\text{bulk}}$. The claim for $\mathcal{A}_X$ can be proved similarly. $\square$

Next, we examine in what cases the model has GSD. It turns out the GSD, for example, on a $(d + 1)$-dimensional torus, only depends on properties of non-local symmetries in both the original (generalized Ising) model and its dual. Specifically, we take periodic boundary condition along the out-of-layer direction, i.e. identify the first and the $L$-th layers for some odd $L$. We obtain the GSD through a thorough counting, which we elaborate in Appendix D.

Here instead, let us prove the existence of a degeneracy by finding a pair of anticommuting operators that act on the ground subspace. As a general mathematical fact, given a set of independent $X$-type operators $A_1^X, A_2^X, \cdots, A_n^X$ acting on an arbitrary multiple-qubit Hilbert space, there is always a set of $Z$-type operators $B_1^Z, \cdots, B_n^Z$ such that $B_k^Z$ anticommutes with $A_k^X$ but commutes with all the other $A^X$ operators.[8] This means that we can always find some operators $V_k^Z$ acting on the Hilbert space of our original $d$-dimensional lattice, such that $V_k^Z$ anticommutes with $U_k^X$ but commutes with all the other $X$-type symmetry generators (local or nonlocal). Let us then consider the operator $U_{k,2l_0-1}^X$ for some $k$ and $l_0$, which is $U_k^X$ acting on an original lattice layer $2l_0 - 1$, and the operator

$$W_k = \prod_l V_{k,2l-1}^Z, \tag{11}$$

---

[8] The $A_k^X$ operators can be represented by column vectors with elements in $\mathbb{F}_2$. The search for $B^Z$ operators is then equivalent to the search for dual vectors. Dual vectors exist because full-rank matrices with $\mathbb{F}_2$ elements are invertible.

where $V_{k,2l-1}^Z$ is $V_k^Z$ acting on the $(2l-1)$-th layer. The two operators $U_{k,2l_0-1}^X$ and $W_k$ both commute with $H_{\text{bulk}}$, thus acting within the ground state subspace, and the two operators anticommute with each other. It follows that the ground state subspace can not be one-dimensional. Indeed, let $|\psi\rangle$ be a ground state that is also an eigenstate of $U_{k,2l_0-1}^X$ with some eigenvalue $\lambda = \pm 1$, then $W_k |\psi\rangle$ is another ground state with eigenvalue $-\lambda$ under the $U_{k,2l_0-1}^X$ operator. This analysis actually implies that the $U_{k,2l_0-1}^X$ operators for some fixed $l_0$ and all $k$ can take independent eigenvalues $\pm 1$ within the ground subspace. Similarly, the $\Omega_{k,2l_0}^Z$ operators can also take independent eigenvalues within the ground subspace. Hence, the GSD of the system is at least $2^{n_X+m_Z}$. Our previous assumption $n_X + m_Z \geq 1$ guarantees a nontrivial degeneracy.

If $n_X$ increases with the system size, which, for example, happens in the plaquette Ising model and the model in Eq. 3, the bulk theory will have a fracton order, with a GSD increases with the system size. The scenario for $m_Z$ is similar.

A few simple cases are illuminating, following the formula of GSD given in Theorem 3, which we summarize in the following claims.

**Corollary 1.** *When the GI model has $n_X \geq 1$ number of (compatible) X-type non-local symmetries, the bulk model has degenerate ground states stable against local perturbations, and*

$$\log_2 \text{GSD} \geq n_X. \tag{12}$$

**Corollary 2.** *When the dual of the GI model has $m_Z \geq 1$ number of (compatible) Z-type non-local symmetries, the bulk model has degenerate ground states stable against local perturbations, and*

$$\log_2 \text{GSD} \geq m_Z. \tag{13}$$

**Corollary 3.** *When there is no compatible nonlocal symmetry in neither the GI model nor the dual model, the bulk model has a unique ground state.*

## 5 Bulk-boundary correspondence

Now let us analyze the boundary physics of the bulk model constructed above, and show that the GI model we start with can terminate on its boundary. This more precisely means the following: When the bulk model is placed on a certain space with boundary, in the absence of bulk excitation, its low-energy physics below the bulk excitation gap is described by the GI model subject to certain global constraints.

The bulk-boundary correspondence manifests locally, through the matching of Hamiltonian local terms, as well as globally. Especially, we focus on matching the following two types of global constraints on the GI model to the boundary of the bulk model.

- Global symmetry charge projection: for a set of indices $\mathcal{N} \subset \{1, 2, \cdots, n_X\}$,

$$U_{\mathcal{N}}^X \equiv \prod_{k \in \mathcal{N}} U_k^X = \pm 1. \tag{14}$$

- Generalized boundary condition: for a set of indices $\mathcal{S}$ such that $\prod_{\alpha \in \mathcal{S}} \mathcal{O}_\alpha^Z = 1$ modulo $G^Z$'s,

$$\eta_{\mathcal{S}} \equiv \prod_{\alpha \in \mathcal{S}} \text{sign}(J_\alpha) = \pm 1. \tag{15}$$

To understand why we call the latter condition a generalized boundary condition, let us apply the condition to one-dimensional tranverse Ising model on a ring. In this model, $\{\mathcal{O}_\alpha^Z\}$

can be identified with $\{Z_1 Z_2, Z_2 Z_3, \cdots, Z_N Z_1\}$, such that $\prod_\alpha \mathcal{O}_\alpha^Z = 1$, and one way to change the boundary condition is to flip the sign of the coefficient of the $Z_N Z_1$ term in the Hamiltonian; and correspondingly, the sign of $\eta = \prod_\alpha \text{sign}(J_\alpha)$ is changed.

There are two main results. Consider a bulk model with a finite number of layers, and with the top and bottom being odd layers. The first is that the boundary Hilbert space $\mathcal{L}_{\text{bdry}}$ of the bulk model with topological and/or fracton orders is isomorphic to that of two copies of GI models subject to global symmetry constraints. Particularly, under a condition to be specified below, $\mathcal{L}_{\text{bdry}}$ is isomorphic to the sector labeled by +1 eigenvalues of at least one of the following two types of non-local operators. One is

$$U_{\mathcal{N}}^X \otimes U_{\mathcal{N}}^X, \tag{16}$$

dubbed global symmetry charges, and the other is

$$\Omega_{\mathcal{M}}^Z \otimes \Omega_{\mathcal{M}}^Z, \tag{17}$$

dubbed generalized boundary conditions. The notation here deserves some explanation. $U_{\mathcal{N}}^X$ for $\mathcal{N} \subset \{1, 2, \cdots, n_X\}$ are global symmetry operators of the GI model, and hence the operator in Eq. 16 does divide the Hilbert space of two GI models into different eigenvalue sectors. On the other hand, $\Omega_{\mathcal{M}}^Z$ for $\mathcal{M} \subset \{1, 2, \cdots, m_Z\}$ are symmetry operators of the dual model, then how does the operator in Eq. 17 acts on two copies of the original model? In fact, we will show that the eigenvalues of certain $\Omega_{\mathcal{M}}^Z$ in the dual model imply generalized boundary conditions given by (15), through the duality map. Therefore, $\Omega_{\mathcal{M}}^Z \otimes \Omega_{\mathcal{M}}^Z$ actually acts on two copies of the original model as $\eta_{\mathcal{S}} \otimes \eta_{\mathcal{S}}$ for some $\mathcal{S}$.

Furthermore, we find that the boundary Hilbert space $\mathcal{L}_{\text{bdry}}$ can be divided into many sectors labeled by the eigenvalues of some *nonlocal* operators, which are all $X$-type or $Z$-type symmetry operators of the original or dual $d$-dimensional models acting on certain layers. The sectors are all isomorphic, and the boundary Hamiltonian $H_{\text{bdry}}$ is block-diagonal with respect to these sectors. In different sector, the charge projections (14) and generalized boundary conditions (15) either on the top layer or on the bottom layer may be different. Nevertheless, the combinations of their values on both the top and the bottom layer need to be consistent with that $U_{\mathcal{N}}^X \otimes U_{\mathcal{N}}^X = 1$ and $\Omega_{\mathcal{M}}^Z \otimes \Omega_{\mathcal{M}}^Z = 1$.

In this way, the boundary model of a non-invertible phase with long-range orders is described by the GI model with global constraints. When this happens, we dub the constrained GI model to have (*weak*) *non-invertible anomaly*.

The second main result is a concrete condition on the non-local symmetries $U_{\mathcal{N}}^X$, $\Omega_{\mathcal{N}}^Z$ of the original and dual $d$-dimensional models that leads to a bulk model whose boundary theory matches with the GI model with constraints (14) and/or (15). We summarize it in the following claim.

**Claim 1.** *(Necessary and sufficient condition for an anomalous boundary) The boundary theory is anomalous if and only if* either *of the following two conditions is satisfied:*

1. *For some nonempty subset $\mathcal{N} \subset \{1, 2, \cdots, n_X\}$,*

$$U_{\mathcal{N}}^X \equiv \prod_{k \in \mathcal{N}} U_k^X, \tag{18}$$

   *can be written as a product of $\mathcal{O}^X$ operators such that its dual – a product of $\Delta^X$ operators – equals to the identity modulo local symmetry operators $\Gamma^X$.*

2. *For some nonempty subset $\mathcal{M} \subset \{1, 2, \cdots, m_Z\}$,*

$$\Omega_{\mathcal{M}}^Z \equiv \prod_{k \in \mathcal{M}} \Omega_k^Z, \tag{19}$$

*can be written as a product of $\Delta^Z$ operators such that its dual – a product of $\mathcal{O}^Z$ operators – equals to the identity modulo local symmetry operators $G^Z$.*

Since that a product of a few $\mathcal{O}^Z$ equals the identity modulo $G^Z$'s is a generalized boundary condition (15) in the GI model, and there is an analogy for a product of $\Delta^X$. Thus, colloquially speaking, the conditions say that only for those non-local symmetry operators in either the GI or the dual model, which is dual to a generalized boundary conditions, the projections of them lead to the (weak) non-invertible anomaly.

In the special case where $\mathcal{O}^X = X$ and $\Delta^Z = Z$, the condition is simple, that is $n_X + m_Z \geq 1$. The reason is that in this case, any non-local symmetry $U^X$ or $\Omega^Z$ is dual to the identity, because the original (dual) theory does not have any $Z$-type ($X$-type) symmetry.

The following subsections are devoted to analyze the boundary theory from the simplest to the most general case, which leads to the results above. The examples of the boundary theory of toric code model and the X-cube model are presented. A complete and detailed treatment is given in Appendix E.

To begin with, let us define the boundary of our bulk model. We will take an odd number of layers, with layer indices from 1 to $L \in 2\mathbb{Z} + 1$, and take open boundary condition along the out-of-layer direction, so the 1-st and the $L$-th layers are the boundary layers. The two boundary layers both have the original (instead of the dual) lattice structure on which our GI model is defined, cf. Fig. 2 and 3. How about the boundary condition along the intra-layer directions? Previously, we have assumed periodic boundary condition when discussing concrete examples, but our construction does not really demand any particular boundary condition. In the following, we just require that the boundary condition for each original (dual) lattice layer be the same as the original (dual) $d$-dimensional model, but is otherwise arbitrary. However, we emphasize that if one changes the boundary condition for a GI model, its symmetry, the dual model, and the validity of our assumptions should all be reexamined. An example will be given below.

We define the bulk Hamiltonian $H_{\text{bulk}}$ to be of the same form as (7), including all the terms that are completely inside the system. This Hamiltonian determines a degenerate ground state subspace, which we consider as the boundary Hilbert space $\mathcal{L}_{\text{bdry}}$. All additional local operators that commute with local terms in $H_{\text{bulk}}$, and thus act within the boundary Hilbert space $\mathcal{L}_{\text{bdry}}$, are allowed terms in the most general boundary Hamiltonian $H_{\text{bdry}}$. $\mathcal{L}_{\text{bdry}}$ together with $H_{\text{bdry}}$ is the boundary theory that we are going to determine. Note that there is not a unique choice of $H_{\text{bdry}}$, since the product of any two boundary terms is another allowed boundary term. Instead, we will focus on a canonical choice of $H_{\text{bdry}}$. We prove in Appendix E.5 that the boundary terms given in the canonical choice together with the stabilizers in the bulk Hamiltonian are sufficient to generate any allowed boundary local term. Thus the canonical $H_{\text{bdry}}$ we consider is a quite general one.

## 5.1 Simplest situation: $\mathcal{O}^X = X$ and $\Delta^Z = Z$

Let us start with the simplest situation where $\mathcal{O}_i^X = X_i$ and $\Delta_\alpha^Z = Z_\alpha$, with $i$ and $\alpha$ labeling qubits in the original and dual lattices, respectively. In this case, the original model (the dual model) has no $Z$-type ($X$-type) symmetry at all. With periodic boundary condition along the out-of-layer direction, the GSD is determined by the number of non-local $X$-type symmetries in the original model and the number of non-local $Z$-type symmetries in the dual model, independent of the number of layers, $\log_2 \text{GSD} = n_X + m_Z$.

Note that one obvious type of operators that commute with $H_{\text{bulk}}$ is the nonlocal $Z$-type symmetry operator $\Omega_{k,2l_0}^Z$ in any even layer. Thus, we can divide $\mathcal{L}_{\text{bdry}}$ into several sectors labeled by the eigenvalues of the nonlocal operators $\Omega_{k,2l_0}^Z$ ($k = 1, \cdots, m_Z$) for some fixed $l_0$.

Notice that under the Kramers-Wannier duality, $\Omega_k^Z$ from the dual theory corresponds to the identity operator of the original theory. Hence, $\Omega_{k,2l_0}^Z$ is related to $\Omega_{k,2l}^Z$ for any other $l$ by the multiplications of several $Z$-suspension operators. More explicitly, $\Omega_k^Z = \prod_{\alpha \in A} Z_\alpha$ for some set $A$ such that $\prod_{\alpha \in A} \mathcal{O}_\alpha^Z = 1$, then $\prod_{\alpha \in A} Z_{\alpha,2l} \mathcal{O}_{\alpha,2l+1}^Z Z_{\alpha,2l+2} = \Omega_{k,2l}^Z \Omega_{k,2l+2}^Z$. This is the reason that we only need to consider $\Omega_{k,2l_0}^Z$ operators acting on a single layer. Let $\mathcal{L}_{\text{bdry},0} \subset \mathcal{L}_{\text{bdry}}$ be the particular sector where $\Omega_{k,2l_0}^Z = 1$ for all $k$. Denote by $\mathcal{L}$ the Hilbert space for our original $d$-dimensional lattice and by $\mathcal{L}_G$ the gauge invariant subspace of it (a.k.a., the symmetric subspace for all local symmetries).

We claim that $\mathcal{L}_{\text{bdry},0}$ is isomorphic to the following fictitious space,

$$\mathcal{L}_{\text{fic}} := \left\{ |\phi\rangle \in \mathcal{L}_G \otimes \mathcal{L}_G \,\middle|\, U_i^X \otimes U_i^X |\phi\rangle = |\phi\rangle, \quad i = 1, \cdots, n_X \right\}. \tag{20}$$

where the two copies of $\mathcal{L}_G$ represent the two boundary layers of our physical system. That is, $\mathcal{L}_{\text{bdry},0}$ is the $\mathbb{Z}_2^{n_X}$ symmetric sector of $\mathcal{L}_G \otimes \mathcal{L}_G$ under the symmetry generated by $U_i^X \otimes U_i^X$, $i = 1, \cdots, n_X$.

Furthermore, $\mathcal{L}_{\text{bdry},0}$ is an invariant subspace of $H_{\text{bdry}}$ whose action in this sector, when represented in $\mathcal{L}_{\text{fic}}$, can take the form

$$H_{\text{GI}}^{\text{I}}(J_\alpha, h_i) + H_{\text{GI}}^{\text{II}}(J_\alpha', h_i'), \tag{21}$$

where $H_{\text{GI}}^{\text{I}}$ and $H_{\text{GI}}^{\text{II}}$ act on the two copies of $\mathcal{L}_G$ in Eq. 20, respectively.

Let us understand the result of the boundary Hilbert space first. Note that the Pauli-$X$ operator acting on any qubit in the two boundary layers commute with the bulk Hamiltonian. States in $\mathcal{L}_{\text{bdry},0}$ can be labeled by the eigenvalues of these Pauli-$X$ operators, subject to the following two constraints. First, each local $X$-type symmetry generator equals to 1, since the generator is a local term in the bulk Hamiltonian. This constraint gives rise to the gauge invariance requirement in Eq. 20. Second, since $U_k^X$ is dual to the identity operator under the Kramers-Wanner duality,[9] $U_{k,1}^X U_{k,L}^X$ is equal to the product of several $X$-suspension operators, and thus, is equal to 1. This constraint leads to the $\mathbb{Z}_2^{n_X}$ symmetry projection.

Now we consider the boundary Hamiltonian local terms. The Pauli-$X$ operators on the two boundary layers are allowed, since, as just mentioned, they commute with the bulk Hamiltonian. Furthermore, these operators commute with $\Omega_{k,2l_0}^Z$, and thus act within each sector of the boundary Hilbert space. Under the isomorphism from $\mathcal{L}_{\text{bdry},0}$ to $\mathcal{L}_{\text{fic}}$, these operators take the same form,

$$X_{i,1} \mapsto X_i \otimes 1, \qquad X_{i,L} \mapsto 1 \otimes X_i. \tag{22}$$

Another set of operators that can be added to $H_{\text{bdry}}$ are $\mathcal{O}_{\alpha,1}^Z Z_{\alpha,2}$ and $Z_{\alpha,L-1} \mathcal{O}_{\alpha,L}^Z$. They all commute with the bulk Hamiltonian and with $\Omega_{k,2l_0}^Z$ as well. Their image in $\mathcal{L}_{\text{fic}}$ is,

$$\mathcal{O}_{\alpha,1}^Z Z_{\alpha,2} \mapsto \mathcal{O}_\alpha^Z \otimes 1, \qquad Z_{\alpha,L-1} \mathcal{O}_{\alpha,L}^Z \mapsto 1 \otimes \mathcal{O}_\alpha^Z. \tag{23}$$

This map may seem obvious since it preserves the commuting/anticommuting relations with the boundary $X$ operators, but a careful proof is actually necessary. For example, extra minus sign factors also seem allowed and it is not immediately clear whether they can be gauged away. Our proof for this result is given in Appendix E.2. A crucial ingredient in the proof is that $\Omega_{k,2l_0}^Z = 1$ for all $k$; one can see this by noticing that each $\Omega_{k,2}^Z$ is equal to the product of several $\mathcal{O}_{\alpha,1}^Z Z_{\alpha,2}$.

With the above analysis, we conclude that the $\mathcal{L}_{\text{bdry},0}$ block of $H_{\text{bdry}}$, when represented in $\mathcal{L}_{\text{fic}}$, may take the form (21), where $H_{\text{GI}}^{\text{I}}$ and $H_{\text{GI}}^{\text{II}}$ act on the two copies of $\mathcal{L}_G$ in Eq. 20,

---

[9] $U_k^X$ is a product of several $X_i$ operators. One can obtain a dual operator by applying the Kramers-Wannier operator map (4). Such a dual operator has to commute with all the $Z_\alpha$ operators, so it must be the identity.

respectively. We will refer to Eq. 21 as the *effective* boundary Hamiltonian in the $\mathcal{L}_{\mathrm{bdry},0}$ sector, where the "effectiveness" is in the sense that the Hamiltonian acts on the fictitious space $\mathcal{L}_{\mathrm{fic}}$.

Other sectors with different eigenvalues of $\Omega_{k,2l_0}^Z$ can be analyzed by considering unitary operators that map them to $\mathcal{L}_{\mathrm{bdry},0}$. Denote by $\mathcal{L}'$ the Hilbert space for the $d$-dimensional dual lattice. We can find some $X$-type operators $\Theta_k^X$ acting on $\mathcal{L}'$ such that $\Theta_k^X$ anticommutes with $\Omega_k^Z$ but commutes with all the other $Z$-type symmetry generators (local or nonlocal); this is always possible as we mentioned earlier. It follows that

$$\prod_{l=1}^{(L-1)/2} \Theta_{k,2l}^X \,,$$

is an operator that commutes with the bulk Hamiltonian and can flip the eigenvalue of $\Omega_{k,2l_0}^Z$. Hence, we have found that each of the $m_Z$ sectors of $\mathcal{L}_{\mathrm{bdry}}$ is isomorphic to $\mathcal{L}_{\mathrm{fic}}$ defined in Eq. 20. The above unitary operator that can alter the sign of $\Omega_{k,2l_0}^Z$ commutes with all the $\mathcal{O}^X = X$ operators on the two physical boundaries, but necessarily anticommute with some $\mathcal{O}_{\alpha,1}^Z Z_{\alpha,2}$ and $Z_{\alpha,L-1}\mathcal{O}_{\alpha,L}^Z$ operators.

In consequence, the effective boundary Hamiltonian in each of the other sectors still takes the form of Eq. 21, but the signs of some GI terms in both $H_{\mathrm{GI}}^{\mathrm{I}}$ and $H_{\mathrm{GI}}^{\mathrm{II}}$ are flipped compared to those in the $\mathcal{L}_{\mathrm{bdry},0}$ sector.

Crucially, such sign changes *cannot* be canceled by any unitary rotation in $\mathcal{L}_{\mathrm{fic}}$. To see this, we write $\Omega_k^Z = \prod_{\alpha \in A} Z_\alpha$ for some subset $A$. Then under the generalized Kramers-Wannier duality map, $\Omega_k^Z \leftrightarrow \prod_{\alpha \in A} \mathcal{O}_\alpha^Z = 1$.[10] It leads to that in an arbitrary sector of $\mathcal{L}_{\mathrm{bdry}}$,

$$\Omega_{k,2l_0}^Z = \Omega_{k,2}^Z = \prod_{\alpha \in A}(\mathcal{O}_{\alpha,1}^Z Z_{\alpha,2}) \,. \tag{24}$$

In $\mathcal{L}_{\mathrm{fic}}$, we have

$$1 = \prod_{\alpha \in A}(\mathcal{O}_\alpha^Z \otimes 1) \,. \tag{25}$$

Suppose there is an isomorphism from this sector of $\mathcal{L}_{\mathrm{bdry}}$ to $\mathcal{L}_{\mathrm{fic}}$, such that

$$\mathcal{O}_{\alpha,1}^Z Z_{\alpha,2} \mapsto \eta_\alpha \mathcal{O}_\alpha^Z \otimes 1 \quad (\eta_\alpha = \pm 1) \,, \tag{26}$$

then we necessarily have

$$\Omega_{k,2l_0}^Z = \prod_{\alpha \in A} \eta_\alpha \,. \tag{27}$$

It means that as we go from one sector to another with a different $\Omega_{k,2l_0}^Z$, some of the $\eta_\alpha$ must change signs! A similar statement holds for the $Z_{\alpha,L-1}\mathcal{O}_{\alpha,L}^Z$ operators. The Hamiltonian in this sector, is isomorphic to the following one in $\mathcal{L}_{\mathrm{fic}}$,

$$H_{\mathrm{GI}}^{\mathrm{I}}(\eta_\alpha J_\alpha, h_i) + H_{\mathrm{GI}}^{\mathrm{II}}(\eta_\alpha J_\alpha', h_i') \,. \tag{28}$$

We have seen that $H_{\mathrm{bdry}}$ has a block-diagonal action on $\oplus_a \mathcal{L}_{\mathrm{bdry},a}$, where $a$ is the sector index. In fact, there are no local operators (but only non-local ones) that can map between sectors, as well as commuting with all bulk Hamiltonian local terms. This is because any local operator commuting with the bulk Hamiltonian local terms can be generated by the set of boundary local terms considered above, as we mentioned previously and proved in Appendix E.5.

---

[10]This comes from the fact that $\prod_{\alpha \in A} \mathcal{O}_\alpha^Z$ should commute with all Hamiltonian local terms in the GI model, yet the model, in which $O^X = X$, has no $Z$-type symmetry .

Let us now discuss some examples. Consider the two-dimensional toric code model constructed in Eq. 8 as a bulk theory for the standard one-dimensional Ising model with *periodic boundary condition*. Following our prescription, we shall take open boundary condition along the vertical direction, while keep periodic boundary condition along the horizontal direction; see Fig. 3. The Ising model has only one $U^X$ operator, given by $U^X = \prod_i X_i$, and similarly, its standard dual model has only one $\Omega^Z$ operator given by $\Omega^Z = \prod_i Z_{i+1/2}$. Thus, from our discussion above, $\mathcal{L}_{\text{bdry}}$ can be divided into two sectors with $\Omega^Z_{2l_0} = \pm 1$. Each of the two sectors can be regarded as two spin chains, subject to the symmetry condition $U^X \otimes U^X = 1$. The boundary Hamiltonian in one of the two sectors may be the sum of two Ising model Hamiltonians acting on the two fictitious spin chains, respectively, and with periodic boundary condition. Then the boundary Hamiltonian in the other sector will again be the sum of two Ising model Hamiltonians, but now with antiperiodic boundary condition, i.e. one Ising term on each of the two spin chains changes its sign.

To give a complementary perspective, we may alternatively start from the standard one-dimensional Ising model with *open boundary condition*, namely

$$H = -J \sum_{i=1}^{N-1} Z_i Z_{i+1} - h \sum_{i=1}^{N} X_i, \tag{29}$$

defined on a chain of $N$ spins labeled by $1, 2, \cdots, N$. Again, the model has one $\mathbb{Z}_2$ symmetry generator $U^X = \prod_{i=1}^{N} X_i$. Its standard dual theory is

$$H' = -J \sum_{i=1}^{N-1} Z_{i+1/2} - h \left( X_{3/2} + \sum_{i=2}^{N-1} X_{i-1/2} X_{i+1/2} + X_{N-1/2} \right), \tag{30}$$

defined on a chain of $N-1$ spins labeled by $3/2, 5/2, \cdots, N-1/2$. This dual model has *no $\mathbb{Z}_2$* symmetry at all! One can construct the bulk theory accordingly, which now lives on a lattice with left and right boundaries. We can take open boundary condition along the vertical direction as well, and analyze its boundary theory with the result established above. We see that the boundary Hilbert space contains only one sector, and can be regarded as two disconnected open spin chains under a $\mathbb{Z}_2$ symmetry projection $U^X \otimes U^X = 1$. The boundary Hamiltonian may take the form of an Ising Hamiltonian on each of the two chains. The two effective open spin chains are not connected because our formalism does not allow any boundary terms on the left and right boundaries. This is actually just a matter of choice. We may redefine the bulk Hamiltonian by removing certain terms near the left and right boundaries. This will enlarge the boundary Hilbert space a bit, and allow boundary terms acting on the left and right boundaries. One can check that, with a rectangular geometry, the low-energy physics of the toric code model can be a one-dimensional Ising model defined on a closed chain with periodic boundary condition and the $\mathbb{Z}_2$ even projection, as discussed in Refs. [36, 37, 47].

We have seen that when $n_X \geq 1$, there are the symmetry projections $U^X_k \otimes U^X_k = 1$, $k = 1, \cdots, n_X$. When $m_Z \geq 1$, the boundary conditions for $H^{\text{I}}_{\text{GI}}$ and $H^{\text{II}}_{\text{GI}}$ in the effective boundary Hamiltonian will simultaneously change as we alter the values of the $\Omega^Z_{k,2l_0}$ operators. Either phenomenon implies the boundary theory to be anomalous. Conversely, when $n_X + m_Z = 0$, the whole boundary Hilbert space $\mathcal{L}_{\text{bdry}}$ is simply isomorphic to $\mathcal{L}_G \otimes \mathcal{L}_G$, on which the effective boundary Hamiltonian takes the form of Eq. 21, or more generally consists of local terms generated by those in Eq. 21. This is a nonanomalous theory. Therefore, we reach the following conclusion.

**Necessary and sufficient condition for an anomalous boundary.** In the special case where $\mathcal{O}^X = X$ and $\Delta^Z = Z$, non-invertible anomaly on the boundary exists if and only if

$$n_X + m_Z \geq 1, \tag{31}$$

or equivalently, the GSD of the bulk model with periodic boundary condition along the out-of-layer direction satisfies

$$\text{GSD} > 1 \, . \tag{32}$$

That is, we can always build a bulk model on a lattice with odd layers, such that its boundary has non-invertible anomaly, that can be matched by a GI model with distinct sectors of Hilbert space. The equivalent condition (32) follows from that in the case $\mathcal{O}^X = X$ and $\Delta^Z = Z$, $\text{GSD} = 2^{n_X + m_Z}$.

## 5.2 Less simple situation: $\Delta^Z = Z$

Next, we consider the less simple situation where $\mathcal{O}_i^X$ are general but $\Delta_\alpha^Z = Z_\alpha$. That is, we adopt the standard dual theory. This includes the model in Eq. 10 with the X-cube fracton order that we constructed as a bulk theory for Eq. 3.

The analysis of the boundary theory is similar to the simplest case. We track how non-local operators $\Omega_m^Z$, $U_n^Z$ and $U_{n'}^X$ manifest on the boundary $\mathcal{L}_{\text{bdry}}$.

Again, we divide $\mathcal{L}_{\text{bdry}}$ into different sectors. These sectors are now labeled by the eigenvalues of not only $\Omega_{k,2l_0}^Z$ ($k = 1, \cdots, m_Z$), but also $U_{k,2l-1}^Z$ ($k = 1, \cdots, n_Z$) for all the *internal* layers, namely $3 \leq 2l - 1 \leq L - 2$. The $\mathcal{L}_{\text{bdry},0}$ sector, defined by $\Omega_{k,2l_0}^Z = 1$ and $U_{k,2l-1}^Z = 1$ for all the internal layers, is isomorphic to a fictitious space of the same form as Eq. 20. Now, states in $\mathcal{L}_{\text{bdry},0}$ are labeled by the eigenvalues of all the $\mathcal{O}^X$ and $U^Z$ operators acting on the two boundary layers, subject to the constraints $G_{s,1}^X = G_{s,L}^X = 1$ and $U_{k,1}^X U_{k,L}^X = 1$. As in the previous situation, $U_{k,1}^X U_{k,L}^X$ is generated by the $X$-suspension operators. The operator map from $\mathcal{L}_{\text{bdry},0}$ to $\mathcal{L}_{\text{fic}}$ is again

$$\begin{aligned}
\mathcal{O}_{i,1}^X &\mapsto \mathcal{O}_i^X \otimes 1 \, , & \mathcal{O}_{i,L}^X &\mapsto 1 \otimes \mathcal{O}_i^X \, , \\
\mathcal{O}_{\alpha,1}^Z Z_{\alpha,2} &\mapsto \mathcal{O}_\alpha^Z \otimes 1 \, , & Z_{\alpha,L-1} \mathcal{O}_{\alpha,L}^Z &\mapsto 1 \otimes \mathcal{O}_\alpha^Z \, ,
\end{aligned} \tag{33}$$

thus the $\mathcal{L}_{\text{bdry},0}$ block of $H_{\text{bdry}}$ takes the same form as Eq. 21.

Other sectors of $\mathcal{L}_{\text{bdry}}$ can again be analyzed by establishing isomorphisms to $\mathcal{L}_{\text{bdry},0}$, and thus to $\mathcal{L}_{\text{fic}}$. The eigenvalue of each $\Omega_{k,2l_0}^Z$ can be adjusted without affecting any $U_{k,2l-1}^Z$ by the operator $\prod_{l=1}^{(L-1)/2} \Theta_{k,2l}^X$ whose definition is the same as that in Section 5.1. The eigenvalues of the internal-layer $U^Z$ operators can be altered with some $X$-type operators that commute with not only the bulk Hamiltonian, but also with $\mathcal{O}_{\alpha,1}^Z Z_{\alpha,2}$ and $Z_{\alpha,L-1} \mathcal{O}_{\alpha,L}^Z$, and thus act trivially on the effective boundary Hamiltonian. That is, eigenvalues of $U^Z$ labels extra degeneracy of the boundary model that are unrelated to symmetry charge projections or boundary conditions. More explicit description of such operators is given in Appendix E.2.

Suppose for a nonempty subset $\mathcal{M} \subset \{1, 2, \cdots, m_Z\}$, $\Omega_{\mathcal{M}}^Z \equiv \prod_{k \in \mathcal{M}} \Omega_k^Z$ is dual to the identity modulo the $G^Z$ operators. One can show that altering the eigenvalue of $\Omega_{\mathcal{M},2l_0}^Z$ will necessarily flip the signs of some $\mathcal{O}^Z$ terms in both $H_{\text{GI}}^{\text{I}}$ and $H_{\text{GI}}^{\text{II}}$, with a proof essentially the same as that in Section 5.1. We also prove in Appendix E.2 that, if such $\mathcal{M}$ does not exist, and at the same time $n_X = 0$, the boundary theory is nonanomalous. That is to say that the boundary theory in this case is a direct sum of *identical* sectors. The Hilbert space of each sector is isomorphic to $\mathcal{L}_G \otimes \mathcal{L}_G$ with the operator mapping rule in Eq. 33. The effective boundary Hamiltonian of each sector takes the form of Eq. 21, or more generally consists of local terms generated by those in Eq. 21.

**Necessary and sufficient condition for an anomalous boundary.** We conclude that, in the special case where $\Delta^Z = Z$, the boundary theory is anomalous if and only if *either* of the following two conditions is satisfied:

1. $n_X \geq 1$, so that there are the symmetry charge constraints $U_k^X \otimes U_k^X = 1$.

Figure 4: Examples of (a) a $U^Z$ operator, (b) a $U^X$ operator, and (c) an $\Omega^Z$ operator of the bulk model in Eq. 10, all viewed from $z$ direction.

2. For some nonempty subset $\mathcal{M} \subset \{1, 2, \cdots, m_Z\}$, $\prod_{k \in \mathcal{M}} \Omega^Z_k$ is dual to the identity modulo the $G^Z$ operators.

**The boundary of the X-cube model.** Now we illustrate with the example of the model in Eq. 10, which describes the X-cube fracton order, and is constructed from Eq. 3 and its standard dual in Eq. 6. We put the model on a 3D cubic lattice with $L_x \times L_y \times L_z$ number of *vertices*, with periodic boundary condition along $x$ and $y$, and open boundary condition along $z$. The two boundary surfaces are "smooth", and $L$ is related to $L_z$ by $L = 2L_z - 1$ (the height of the system is $L_z - 1$ number of lattice constants). We label the lattice vertices by integer coordinates $\mathbf{r} = (x, y, z) \in \mathbb{Z}^3$ such that $x \sim x + L_x$, $y \sim y + L_y$, and $1 \leq z \leq L_z$. We denote by $Z(\mathbf{r}; \Delta\mathbf{r})$ the Pauli $Z$ operator acting on the link connecting the two neighboring vertices $\mathbf{r}$ and $\mathbf{r} + \Delta\mathbf{r}$ with $\Delta\mathbf{r} = \hat{x}, \hat{y}, \hat{z}$; similar for Pauli $X$ operators. Symmetries of the two-dimensional models have been described in words previously. A careful analysis of degeneracy relations shows that $n_Z = 2$, $n_X = L_x + L_y - 2$, $m_X = 0$, and $m_Z = L_x + L_y - 1$. More explicitly, the $U^Z$ operators acting on the $(2z-1)$-th layer can be chosen as

$$\prod_{x=1}^{L_x} Z(x, y_0, z; \hat{y}) \quad \text{for some } y_0, \tag{34}$$

$$\text{and} \quad \prod_{y=1}^{L_y} Z(x_0, y, z; \hat{x}) \quad \text{for some } x_0. \tag{35}$$

See Fig. 4a for an example. The $U^X$ operators acting on the $(2z-1)$-th layer can be chosen as

$$\prod_{x=1}^{L_x} X(x, y, z; \hat{x}) X(x, y+1, z; \hat{x}) \quad (y = 1, 2, \cdots, L_y - 1), \tag{36}$$

$$\text{and} \quad \prod_{y=1}^{L_y} X(x, y, z; \hat{y}) X(x+1, y, z; \hat{y}) \quad (x = 1, 2, \cdots, L_x - 1), \tag{37}$$

where we have excluded $y = L_y$ in (36) and $x = L_x$ in (37) because they are not independent. See Fig. 4b for an example. The $\Omega^Z$ operators acting on the $2z_0$-th layer can be chosen as

$$\prod_{x=1}^{L_x} Z(x, y, z_0; \hat{z}) \quad (y = 1, 2 \cdots, L_y), \tag{38}$$

$$\text{and} \quad \prod_{y=1}^{L_y} Z(x, y, z_0; \hat{z}) \quad (x = 1, 2 \cdots, L_x - 1), \tag{39}$$

where we have excluded $x = L_x$ in (39) because it is not independent. See Fig. 4c for an example. According to our general theory, $\mathcal{L}_{\text{bdry}}$ can be divided into $2^{m_Z + n_Z(L-3)/2}$ number of

sectors. The boundary theory in $\mathcal{L}_{\text{bdry},0}$ may be two copies of the model in Eq. 3, subject to the $n_X$ number of symmetry projections $U_k^X \otimes U_k^X = 1$. Other sectors are all isomorphic to $\mathcal{L}_{\text{bdry},0}$, and the isomorphisms may flip the signs of some GI terms in both $H_{\text{GI}}^{\text{I}}$ and $H_{\text{GI}}^{\text{II}}$ in the effective boundary Hamiltonian. More explicitly, the eigenvalue of each $U^Z$ operator acting on the $(2z-1)$-th layer with $3 \leq 2z-1 \leq L-3$ can be independently flipped by the operators

$$\prod_{y=1}^{L_y} X(x_1, y, z; \hat{y}) \quad \text{for any fixed } x_1, \tag{40}$$

$$\text{and} \quad \prod_{x=1}^{L_x} X(x, y_1, z; \hat{x}) \quad \text{for any fixed } y_1, \tag{41}$$

which correspond to the two operators in (34) and (35), respectively. It is not generically true that the eigenvalues of the internal-layer $U^Z$ operators can be adjusted with single-layer operators as in this example, but it is indeed true that these adjustment operators can be chosen to commute with $H_{\text{bdry}}$. Given some fixed even layer $2z_0$, one can independently flip the signs of the $\Omega^Z$ operators acting on this layer by the string operators

$$\prod_{z=1}^{L_z-1} X(L_x, y, z; \hat{z}) \quad (y = 1, 2, \cdots, L_y), \tag{42}$$

$$\text{and} \quad \prod_{z=1}^{L_z-1} X(L_x, y_2, z; \hat{z}) X(x, y_2, z; \hat{z}) \quad (x = 1, 2, \cdots, L_x - 1, \ y_2 \text{ arbitrary}), \tag{43}$$

which are in one-to-one correspondence with the operators in (38) and (39). Each of the above anticommutes with some terms in $H_{\text{bdry}}$. For example, the string operator $\prod_{z=1}^{L_z-1} X(L_x, y, z; \hat{z})$ anticommutes with $Z(L_x - 1, y, 1; \hat{x}) Z(L_x, y, 1; \hat{x}) Z(L_x, y, 1; \hat{z})$ and $Z(L_x - 1, y, L_z; \hat{x}) Z(L_x, y, L_z; \hat{x}) Z(L_x, y, L_z - 1; \hat{z})$. One can verify that the two conditions for anomaly are both satisfied, thus the boundary theory of this model is indeed anomalous.

## 5.3 The most general situation

Now we briefly discuss the most general situation: no further assumption on either $\mathcal{O}^X$ or $\Delta^Z$.

We may again divide $\mathcal{L}_{\text{bdry}}$ into several sectors, which are now labeled by the eigenvalues of $U_{k,2l-1}^Z$ for all internal layers ($3 \leq 2l-1 \leq L-2$), $\Omega_{k,2l}^X$ for all $l$, and $\Omega_{k,2l_0}^Z$ for some fixed layer $2l_0$, as they are non-local operators commuting with the bulk Hamiltonian. A caveat is that the eigenvalues of either the $U^Z$ operators or the $\Omega^X$ operators may not be totally independent. It may happen that a product of several $Z$-suspension operators centered on some internal odd layer $2l-1$ equals to a nonlocal $Z$-type symmetry generator (independent of $G^Z$'s) acting on that layer. This will induce some relation between the $U^Z$ operators on the same internal layer. In terms of the $d$-dimensional GI model, this means that a product of several $\mathcal{O}^Z$ operators equals to a nonlocal $Z$-type symmetry generator (independent of $G^Z$'s) and is dual to the identity. A similar possibility exists for the $\Omega^X$ operators. These possible degeneracy relations reduce the apparent number of sectors in $\mathcal{L}_{\text{bdry}}$. Without loss of generality, we may assume there are some integers $\nu$ and $\mu$, such that the many sectors of $\mathcal{L}_{\text{bdry}}$ are labeled by the *independent* eigenvalues of $U_{k>\nu,2l-1}^Z$ for all internal layers, $\Omega_{k>\mu,2l}^X$ for all $l$, and $\Omega_{k,2l_0}^Z$.

As before, we define $\mathcal{L}_{\text{bdry},0}$ to be the sector where the $U^Z$, $\Omega^X$ and $\Omega^Z$ operators just mentioned all equal to 1. One can show that $\mathcal{L}_{\text{bdry},0}$ is again isomorphic to the fictitious space in Eq. 20 with a similar operator mapping:

$$\begin{aligned} \mathcal{O}_{i,1}^X &\mapsto \mathcal{O}_i^X \otimes 1, & \mathcal{O}_{i,L}^X &\mapsto 1 \otimes \mathcal{O}_i^X, \\ \mathcal{O}_{\alpha,1}^Z \Delta_{\alpha,2}^Z &\mapsto \mathcal{O}_\alpha^Z \otimes 1, & \Delta_{\alpha,L-1}^Z \mathcal{O}_{\alpha,L}^Z &\mapsto 1 \otimes \mathcal{O}_\alpha^Z. \end{aligned} \tag{44}$$

The $\mathbb{Z}_2^{n_X}$ symmetry projection exists because $U_{k,1}^X U_{k,L}^X$ can be generated by the $X$-suspension operators, the $\Gamma^X$ operators, and the $\Omega^X$ operators.

Other sectors are all isomorphic to $\mathcal{L}_{\mathrm{bdry},0}$, and thus to $\mathcal{L}_{\mathrm{fic}}$. The discussions for the $\Omega^Z$ and internal-layer $U^Z$ operators turn out to be very similar to the $\Delta^Z = Z$ case, and will not be repeated. The eigenvalue of $\Omega_{k>\mu,2l}^X$ can be adjusted with a $Z$-type operator that may anticommute with some $\mathcal{O}_{i,1}^X$ operators, and thus may flip the signs of some $\mathcal{O}^X$ terms in $H_{\mathrm{GI}}^{\mathrm{I}}$ while leaving $H_{\mathrm{GI}}^{\mathrm{II}}$ invariant. We refer the readers to Appendix E.3 for a more explicit description of those isomorphisms.

The change of eigenvalues of $\Omega_{k>\mu,2l}^X$ may conflict with the global conditions $U_{k'}^X \otimes U_{k'}^X = 1$ in the following sense. Changing the signs of certain $\mathcal{O}^X$ terms in a GI model is equivalent to altering some $X$-type symmetry charges, since $U_{k'}^X$ for all $1 \le k' \le n_X$ is a product of several $\mathcal{O}^X$ operators.[11] In order to have an anomalous boundary, we expect some of the symmetry charge projections, such as $U_{k'}^X \otimes U_{k'}^X = 1$ for some $k'$, should hold in all sectors of the boundary theory.

Fortunately, we prove in Appendix E.3 the following result: If for a nonempty subset $\mathcal{N} \subset \{1, 2, \cdots, n_X\}$, $U_{\mathcal{N}}^X \equiv \prod_{p \in \mathcal{N}} U_p^X$ can be written as a product of $\mathcal{O}^X$ operators such that the product is dual to the identity modulo the $\Gamma^X$ operators, then altering the eigenvalue of $\Omega_{k>\mu,2l}^X$ does not affect the corresponding symmetry charge projection $U_{\mathcal{N}}^X \otimes U_{\mathcal{N}}^X = 1$. A clue for the claim is that $U_{\mathcal{N},1}^X U_{\mathcal{N},L}^X$ is a product of the $X$-suspension and $\Gamma^X$ operators, and thus does not depend on the value of $\Omega_{k>\mu,2l}^X$.

The necessary and sufficient condition for an anomalous boundary in the most general situation is given in Claim 1. The first sufficient condition about $U^X$ operators is discussed above. The second sufficient condition about $\Omega^Z$ operators is a direct generalization of the one in the previous subsection and can be proved analogously. When both sufficient conditions are violated, we prove in Appendix E.3 that the boundary theory is nonanomalous. More precisely, the boundary theory in this case is a direct sum of *identical* sectors.[12] The Hilbert space of each sector is isomorphic to $\mathcal{L}_G \otimes \mathcal{L}_G$ with the operator mapping rule in Eq. 44. The effective boundary Hamiltonian of each sector takes the form of Eq. 21, or more generally consists of local terms generated by those in Eq. 21.

We would like to also remind the readers that each $U^X$ ($\Omega^Z$) operator can always be expanded as a product of some $\mathcal{O}^X$ ($\Delta^Z$) operators, but there may be multiple ways of doing it. The first (second) condition in Claim 1 holds as long as there is one expansion of $U_{\mathcal{N}}^X$ ($\Omega_{\mathcal{M}}^Z$) in terms of the $\mathcal{O}^X$ ($\Delta^Z$) operators such that the requirement is satisfied.

# 6  Examples

We have shown that our basic construction can generate a bulk model with prototypes of topological orders, such as $\mathbb{Z}_2$ topological order in any dimensions greater than 2 and the X-cube model. Now let us explore further examples. They exploit the capacity of our construction: (1) the construction can produce lattice gauge theories whose gauge group is beyond $\mathbb{Z}_2$, (2) the construction can provide a bulk topological order from a GI model describing a symmetry protected topological (SPT) phase, (3) the construction can provide bulk topological orders with only quasi-loop excitations, (4) the same GI model can be matched with more than one

---

[11]Think about flipping the sign of a single transverse-field term in the one-dimensional transverse field Ising model. The sign-flip can be undone via a basis rotation. Nevertheless, the rotation anti-commutes with the $\mathbb{Z}_2$ symmetry generator.

[12]Note that each (new) sector here may be the sum of several (old) sectors discussed above with different values of $U_k^X \otimes U_k^X$.

bulk model with distinct fracton orders. In other words, when fracton order is present, the boundary cannot determine a unique bulk.

## 6.1 Bulk $\mathbb{Z}_2 \times \mathbb{Z}_2$ topological order in two dimensions

The following example shows that the bulk construction can produce bulk topological orders whose gauge group is beyond $\mathbb{Z}_2$. The GI model we start with is a one-dimensional model with two global $X$-type symmetries.

$$H_{\mathrm{GI}} = -J \sum_{i=1}^{N_x} Z_{i-1} Z_i Z_{i+1} - h \sum_{i=1}^{N_x} X_i \,. \tag{45}$$

The $\mathbb{Z}_2 \times \mathbb{Z}_2$ symmetry is generated by $\prod_i X_{3i} X_{3i+1}$ and $\prod_i X_{3i+1} X_{3i+2}$.

The standard dual model obtained from the generalized Kramers Wannier duality is the following,

$$H'_{\mathrm{GI}} = -J \sum_i Z_i + h \sum_i X_{i-1} X_i X_{i+1} \,. \tag{46}$$

The bulk model generated through our construction is the following.

$$H = -\sum_{i=1}^{N_x} \sum_{j=1}^{N_y/2} \left( A_{i,2j-1} + B_{i,2j} \right) , \tag{47}$$

$$A_{i,j} = Z_{i,j-1} Z_{i-1,j} Z_{i,j} Z_{i+1,j} Z_{i,j+1} \,,$$

$$B_{i,j} = X_{i,j-1} X_{i-1,j} X_{i,j} X_{i+1,j} X_{i,j+1} \,.$$

We prove in the appendix that this model is topologically ordered. The GSD on a torus is $2^4$. Alternatively, we may obtain the GSD from observing that the Hamiltonian local terms satisfy in total two constrains,

$$\prod_{j=1}^{N_y/2} \sum_{i=1}^{N_x/3} A_{3i,j} A_{3i+1,j} = 1 \,, \tag{48}$$

$$\prod_{j=1}^{N_y/2} \sum_{i=1}^{N_x/3} A_{3i+1,j} A_{3i+2,j} = 1 \,. \tag{49}$$

In other words, the anyon theory of this stabilizer code has total quantum dimension $D = 2^2$. In fact, the anyon theory of the bulk model is the $\mathbb{Z}_2 \times \mathbb{Z}_2$ topological order, equivalent to that of the stack of two toric code models. This is due to a proof showing that the topological phase of any 2d stabilizer code on qubits *with translation symmetry* is uniquely determined by its total quantum dimension $D = 2^n$. Its anyon theory is the same as that of $n$ copies of toric code. [85] In our case, $n = 2$.

The phase diagram of the model (45) is simple. When $J \geq h$, the ground state breaks both $\mathbb{Z}_2$ symmetries spontaneously, and when $0 < J \leq h$, the ground state is the trivial $\mathbb{Z}_2 \times \mathbb{Z}_2$ symmetric state. Comparing (45) and (46) we observe that the model (45) enjoys a self-duality at $J = h$. In fact, the ground state at $J = h$ is the critical state described by the 4-state Potts model, whose central charge equals to 1. [86–89] Due to the bulk-boundary correspondence, all these phases appear as well on the boundary of $\mathbb{Z}_2 \times \mathbb{Z}_2$ topological order.

## 6.2 Bulk topological order from a symmetry protected topological model

The one-dimensional spin systems with $\mathbb{Z}_2 \times \mathbb{Z}_2$ symmetry have more phases than those described by the model (45). Particularly, there is a SPT phase which is usually described by the following stabilizer models,

$$H = -\sum_i Z_{2i} Z_{2i+1} Z_{2i+2} - \sum_i X_{2i-1} X_{2i} X_{2i+1} . \tag{50}$$

In this convention, the symmetry is generated by

$$U_{\text{odd}}^Z = \prod_i Z_{2i+1} , \qquad U_{\text{even}}^X = \prod_i X_{2i} . \tag{51}$$

We ask if we can obtain a solvable model with a topological order adapting the bulk construction to SPT models. Indeed, we find that given a minimal variation to the bulk construction, we obtain a two-dimensional $\mathbb{Z}_2 \times \mathbb{Z}_2$ topological order from a GI model, in whose phase diagram, the SPT is one gapped phase.

To show this, we begin with the GI model. It is the Hamiltonian (50) with additional transverse field terms.

$$H_{\text{GI}} = -J \left( \sum_i Z_{2i} Z_{2i+1} Z_{2i+2} + \sum_i X_{2i-1} X_{2i} X_{2i+1} \right) - h_{\text{even}} \sum_i X_{2i} - h_{\text{odd}} \sum_i Z_{2i+1} . \tag{52}$$

The next step is to obtain the dual model, whose Hamiltonian is the following,

$$H_{\text{GI}}' = -J \sum_i (Z_{2i+1} + X_{2i}) - h_{\text{even}} \sum_i X_{2i-1} X_{2i+1} - h_{\text{odd}} \sum_i Z_{2i} Z_{2i+2} . \tag{53}$$

Obviously, the dual model is the same as two copies of Ising models in transverse fields.

**Variant construction** Now to construct the bulk model, note that (52) does not satisfy an assumption on the GI models – the local operators $\{O_i^X\}$ and $\{G_r^Z\} = \emptyset$ here do not form a CSLO. The price is that the GSD in the bulk model we would obtain from the basic construction is not robust. Part of the degeneracy originates from symmetry breaking orders.

Nevertheless, the violation is modest, and the construction, with a slight variation, can still generate a topological ordered bulk model. Let us give a minimal variation of the basic construction, essentially we modify the rule how we assign the terms across three layers to be centered on odd or even layers, based on the commutation relations of Hamiltonian local operators $O_\alpha^Z$'s and $O_i^X$'s. The variation is based on the observation that in (52) $\{X_{2i}\}$ and $\{Z_{2i+1}\}$ actually form a CSLO. The spirit is that we now assign the three-layer Hamiltonian local terms built with these terms in a CSLO to be centered in the same (odd) layers. We elaborate on the prescription in Appendix G. In particular, we give a sufficient condition when the bulk model from the variant construction has robust ground state subspace. Applying to the current example with SPT order, the modification is enough to provide us a pure topologically ordered bulk model.

The bulk Hamiltonian is

$$H = -\sum_{i,j} \left( \begin{array}{cccc} Z_{i+1,j+1,\hat{y}} & X_{i+1,j+1,\hat{x}+\hat{y}} & X_{i,j+1,\hat{x}} & Z_{i,j+1,\hat{0}} \\ \circ & \circ & \bullet & \bullet \\ Z_{i,j+1,\hat{x}} Z_{i+1,j+1,\hat{0}} Z_{i+1,j+1,\hat{x}} + & X_{i,j+1,\hat{0}} X_{i+1,j+1,\hat{x}} X_{i+2,j+1,\hat{0}} + & X_{i,j,\hat{y}} \quad X_{i+1,j,\hat{y}} + & Z_{i,j,\hat{x}+\hat{y}} \quad Z_{i+1,j,\hat{x}+\hat{y}} \\ \bullet \quad \bullet \quad \bullet & \bullet \quad \bullet \quad \bullet & \circ \quad \circ & \circ \quad \circ \\ Z_{i+1,j,\hat{y}} & X_{i+1,j,\hat{x}+\hat{y}} & X_{i,j,\hat{x}} & Z_{i,j,\hat{0}} \\ \circ & \circ & \bullet & \bullet \end{array} \right) . \tag{54}$$

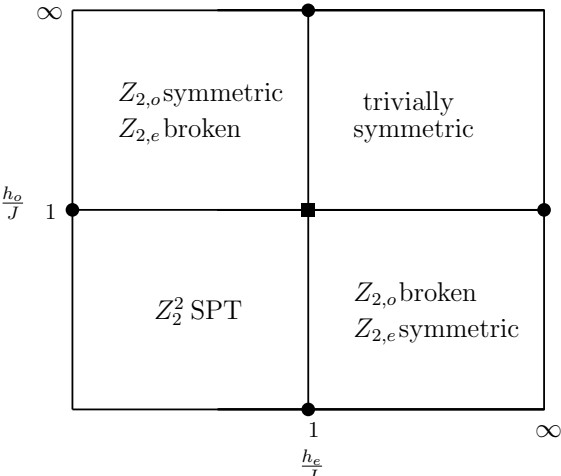

Figure 5: The phase diagram of the model (52) with $\mathbb{Z}_2 \times \mathbb{Z}_2$ symmetry. The black square locates the critical point described by free boson conformal field theory. All other transition between two gapped phase are described by the Ising conformal field theory.

This model has a topological order, and its GSD on a torus is $2^4$, when there are in total even number of layers. This can be derived through the standard polynomial representation, as we show in appendix F. As the model is a stabilizer code with translation symmetry, we can conclude that its ground state is the $\mathbb{Z}_2 \times \mathbb{Z}_2$ topologically ordered state. [85].

We show the phase diagram of the model (52) with $\mathbb{Z}_2 \times \mathbb{Z}_2$ symmetry in Fig.5. The phase diagram as well as the critical phases is most easily determined from the dual model (53).

## 6.3 Pure-loop toric code in four dimensions

There is a toric code model in four dimensions that potentially can be used as a thermally stable quantum memory [90,91]. This comes from that the model has no point-like topological excitations, but only loop-like ones. Correspondingly, the effective topological field theory for the model, given by the action $S = \frac{2}{2\pi} \int a\,db$ in 5D Euclidean spacetime, where $a, b$ are two-forms, has two 2-form $\mathbb{Z}_2$ symmetries generated by membrane operators $e^{\oint a}$ and $e^{\oint b}$. In this subsection, we show how this model comes out of our construction.

Consider the GI model on a 3d cubic lattice, described by the following Hamiltonian

$$H = -J \left( Z\,\square\,Z + \text{other orientations} \right) - h \underset{X}{\bigcirc} , \tag{55}$$

where qubits live on the links of a 3D cubic lattice with periodic boundary condition, and $p, l$ label plaquettes and links, respectively. The model is homogeneous in the three directions $x, y, z$. There is obviously no $Z$-type symmetry. To find the $X$-type symmetries, it is helpful to note that the four-$Z$ terms in $H$ also appear in the Hamiltonian of the three-dimensional toric code model. We can thus take the local $X$-type symmetry generators to be star operators of the following form,

namely products of six Pauli $X$ operators sharing a single vertex. The model has $n_X = 3$. Take three planes that cut through links and are perpendicular to the $x, y, z$ directions, respectively. The three nonlocal symmetry generators can be taken as the products of Pauli $X$ operators on the links cut by these three planes, respectively. It is illuminating to view the model as having a 1-form symmetry: given each closed membrane that intersect with links, the product of Pauli $X$ operators on the intersecting links commutes with the Hamiltonian.

We consider the standard dual theory of the above model. It is convenient to place the dual model on the same cubic lattice, but with qubits living on plaquettes. Each four-$Z$ term in $H$ is dual to a single-$Z$ term associated with the corresponding plaquette. Each single-$X$ term in $H$ is dual to the product of four Pauli $X$ operators on the plaquettes sharing the corresponding link. The dual model is actually equivalent to the original one up to the substitutions $J \leftrightarrow h$ and $Z \leftrightarrow X$.

We can now construct the bulk model according to our prescription. To this end, it is convenient to imagine expanding each link in the original model (55) along the 4th spatial direction $w$, such that the links become plaquettes parallel to the $w$ axis. In this way, the constructed bulk model can be naturally placed on a 4-dimensional cubic lattice with qubits living on plaquettes. We can write the bulk Hamiltonian as

$$H_{\text{bulk}} = -\sum_l \prod_{\partial p \ni l} X_p - \sum_c \prod_{p \in \partial c} Z_p \,, \tag{56}$$

where $l$, $p$, and $c$ label links, plaquettes (squares), and cubes, respectively. Each term in the first sum is the product of six Pauli-$X$ operators on plaquettes common to a link, and each term in the second sum is the product of six Pauli-$Z$ operators on plaquettes forming a cube.

## 6.4 Same anomalous boundary for two distinct bulks

We have seen that the two-dimensional plaquette Ising model (2) has two different dual models: one is Eq. 6, and the other is Eq. 3 with the substitutions $X \leftrightarrow Z$ and $J \leftrightarrow h$. Periodic boundary condition has been assumed for these models in our previous discussion. As a result, two distinct bulk models can be constructed: one with the Hamiltonian Eq. 9 hosts point-like quasiparticle restricted to move along inter-layer direction; and the other, whose Hamiltonian is Eq. 10 with the substitution $X \leftrightarrow Z$, has X-cube fracton order. This example suggests that two different fracton models can share the same boundary theory.

Indeed, let us construct one boundary theory that can terminate the above two distinct bulk models. The literal boundary theory with periodic boundary condition along the intra-layer directions does not work straightforwardly. This is because the boundary Hilbert space $\mathcal{L}_{\text{bdry}}$ for the two bulk models have different numbers of sectors. To get around, we take the *open boundary conditions* along intra-layer directions. In result, $\mathcal{L}_{\text{bdry}}$ has a single sector for both models.

Now we describe the construction. We place the two-dimensional plaquette Ising model on a 2D open square lattice with $L_x \times L_y$ sites, with the same Hamiltonian (2) (only include terms that are completely inside the system). We take the first dual model to be its standard dual, defined on an open square lattice with $(L_x - 1) \times (L_y - 1)$ sites. One can verify that this dual model has no $\mathbb{Z}_2$ symmetry at all. We define the second dual model on an open square lattice of the same size ($L_x \times L_y$ number of vertices), but now with qubits living on the links. As in the periodic case, we take the dual of each GI term $\mathcal{O}_\alpha^Z$ of the plaquette Ising model to be the product of four Pauli-$Z$ operators around a plaquette. The $X$-type gauge symmetries of the dual model are generated by the product of all Pauli-$X$ operators around each vertex. The dual of each transverse-field term $X_i$ of the plaquette Ising model is uniquely determined modulo the gauge symmetry terms. One can verify that this second dual model has no $Z$-type symmetry at all. We can then construct two bulk models according to the two dual theories,

and take open boundary condition along the out-of-layer direction as well. The boundary Hilbert space $\mathcal{L}_{\text{bdry}}$ for both models, following the treatment in the Section 5, only has one sector now.

We note that although open boundary condition is taken along the intra-layer directions, our construction actually forbids boundary terms on the four surface planes perpendicular to the layers, according to our result in Appendix E.5. The $X$-suspension, $Z$-suspension, and gauge symmetry operators ("bulk" operators) near these surface areas have coefficients as large as those deep inside the bulk, and therefore fix all local degrees of freedom there. With this somewhat artificial setup, the boundary theory only contains local degrees of freedom near the 1st and the $L$-th layers.

Consequently, we can have exactly the same anomalous boundary theory for these two models: two copies of the two-dimensional plaquette Ising model (possibly with different $J$ and $h$ coefficients), subject to the symmetry projections $U_k^X \otimes U_k^X = 1$ for $k = 1, 2, \cdots, n_X$ where $n_X = L_x + L_y - 1$.

# 7 Conclusions and Discussions

In this paper, we systematically construct $d + 1$-dimensional commuting projector models with long-range orders from $d$-dimensional GI models – qubit lattice models with non-commuting Hamiltonian local terms. The simplicity of the construction allows us to analyze the precise correspondence between the boundary of the long-range ordered state and the GI model. Under a certain condition, the boundary model is subject to global constraints, which are either symmetry charge projections, and/or boundary conditions, implying that the boundary theory to be anomalous. Furthermore, the anomalous boundary model is isomorphic to two copies of the GI models from which the bulk model is constructed, also subject to global constraints. The condition for the anomaly is a surprising requirement on the non-local symmetries in either the GI model or its dual model. This is in contrary to the most common intuition that for *all* non-local symmetries appearing on the boundary of a long-range ordered state, up to those in the system trivially stacked onto the boundary, only the charge-neutral states under the symmetry contribute to the boundary Hilbert space.

Many open questions following this work worth future exploration. For example, we have not discussed the consequences of bulk anyon fluxes terminating on the boundary theory. Such results will be part of the properties of non-invertible anomaly in higher dimensions. Our construction also has the potential leading to many tangible and interesting generalizations. In particular, qubit stabilizer models only describe a limited class of topological phases [92]. An immediate generalization to $\mathbb{Z}_n$ qudit lattice models is worthwhile. With it, we postulate that the bulk models with topological/fracton orders can be constructed staring with a qudit lattice models with *any* discrete finite Abelian symmetries $G$ in transverse fields, since $G$ can always be written as $G = \prod_{i=1}^m \mathbb{Z}_{n_i}$, with a positive integer $m$, and integers $n_i \geq 2$. One question with further stretch is whether the generalization of our models to topological models beyond stabilizer models can generate new topological lattice models, especially in three dimensions or higher. More particularly, the lattice models of topological phases related by Morita equivalence are related by generalized Kramers-Wannier duality. [93–95] It is interesting to adapt our alternating layer construction to Morita equivalent topological lattice models and study the topological properties of the resulting bulk model. The Haah's code model [96] does not seem to fit into our construction. Nevertheless, the stabilizers in its Hamiltonian do have a clear trilayer structure, viewed from the $(1, 1, 1)$-direction. It is interesting to figure out whether some generalized construction works for this representative type-II fracton model. The low energy effective field description for our alternating layer in generating topological theories

in one dimensional higher is also in demand. Our construction might remind the readers of the coupled-layer construction in generating topologically ordered models, fracton models, or their hybrids [83, 97–104], in which the model in each layer to begin with is the same, and inter-layer coupling terms are introduced. We leave it for future works to unravel whether there is a relation between our approach and the conventional coupled-layer constructions. Last but not least, our constructed bulk model is straightforward so that their boundary phase diagrams with various Abelian global symmetries can in principle be accessed via numerics. This suggests the possibility of the numerical verification for the conjecture that gapped phases on the boundary of $(d+1)$-dimensional topological order with a discrete gauge group $G$ have one-to-one correspondence with the gapped phases on the $d$-dimensional system with the global $G$ symmetry.

# Acknowledgments

We would like to thank Nayan Myerson-Jain, Cenke Xu, and Sagar Vijay for a previous collaboration and related discussions that inspired this work. We are also grateful to Xiao-Gang Wen, Ashvin Vishwanath, Cenke Xu, Sagar Vijay, Nat Tantivasadakarn, Kaixiang Su, Zhu-Xi Luo, Yi-Zhuang You and Zhen Bi for illuminating discussions.

*Note added.* We came to know of Ref. [105] which also considers correspondence between bulk and boundary of fracton orders. In particular, it also points out that the same boundary theory may be realized for two distinct bulk fracton orders.

**Funding information** S. L. is supported by the Gordon and Betty Moore Foundation under Grant No. GBMF8690 and the National Science Foundation under Grant No. NSF PHY-1748958. W. J. is supported by the Simons Foundation. Also, W. J. gratefully acknowledges support from the Simons Center for Geometry and Physics, Stony Brook University at which some of the research for this paper was performed.

# A    Elementary results in the stabilizer formalism

In this section, we introduce some elementary results in the stabilizer formalism, and set up a few terminologies that will be used in the rest of the appendix. Basics of the stabilizer formalism can also be found in [106, 107].

A set of **stabilizers** is a set of operators satisfying the following properties:

- They are tensor products of $I, X, Y, Z$ multiplied by $\pm$ sign factors.

- They mutually commute.

- The group generated by them does not contain $-1$.

The last property guarantees that all the stabilizers are able to simultaneously take the eigenvalue $+1$. A set of stabilizers is called **independent** if any product of a nonempty subset of those operators is not proportional to the identity. Specifying the eigenvalue of each independent stabilizer will reduce the Hilbert space dimension by a half. To see this, suppose we have chosen the eigenvalues $\lambda_1, \lambda_2, \cdots, \lambda_k$ for $k$ independent stabilizers $U_1, U_2, \cdots, U_k$, respectively, with $\lambda_i = \pm 1$ for all $i$. One can see that the $(k+1)$-th independent stabilizer $U_{k+1}$ has zero

trace *in this eigen-subspace*, i.e.

$$\text{Tr}\left[U_{k+1}\prod_{i=1}^{k}\left(\frac{1+\lambda_i U_i}{2}\right)\right] = 0,\tag{A.1}$$

due to the zero trace of Pauli operators and the requirement of independence. Thus, $U_{k+1}$ has equal number of $+1$ and $-1$ eigenvalues in this eigen-subspace, and specifying its eigenvalue will further cut down the Hilbert space dimension by a half. As a consequence, it is impossible to find more than $N$ number of independent stabilizers where $N$ is the total number of qubits. A set of stabilizers is called **complete** if it includes $N$ number of independent stabilizers. Each possible list of eigenvalues of a complete set of stabilizers uniquely determines a state in the Hilbert space.

The following two lemmas about stabilizer extension are useful.

**Lemma 1.** *Any set of independent stabilizers can be extended to a complete and independent one.*

*Proof.* Suppose we have $N-k$ number of independent stabilizers with $k > 0$. Any set of eigenvalues for those operators determines $2^k$ number of degenerate eigenstates. It is therefore always possible to find some operator $\mathcal{A}$, not necessarily a tensor product of $I, X, Y, Z$, that is independent from and commutes with the existing stabilizers. We can expand $\mathcal{A}$ as a linear combination of the $4^N$ number of tensor product operators of $I, X, Y, Z$ which are linearly independent. Let $U$ be any existing stabilizer, $U\mathcal{A}U^{-1} = \mathcal{A}$ implies that any tensor product operator entering the expansion of $\mathcal{A}$ with a nonzero coefficient must also commute with $U$. Therefore, we can always find a tensor product of $I, X, Y, Z$ that is independent from and commutes with the existing stabilizers. We can add this new operator to the list of stabilizers and start over again, until the list is complete. $\square$

**Corollary 4.** *A set of independent stabilizers is complete if and only if it is maximal, i.e. not belonging to a larger set of independent stabilizers.*

**Lemma 2.** *Suppose we are given a set of independent stabilizers that are all products of Pauli X operators (and the identity operators on the other sites; same below). We can extend the set of stabilizers to a complete one by adding operators that are products of Pauli Z operators.*

*Proof.* Let $N$ be the number of qubits in the Hilbert space and let $M$ be the number of independent stabilizers given. We can represent those stabilizers by an $N \times M$ matrix in $\mathbb{F}_2$, the field of integers modulo 2, such that the $(i, j)$ entry of the matrix is 1 when the $j$-th stabilizer contains an $X$ operator acting on the $i$-th qubit, and is 0 otherwise. Each column of the matrix corresponds to a stabilizer, and each row corresponds to a qubit in the Hilbert space. Multiplying one stabilizer onto another one will lead to an equivalent set of independent stabilizers, which amounts to adding one column of the matrix onto another one. We are also free to permute the rows of the matrix, which amounts to reordering the qubits. Using these two types of operations, this stabilizer matrix can be cast to the following canonical form

$$\begin{pmatrix} I_M \\ A \end{pmatrix},\tag{A.2}$$

where $I_M$ is the $M \times M$ identity matrix, and $A$ is an arbitrary $(N-M) \times M$ matrix. To prove the claim, we will now introduce $N-M$ number of new stabilizers, each of which is a product of Pauli $Z$ operators. Those new stabilizers can be similarly represented by an $N \times (N-M)$ matrix in $\mathbb{F}_2$; the $(i, j)$ entry of the matrix is 1 when the $j$-th new stabilizer contains a $Z$ operator acting on the $i$-th qubit, and is 0 otherwise. We choose this new stabilizer matrix to be

$$\begin{pmatrix} A^T \\ I_{N-M} \end{pmatrix}.\tag{A.3}$$

One can check that the $N$ operators represented by those two matrices indeed mutually commute and are independent. This completes the proof. $\qquad\square$

## B  Hilbert space isomorphism in the generalized Kramers-Wannier duality

In the main text, we introduced the generalized Kramers-Wannier duality by the operator map in Eq. 4. In this section, we will prove that this operator map follows from an isomorphism of symmetric subspaces.

The following result that follows from our definition of the symmetry groups in the main text and Lemma 2 will be useful.

**Corollary 5.** *Let $\mathcal{L}$ ($\mathcal{L}'$) be the full Hilbert space of the original (dual) model. The following statements hold.*

1. *$\{\mathcal{O}_i^X\}$ ($\{\Delta_\alpha^Z\}$) together with all the Z-type (X-type) symmetry generators of the original (dual) theory form a complete set of stabilizers in $\mathcal{L}$ ($\mathcal{L}'$).*

2. *$\{\mathcal{O}_\alpha^Z\}$ ($\{\Delta_i^X\}$) together with all the symmetry generators of the original (dual) theory form a complete set of stabilizers in $\mathcal{L}$ ($\mathcal{L}'$).*

It is also useful to establish the following result.

**Lemma 3.** *Let $\{U_1^Z, U_2^Z, \cdots, U_m^Z, U_1^X, U_2^X, \cdots, U_n^X\}$ be a set of stabilizers acting on some multiple-qubit Hilbert space $\mathcal{L}$, such that each $U_p^Z$ ($U_q^X$) is a product of several Pauli Z (Pauli X) operators acting on different qubits. Let $\mathcal{L}_0$ be the subspace where $U_p^Z = U_q^X = 1$ for all $p$ and $q$. There exists an orthonormal basis of $\mathcal{L}_0$ such that*

1. *Any operator $\mathcal{O}^Z$ that is a product of several Pauli Z operators and commutes with all the stabilizers is diagonal in this basis.*

2. *Any operator $\mathcal{O}^X$ that is a product of several Pauli X operators and commutes with all the stabilizers has matrix elements in this basis equal to either 0 or 1.*

*Proof.* Let $|\{Z_i\}\rangle$ be the standard eigenbasis for all Pauli $Z$ operators in $\mathcal{L}$, i.e. tensor products of the states $(1,0)^T$ and $(0,1)^T$. Consider the following list of states,

$$\overline{|\{Z_i\}\rangle} := \prod_q \left(\frac{1+U_q^X}{2}\right) \prod_p \left(\frac{1+U_p^Z}{2}\right) |\{Z_i\}\rangle . \tag{B.1}$$

Notice that these states generate all states in $\mathcal{L}_0$. Moreover, the above states have the following simple properties: (1) $\overline{|\{Z_i\}\rangle} = 0$ if and only if the spin configuration $\{Z_i\}$ violates the $U^Z$ constraints. (2) Given any two nonzero states $\overline{|\{Z_i\}\rangle}$ and $\overline{|\{Z_i'\}\rangle}$, they are equal if $|\{Z_i\}\rangle$ can be mapped to $|\{Z_i'\}\rangle$ by the action of several $U^X$ operators, otherwise they are orthogonal. We can restrict the above list of states to a nonzero and nonequal subset. After normalization, this subset of states form an orthogonal basis of $\mathcal{L}_0$, and it has the desired properties claimed in the statement of the lemma. $\qquad\square$

The key result for this section is the following.

**Theorem 2.** *The operator map in the generalized Kramers-Wannier duality follows from an isomorphism of Hilbert spaces when we restrict to the symmetric sectors on both sides.*

*Proof.* We start by defining some notations. Let $\mathcal{L}$ and $\mathcal{L}'$ be the full Hilbert spaces of the original theory and the dual theory, respectively. We denote by $\mathcal{L}_Z \subset \mathcal{L}$ the symmetric subspace for all $Z$-type symmetries, and by $\mathcal{L}_0 \subset \mathcal{L}$ the symmetric subspace for both $Z$-type and $X$-type symmetries. Thus $\mathcal{L}_0 \subset \mathcal{L}_Z \subset \mathcal{L}$. Similarly, we define $\mathcal{L}'_X$ and $\mathcal{L}'_0$ such that $\mathcal{L}'_0 \subset \mathcal{L}'_X \subset \mathcal{L}'$.

Each possible list of eigenvalues of a complete set of stabilizers uniquely determines a state in the Hilbert space (see Appendix A). Therefore, according to Corollary 5, the eigenvalues of all $\mathcal{O}^Z_\alpha$ uniquely determine a state in $\mathcal{L}_0$, and the eigenvalues of all $\Delta^Z_\alpha$ uniquely determine a state in $\mathcal{L}'_X$. Denote by $|\{\mathcal{O}^Z_\alpha\}\rangle \in \mathcal{L}_0$ the simultaneous eigenstates of all $\mathcal{O}^Z_\alpha$ (here we use the same notation for an operator and its eigenvalue). By Lemma 3 just proved above, we can always choose the phase factors of those states such that the matrix elements of each $\mathcal{O}^X_i$ in this basis are either 0 or 1. Similarly, we denote by $|\{\Delta^Z_\alpha\}\rangle \in \mathcal{L}'_X$ the simultaneous eigenstates of all $\Delta^Z_\alpha$, such that the matrix elements of $\Delta^X_i$ are either 0 or 1.

We define a homomorphism $f : \mathcal{L}_0 \to \mathcal{L}'_X$ such that $f$ maps $|\{\mathcal{O}^Z_\alpha\}\rangle$ to the state $|\{\Delta^Z_\alpha\}\rangle$ with the eigenvalues satisfying $\Delta^Z_\alpha = \mathcal{O}^Z_\alpha$. As long as $f$ is well-defined which is to be proved below, the operator $\mathcal{O}^X_i$ is mapped to $\Delta^X_i$ under this map, i.e. $f\mathcal{O}^X_i = \Delta^X_i f$. This is because the commuting or anticommuting relations between $\{\mathcal{O}^Z_\alpha\}$ and $\{\mathcal{O}^X_i\}$ are the same as those between $\{\Delta^Z_\alpha\}$ and $\{\Delta^X_i\}$. We will prove that $f$ is actually an isomorphism from $\mathcal{L}_0$ to the subspace $\mathcal{L}'_0 \subset \mathcal{L}'_X$.

$$
\begin{array}{ccc}
\mathcal{L}_0 \lhook\joinrel\longrightarrow \mathcal{L}_Z \lhook\joinrel\longrightarrow \mathcal{L} \\[4pt]
\cong \Big\updownarrow \quad \searrow{\scriptstyle f} \\[4pt]
\mathcal{L}'_0 \lhook\joinrel\longrightarrow \mathcal{L}'_X \lhook\joinrel\longrightarrow \mathcal{L}'
\end{array}
$$

Since the $\Delta^Z_\alpha$ operators are not necessarily independent, we need to first confirm that $f$ is well-defined, i.e. the desired image state always exists in $\mathcal{L}'_X$. Although $\Delta^Z_\alpha$ may not be independent, they are generated from an independent subset of operators which are able to independently take eigenvalues $\pm 1$. Therefore, to prove that $f$ is well-defined, it suffices to show that whenever $\prod_{\alpha \in S} \Delta^Z_\alpha = 1$ in $\mathcal{L}'$ for some set $S$, $\prod_{\alpha \in S} \mathcal{O}^Z_\alpha = 1$ is also true in $\mathcal{L}_0$. This is not hard; since $\prod_{\alpha \in S} \Delta^Z_\alpha$ commutes with all $\Delta^X_i$, $\prod_{\alpha \in S} \mathcal{O}^Z_\alpha$ must commute with all $\mathcal{O}^X_i$ by the duality, and is therefore either the identity or a $Z$-type symmetry of the original theory which is set to 1 in $\mathcal{L}_0$.

Next, we would like to show $\mathrm{Im} f \subset \mathcal{L}'_0$. Corollary 5 says that $\{\Delta^Z_\alpha\}$ and all the $X$-type symmetry generators of the dual theory form a complete set of stabilizers in $\mathcal{L}'$. It follows that each $Z$-type symmetry generator of the dual theory is a product of some $\Delta^Z_\alpha$ operators, which is then dual to either the identity or a $Z$-type symmetry generator of the original theory. Given that all symmetry generators are set to 1 in $\mathcal{L}_0$, the image of $f$ must be contained in $\mathcal{L}'_0$.

From the definition, we see that $f$ maps a basis of $\mathcal{L}_0$ to a set of linearly independent states in $\mathcal{L}'_0$. It follows that $f$ is injective and $\dim \mathcal{L}_0 = \dim \mathrm{Im} f \leq \dim \mathcal{L}'_0$. Now, we can construct another map $g : \mathcal{L}'_0 \to \mathcal{L}_Z$ similar to $f$ (the roles of $X$ and $Z$ need to be exchanged), and eventually show that $\dim \mathcal{L}'_0 \leq \dim \mathcal{L}_0$. Therefore, we must have $\dim \mathcal{L}'_0 = \dim \mathcal{L}_0 = \dim \mathrm{Im} f$, which implies that the restricted map $f : \mathcal{L}_0 \to \mathcal{L}'_0$ is one-to-one. This is the isomorphism claimed by the theorem. $\qquad \square$

## C  A toy example for sanity checks

In this subsection, we introduce a toy example that will be used later for testing our results. Consider the model

$$
H = -J \overset{Z}{-\!\!\circ\!\!-} - h \ X\!\!\overset{X}{\underset{X}{\circ\!\!\overset{\circ}{\square}\!\!\circ}}\!\!X \ , \tag{C.1}
$$

and its dual model

$$H' = -J \ Z\!\!\begin{array}{c} Z \\ \bullet\!\!-\!\!\bullet \\ \bullet\!\!-\!\!\bullet \\ Z \end{array}\!\!Z \ -h-\!\!\bullet\!\!- \overset{X}{} \,, \tag{C.2}$$

both defined on a 2D square lattice of the size $L_x \times L_y$, with periodic boundary condition, and with qubits living on the links. One may shift the dual lattice relative to the original one along $y$ by half the lattice constant, such that the centers of each dual pair of operators coincide. We emphasize that there is no vertical-link single $Z$ (single $X$) term in $H$ ($H'$). The $Z$-type symmetries of $H$ are the same as those for Eq. 3. Each $X$-type symmetry generator of $H$ is a product of all vertical-link $X$ operators along even number of vertical lines, which is nonlocal. The dual model $H'$ is equivalent to $H$ under the substitutions $J \leftrightarrow h$ and $Z \leftrightarrow X$. One may check that $n_Z = m_X = 2$ and $n_X = m_Z = L_x - 1$. A three-dimensional bulk model can thus be constructed according to our prescription.

## D  Ground state degeneracy of the bulk model

In this section, we compute the GSD of the bulk model with periodic boundary condition along the out-of-layer direction. We denote by $L \in 2\mathbb{Z}$ the total number of layers.

We need to introduce some new parameters that enter our final result. Let $\mathcal{G}_Z = \left\langle U_1^Z, \cdots, U_{n_Z}^Z \right\rangle$ be the group generated by the $U^Z$ operators of the original theory. We define two subgroups of it:

- $\mathcal{G}_{Z,1} = \{g \in \mathcal{G}_Z \,|\, g = \prod_{\alpha \in A} \mathcal{O}_\alpha^Z \prod_{r \in R} G_r^Z$ for some sets $A$ and $R\}$.

- $\mathcal{G}_{Z,0} = \{g \in \mathcal{G}_Z \,|\, g = \prod_{\alpha \in A} \mathcal{O}_\alpha^Z \prod_{r \in R} G_r^Z$ for some sets $A$ and $R$ satisfying $\prod_{\alpha \in A} \Delta_\alpha^Z = 1\}$.

Obviously, $\mathcal{G}_{Z,0} \subset \mathcal{G}_{Z,1}$. We can always redefine the $U^Z$ operators such that, for some integers $\nu$ and $\bar{\nu}$, with $\bar{\nu} \geq \nu$ $\mathcal{G}_{Z,0} = \left\langle U_1^Z, \cdots, U_\nu^Z \right\rangle$ and $\mathcal{G}_{Z,1} = \left\langle U_1^Z, \cdots U_{\bar{\nu}}^Z \right\rangle$. Similarly, let $\mathcal{G}_X' = \left\langle \Omega_1^X, \cdots, \Omega_{m_X}^X \right\rangle$ be the group generated by the $\Omega^X$ operators of the dual theory. We define the following two subgroups:

- $\mathcal{G}_{X,1}' = \{g \in \mathcal{G}_X' \,|\, g = \prod_{i \in I} \Delta_i^X \prod_{\rho \in R} \Gamma_\rho^X$ for some sets $I$ and $R\}$.

- $\mathcal{G}_{X,0}' = \{g \in \mathcal{G}_X' \,|\, g = \prod_{i \in I} \Delta_i^X \prod_{\rho \in R} \Gamma_\rho^X$ for some sets $I$ and $R$ satisfying $\prod_{i \in I} \mathcal{O}_i^X = 1\}$.

We have $\mathcal{G}_{X,0}' \subset \mathcal{G}_{X,1}'$. We can always redefine the $\Omega^X$ operators such that, for some integers $\mu$ and $\bar{\mu}$, with $\bar{\mu} \geq \mu$, $\mathcal{G}_{X,0}' = \left\langle \Omega_1^X, \cdots, \Omega_\mu^X \right\rangle$ and $\mathcal{G}_{X,1}' = \left\langle \Omega_1^X, \cdots, \Omega_{\bar{\mu}}^X \right\rangle$. We show two examples below in which $n_Z > \bar{\nu}$.

Our main result for this section is the following.

**Theorem 3.** *Let $D$ be the GSD of the bulk model, then*

$$\log_2 D = n_X + m_Z - [(\bar{\nu} - \nu) + (\bar{\mu} - \mu)] + [(n_Z - \nu) + (m_X - \mu)](L/2). \tag{D.1}$$

**Examples.** As the first example, consider the X-cube order described by the Hamiltonian in Eq. 10 that is constructed from Eq. 3 and Eq. 6. We put the model on a 3D cubic lattice with $L_x \times L_y \times L_z$ number of vertices, with periodic boundary condition along all three directions. The number of layers $L$ is given by $L = 2L_z$. The model has $n_Z = 2$, $n_X = L_x + L_y - 2$, $m_X = 0$, $m_Z = L_x + L_y - 1$, and $\bar{\nu} = \nu = \bar{\mu} = \mu = 0$. It follows that $\log_2 D = 2(L_x + L_y + L_z) - 3$, which is a well-known result [59].

As the second example, consider the model in Appendix C. We take periodic boundary condition along the out-of-layer direction, and the number of layers $L$ has to be an even integer. The model has $n_Z = m_X = 2$, $n_X = m_Z = L_x - 1$, $\bar{\nu} = \bar{\mu} = 1$, and

$$\nu = \mu = \begin{cases} 1 & (L_x \text{ odd}), \\ 0 & (L_x \text{ even}). \end{cases} \tag{D.2}$$

It follows that

$$\log_2 D = \begin{cases} 2L_x + L - 2 & (L_x \text{ odd}), \\ 2L_x + 2L - 4 & (L_x \text{ even}), \end{cases} \tag{D.3}$$

which we have verified numerically.

The rest of this section is to prove the theorem. Our strategy for counting the GSD is as follows. We extend the set of stabilizers in $H_{\text{bulk}}$ to a complete one by including $r$ number of additional stabilizers $C_1, C_2, \cdots, C_r$ which are independent modulo the stabilizers in $H_{\text{bulk}}$, i.e. each $C_k$ can not be generated by the other $C_{k'}$ operators and the stabilizers in $H_{\text{bulk}}$. Then the GSD is nothing but $2^r$. The first useful result is the following.

**Lemma 4.** *The stabilizers in $H_{bulk}$ together with all the nonlocal symmetry operators, namely $U_{k,2l-1}^Z$, $U_{k,2l-1}^X$, $\Omega_{k,2l}^X$, and $\Omega_{k,2l}^Z$, form a complete set of stabilizers.*

*Proof.* As an equivalent statement, any operator $\mathcal{A}$ that is a product of Pauli operators and commutes with all these stabilizers can be generated by them (see Corollary 4). As in Theorem 1, we write $\mathcal{A} = \mathcal{A}_Z \mathcal{A}_X$. If suffices to show that both $\mathcal{A}_Z$ and $\mathcal{A}_X$ are generated by the stabilizers in the theorem statement.

We will focus on $\mathcal{A}_X$, and $\mathcal{A}_Z$ is similar. Since $\mathcal{A}_X$ commutes with all the $G^Z$ and $U^Z$ operators, each odd layer of $\mathcal{A}_X$ is a product of several $\mathcal{O}^X$ operators. Therefore, using the $X$-suspension operators, we are able to remove the $X$ operators in $\mathcal{A}_X$ on all but one odd layers. Say this remaining odd layer is just the 1st layer. We may thus assume $\mathcal{A}_X$ only contains $X$ operators on the 1st layer or with even layer indices. To commute with the $Z$-suspension terms, different even layers of $\mathcal{A}_X$ must differ from each other only by some $\Gamma^X$ and $\Omega^X$ operators. Thus, using the $\Gamma^X$ and $\Omega^X$ operators, $\mathcal{A}_X$ can be cast to the form $\mathcal{A}_X^1 \prod_{p \in \Lambda} \prod_{l=1}^{L/2} X_{p,2l}$ for some set $\Lambda$, where $\mathcal{A}_X^1$ is an operator that only acts on the 1st layer. This $\mathcal{A}_X$ has to commute with the $\Omega_{k,2l}^Z$ and $\Gamma_{\sigma,2l}^Z$ operators, which are the $Z$-type symmetry generators for the dual theory. It follows from our Lemma 2 that $\prod_{p \in \Lambda} X_p$ must be a product of $\Delta_i^X$, $\Gamma_\rho^X$ and $\Omega_k^X$. Therefore, using the $X$-suspension operators, the $\Gamma^X$ and the $\Omega^X$ operators, we are able to reduce $\mathcal{A}_X$ to an operator that acts on the 1st layer. In order to commute with the $G^Z$, $U^Z$ and $Z$-suspension operators, this single-layer operator is an $X$-type symmetry generator of the original theory, and thus a product of the $G^X$ and $U^X$ operators. This completes the proof. $\square$

Next, we shall restrict the nonlocal symmetry operators to a subset such that, modulo the stabilizers in $H_{\text{bulk}}$, the subset is independent and equivalent to the original set. First notice that, starting from $U_{k,2l_0-1}^X$ for some $l_0$, using the $X$-suspension, $\Gamma^X$ and $\Omega^X$ operators, we are able to generate $U_{k,2l-1}^X$ for all $l$. This is because each $U_k^X$ operator acting on the original lattice can be written as a product of several $\mathcal{O}^X$ operators, and is then dual to an $X$-type symmetry generator of the dual theory. It thus suffices to retain $U_{k,2l_0-1}^X$ while dropping the $U^X$ operators acting on all the other layers. Similarly, it suffices to retain $\Omega_{k,2l_0}^Z$ while dropping the $\Omega^Z$ operators acting on all the other layers. Secondly, according to the definition of $\mathcal{G}_{Z,0}$, each $U_{k,2l-1}^Z$ with $k \leq \nu$ is a product of some $Z$-suspension and $G^Z$ operators, and thus can be dropped. Similar for the $\Omega_{k,2l}^X$ operators with $k \leq \mu$. Thirdly, according to the definition of $\mathcal{G}_{Z,1}$, $\prod_{l=1}^{L/2} U_{k,2l-1}^Z$ for $\nu < k \leq \bar{\nu}$ can be generated by the $Z$-suspension and $G^Z$ operators,

thus we can drop the $U_{k,1}^Z$ operators acting on the 1st layer with $\nu < k \leq \bar{\nu}$. Similarly, we can drop the $\Omega_{k,2}^X$ operators acting on the 2nd layer with $\mu < k \leq \bar{\mu}$. After all these steps, we are left with the following reduced set of nonlocal symmetry operators:

- $U_{k,2l_0-1}^X$ for all $k$ and *some* $l_0$; $\Omega_{k,2l_0}^Z$ for all $k$ and *some* $l_0$.

- $U_{k,2l-1}^Z$ for $\nu < k \leq \bar{\nu}$ and $l \neq 1$; $\Omega_{k,2l}^X$ for $\mu < k \leq \bar{\mu}$ and $l \neq 1$.

- $U_{k,2l-1}^Z$ for $\bar{\nu} < k \leq n_Z$ and all $l$; $\Omega_{k,2l}^X$ for $\bar{\mu} < k \leq m_X$ and all $l$.

We complete the proof for Theorem 3 by the following claim.

**Lemma 5.** *The above stabilizers are independent modulo the stabilizers in $H_{\mathrm{bulk}}$. Equivalently, they can independently take the eigenvalues $\pm 1$ in the ground state subspace.*

*Proof.* It suffices to find operators that can independently flip the signs of the above stabilizers without affecting the stabilizers in $H_{\mathrm{bulk}}$. In the main text, we have discussed how to flip the signs of $U_{k,2l_0-1}^X$ or $\Omega_{k,2l_0}^Z$, and let us not repeat it here. According to the definition of $\mathcal{G}_{Z,1}$ and $\bar{\nu}$, each $U_{k>\bar{\nu}}^Z$ is independent from (not a product of) the $\mathcal{O}^Z$ and $G^Z$ operators. As a result, there exist some $X$-type operators $V_{k>\bar{\nu}}^X$ acting on the Hilbert space of the original model such that each $V_{k>\bar{\nu}}^X$ anticommutes with $U_{k>\bar{\nu}}^Z$ but commutes with all the other $U_{k'>\bar{\nu}}^Z$, $\mathcal{O}^Z$ and $G^Z$ operators. We can then use the operator $V_{k>\bar{\nu},2l-1}^X$ to flip the sign of $U_{k>\bar{\nu},2l-1}^Z$, without affecting the other stabilizers listed above and the stabilizers in $H_{\mathrm{bulk}}$. The scenario for the $\Omega_{k>\bar{\mu},2l}^X$ operators is similar. Flipping the sign of a $U_{k,2l-1}^Z$ operator with $\nu < k \leq \bar{\nu}$ and $l \neq 1$ is somewhat complicated, consisting of several steps: (1) Let $V_{k\leq\bar{\nu}}^X$ be $X$-type operators acting on the Hilbert space of the original theory such that $V_k^X$ for each $k \leq \bar{\nu}$ anticommutes with $U_k^Z$ but commutes with all the other local or nonlocal $Z$-type symmetry generators of the original theory ($G_r^Z$ and $U_{k'}^Z$ for all the other $k'$). Apply $V_{k,2l-1}^X$ with some $k \in \{\nu+1, \nu+2, \cdots, \bar{\nu}\}$ to flip the sign of $U_{k,2l-1}^Z$. This will necessarily flip the signs of some $Z$-suspension operators centering on the $(2l-1)$-th layer as well, which we need to fix. (2) We can restrict the $\Delta^Z$ operators of the dual theory to an independent and equivalent subset: $\Delta_{\alpha_1}^Z, \Delta_{\alpha_2}^Z, \cdots$. It suffices to fix the $Z$-suspension operators centering on the $(2l-1)$-th layer with these $\alpha$-indices $(\alpha_1, \alpha_2, \cdots)$ because they can generate all the other $Z$-suspension operators centering on the $(2l-1)$-th layer with the help of the $G_{r,2l-1}^Z$ and $U_{k\leq\nu,2l-1}^Z$ operators. There exists a set of operators $E_{\alpha_1}^X, E_{\alpha_2}^X, \cdots$ such that $E_{\alpha_n}^X$ anticommutes with $\Delta_{\alpha_n}^Z$ while commutes with all the other independent $\Delta^Z$ operators. We can use $\prod_{l'=1}^{l-1} E_{\alpha_n,2l}^X$ ($\prod_{l'=l}^{L/2} E_{\alpha_n,2l}^X$) when $l_0 \geq l$ ($l_0 < l$) to fix the eigenvalues of the $Z$-suspension operators centered on the $(2l-1)$-th layer without affecting $\Omega_{k,2l_0}^Z$. This step will flip the signs of some $Z$-suspension operators centering on the 1st layer. (3) Apply $V_{k,1}^X$ with the same $k$ as in the first step. Notice that the $\Gamma^Z$ operators will not be affected by the above steps, because $\Gamma_{\sigma,2l_0}^Z$ are unaffected, and the other $\Gamma^Z$ operators can all be generated from the $\Gamma_{\sigma,2l_0}^Z$, $Z$-suspension, and $G^Z$ operators. $\qquad\square$

# E   Further details on the bulk-boundary correspondence

This section is devoted to understanding the boundary theory of our bulk model: the boundary Hilbert space $\mathcal{L}_{\mathrm{bdry}}$ and the boundary Hamiltonian $H_{\mathrm{bdry}}$ which are defined in Section 5. The bulk model is defined in $d+1$ spatial dimensions, and we expect the boundary theory to have an effective $d$-dimensional description that will enable us to examine whether it is anomalous or not.

In Appendix E.1, we derive a complete and independent set of stabilizers characterizing the full Hilbert space of our system, which turns out to be useful for understanding $\mathcal{L}_{\text{bdry}}$. We work out a $d$-dimensional effective description of the boundary theory as well as a condition for anomaly in Appendix E.2 for a special case and in Appendix E.3 for the general situation. The analysis uses a classification result about possible local terms in $H_{\text{bdry}}$, which is proved in Appendix E.5. In Appendix E.4, we discuss in what cases the anomaly condition is violated.

## E.1   Stabilizers characterizing the full Hilbert space

Our strategy for analyzing $\mathcal{L}_{\text{bdry}}$ is as follows. We first find a complete and independent set of stabilizers that characterizes the full Hilbert space, done in this subsection. Then we divide this set into two subsets, such that one subset of stabilizers is equivalent to the stabilizers appearing in the bulk Hamiltonian. It follows that the boundary Hilbert space is characterized by the other subset of stabilizers. Once this is done, we will find a natural $d$-dimensional description of $\mathcal{L}_{\text{bdry}}$.

The first useful result is the following.

**Lemma 6.** *The following operators form a complete (but not independent) set of stabilizers.*

1. *All the $\mathcal{O}^X$ operators on the two boundary layers.*

2. *All the $X$-suspension and $Z$-suspension terms in the bulk Hamiltonian.*

3. *All $G^Z_{r,2l-1}$ and $U^Z_{k,2l-1}$, namely the $Z$-type symmetry generators of the original theory acting on all odd layers.*

4. *All $\Gamma^X_{\rho,2l}$ and $\Omega^X_{k,2l}$, namely the $X$-type symmetry generators of the dual theory acting on all even layers.*

5. *$\Gamma^Z_{\sigma,2l_0}$ and $\Omega^Z_{k,2l_0}$ for all $\sigma$, $k$ and some fixed $l_0$.*

*Proof.* Our strategy is similar to that for Theorem 1 in the main text. Suppose $\mathcal{A} = \mathcal{A}_Z \mathcal{A}_X$ commutes with all the above operators, where $\mathcal{A}_Z$ ($\mathcal{A}_X$) is a product of Pauli $Z$ (Pauli $X$) operators. We will show that $\mathcal{A}$ is a product of the above operators. Once this is proved, it follows from Corollary 4 that those stabilizers are complete.

Let us start with $\mathcal{A}_Z$. Similar to the case of Theorem 1, because of the existence of the $X$-suspension operators and the $\mathcal{O}^X$ operators on the boundary layers, we can reduce $\mathcal{A}_Z$ to an operator that acts on a single layer with an even layer index, by repeatedly multiplying it with the $Z$-suspension operators, the $G^Z$ operators and the $U^Z$ operators. This single-layer operator must be a symmetry (local or nonlocal) of the dual theory, namely a product of the $\Omega^Z_{k,2l}$ and $\Gamma^Z_{\sigma,2l}$ operators for some $l$. Note that using the $Z$-suspension operators, the $G^Z$ operators and the $U^Z$ operators, we are able to generate $\Omega^Z_{k,2l}$ and $\Gamma^Z_{\sigma,2l}$ for all $l$ starting from $\Omega^Z_{k,2l_0}$ and $\Gamma^Z_{\sigma,2l_0}$. We have thus proved that $\mathcal{A}_Z$ is a product of the operators in the statement of the lemma.

Next, we consider $\mathcal{A}_X$. Since $\mathcal{A}_X$ commutes with all the $G^Z$ and $U^Z$ operators, each odd layer of $\mathcal{A}_X$ is a product of several $\mathcal{O}^X$ operators. Therefore, using the $\mathcal{O}^X$ operators on the boundary layers and the $X$-suspension operators, we are able to remove all the odd-layer $X$ operators in $\mathcal{A}_X$. We may thus assume $\mathcal{A}_X$ only contains $X$ operators with even layer indices. Then, to commute with all the $Z$-suspension terms, different even layers of $\mathcal{A}_X$ must differ from each other only by some $\Gamma^X$ and $\Omega^X$ operators. Thus, using the $\Gamma^X$ and $\Omega^X$ operators, $\mathcal{A}_X$ can be cast to the form $\prod_{p\in\Lambda} \prod_{l=1}^{(L-1)/2} X_{p,2l}$ for some set $\Lambda$. This $\mathcal{A}_X$ has to commute with the $\Omega^Z_{k,2l_0}$ and $\Gamma^Z_{\sigma,2l_0}$ operators, which are the $Z$-type symmetry generators for the dual theory. It follows from our Corollary 5 that $\prod_{p\in\Lambda} X_p$ must be a product of $\Delta^X_i$, $\Gamma^X_\rho$ and $\Omega^X_k$. One can

then see that $\mathcal{A}_X$ is a product of the $X$-suspension operators, the boundary $\mathcal{O}^X$ operators, the $\Gamma^X$ and the $\Omega^X$ operators. This completes the proof. $\qquad\square$

The next step is to restrict the set of stabilizers in the above lemma to an independent (and still complete) subset. This is helpful because only independent stabilizers can independently take eigenvalues $\pm 1$. Let us start by defining some notations. The local $Z$-type symmetry generators $G_r^Z$ of the original theory may not be independent, and we can restrict it to an independent subset with $\tilde{n}_Z$ number of elements. This process is equivalent to choosing a basis for a vector space over $\mathbb{F}_2$. Similarly, we denote by $\tilde{n}_X$ the number of independent local $X$-type symmetry generators of the original theory, and denote by $\tilde{m}_X$ ($\tilde{m}_Z$) the number of independent local $X$-type ($Z$-type) symmetry generators of the dual theory. Recall that we denote the full Hilbert spaces of the original and the dual GI models by $\mathcal{L}$ and $\mathcal{L}'$, respectively. Let $\dim\mathcal{L} = N$ and $\dim\mathcal{L}' = N'$. The number of independent $\mathcal{O}_i^X$ ($\Delta_\alpha^Z$) operators in the original (dual) theory is then $N - n_Z - \tilde{n}_Z$ ($N' - m_X - \tilde{m}_X$).

First consider the $\mathcal{O}^X$ operators on layer 1, one of the two boundary layers. As mentioned above, only $N - n_Z - \tilde{n}_Z$ number of them are independent. For our convenience later, we replace these $N - n_Z - \tilde{n}_Z$ number of independent $\mathcal{O}_{i,1}^X$ operators by the following equivalent set of stabilizers: $n_X$ number of $U_{k,1}^X$, $\tilde{n}_X$ number of $G_{s,1}^X$, and additional $N - n_Z - \tilde{n}_Z - n_X - \tilde{n}_X$ number of $\mathcal{O}_{i,1}^X$ operators. The same applies to the $\mathcal{O}^X$ operators on layer $L$, the other boundary layer. It turns out that $U_{k,1}^X$ and $U_{k,L}^X$ are not independent from each other. Using the $X$-suspension operators, and the $\Gamma^X$, $\Omega^X$ operators acting on all even layers, one can generate the operators $U_{k,1}^X U_{k,L}^X$, because each $X$-type symmetry generator of the original theory can be written as a product of a few $\mathcal{O}^X$ operators, and the product is dual to the identity or an $X$-type symmetry generator in the dual theory. Similarly, $G_{s,1}^X G_{s,L}^X$ can also be generated. Therefore, we can remove $U_{k,L}^X$ and $G_{s,L}^X$ as redundancies from our list of stabilizers.

The $X$-suspension operators $\mathcal{O}^X \Delta^X \mathcal{O}^X$ are in general not independent. In particular, given any relation between the $\mathcal{O}_i^X$ operators, say $\prod_{i\in\Lambda} \mathcal{O}_i^X = 1$ for some set $\Lambda$, the operator $\prod_{i\in\Lambda} \mathcal{O}_{i,2l-1}^X \Delta_{i,2l}^X \mathcal{O}_{i,2l+1}^X = \prod_{i\in\Lambda} \Delta_{i,2l}^X$ is an $X$-type symmetry generator acting on the lattice layer $2l$, and is therefore a product of the $\Gamma^X$ and $\Omega^X$ operators. This means that for each $l$, it suffices to retain $N - n_Z - \tilde{n}_Z$ number of the $X$-suspension operators $\mathcal{O}_{i,2l-1}^X \Delta_{i,2l}^X \mathcal{O}_{i,2l+1}^X$. Similarly, for each $l$, it suffices to retain $N' - m_X - \tilde{m}_X$ number of the $Z$-suspension operators $\Delta_{2l}^Z \mathcal{O}_{2l+1}^Z \Delta_{2l+2}^Z$. Things become even simpler when the standard dual theory is used, i.e. when $\Delta_\alpha^Z = Z_\alpha$. In this situation, due to the absence of $\Gamma^X$ and $\Omega^X$ operators, each relation between the $\mathcal{O}_i^X$ operators directly translates to a relation between the $X$-suspension operators, namely $\prod_{i\in\Lambda} \mathcal{O}_i^X = \prod_{i\in\Lambda} \mathcal{O}_{i,2l-1}^X \Delta_{i,2l}^X \mathcal{O}_{i,2l+1}^X = 1$, which will be useful later. Moreover, all the $Z$-suspension operators are independent in this special case.

With all the above procedures of removing redundancies, let us count how many stabilizers we are left with. We have

1. $n_X$ number of $U_{k,1}^X$, $\tilde{n}_X$ number of $G_{s,1}^X$, and additional $2(N - n_Z - \tilde{n}_Z - n_X - \tilde{n}_X)$ number of $\mathcal{O}^X$ operators acting on the two boundary layers.

2. $(N' - m_X - \tilde{m}_X)(L-3)/2$ number of $Z$-suspension operators and $(N - n_Z - \tilde{n}_Z)(L-1)/2$ number of $X$-suspension operators.

3. $(n_Z + \tilde{n}_Z)(L+1)/2$ number of original-theory $Z$-type symmetry generators acting on all odd layers and $(m_X + \tilde{m}_X)(L-1)/2$ number of dual-theory $X$-type symmetry generators acting on all even layers.

4. $m_Z + \tilde{m}_Z$ number of dual-theory $Z$-type symmetry generators acting on some fixed layer $2l_0$.

Theorem 2 about the generalized Kramers-Wannier duality implies that $N - n_Z - \tilde{n}_Z - n_X - \tilde{n}_X = N' - m_X - \tilde{m}_X - m_Z - \tilde{m}_Z$. With this relation, one can check that the total number of stabilizers listed above is $N'(L-1)/2 + N(L+1)/2$, same as the total number of qubits. Therefore, the above stabilizers must be independent. A complete basis of states in the Hilbert space are labeled by the independent eigenvalues of those operators.

## E.2  Simple situation: using the standard dual theory

In this subsection, we restrict to the special situation where the standard dual theory is used, i.e. $\Delta_\alpha^Z = Z_\alpha$. This means that the dual theory has no $X$-type symmetry at all. All the $\Gamma^X$ and $\Omega^X$ operators disappear, and $m_X = \tilde{m}_X = 0$.

Now, we divide the complete and independent set of stabilizers found above into two subsets:

- Subset A: the $Z$-suspension and $X$-suspension operators, $G_{r,2l-1}^Z$ for all $l$, $G_{s,1}^X$ acting on the first layer, and $\Gamma_{\sigma,2l_0}^Z$ acting on the layer $2l_0$.

- Subset B: the $2(N - n_Z - \tilde{n}_Z - n_X - \tilde{n}_X)$ number of $\mathcal{O}^X$ operators acting on the two boundary layers, the $n_X$ number of $U_{k,1}^X$, the $m_Z$ number of $\Omega_{k,2l_0}^Z$, and the $n_Z(L+1)/2$ number of $U_{k,2l-1}^Z$.

Subset A is actually equivalent to the set of stabilizers appearing in the bulk Hamiltonian: Firstly, recall that Subset A already contains all the $Z$-suspension operators, and the $X$-suspension operators contained in Subset A are able to generate all the other $X$-suspension operators, thanks to the absence of $\Omega^X$ and $\Gamma^X$. Moreover, it is obvious that all the $G^Z$ operators can be generated by the independent ones we retained here. Finally, all the $G^X$ operators can be generated using the independent set of $G_{s,1}^X$ acting on the first layer and the $X$-suspension operators. Similarly, all the $\Gamma^Z$ operators are generated by the independent set of $\Gamma_{\sigma,2l_0}^Z$ acting on the layer $2l_0$, the $Z$-suspension operators and the $G^Z$ operators. This result implies that $\mathcal{L}_{\text{bdry}}$, the ground subspace of the bulk Hamiltonian, is completely characterized by the stabilizers in Subset B.

**Example.** Consider the bulk model in Eq. 10 that is constructed from Eq. 3 and Eq. 6. We put the model on a 3D cubic lattice with $L_x \times L_y \times L_z$ number of vertices, with periodic boundary condition along $x$ and $y$, and open boundary condition along $z$. The two boundary surfaces are "smooth", and $L$ is related to $L_z$ by $L = 2L_z - 1$ (the height of the system is $L_z - 1$ number of lattice constants). We have $n_Z = 2$, $\tilde{n}_Z = L_x L_y - 1$, $n_X = L_x + L_y - 2$, $\tilde{n}_X = 0$, $m_Z = L_x + L_y - 1$, $N = 2L_x L_y$, and $N' = L_x L_y$. It follows that $\log_2(\dim \mathcal{L}_{\text{bdry}}) = 2L_x L_y + L$ which we have verified numerically.

It is convenient to divide $\mathcal{L}_{\text{bdry}}$ into several sectors labeled by the eigenvalues of $\Omega_{k,2l_0}^Z$ and $U_{k,2l-1}^Z$. Each of these sectors is characterized by a set of stabilizers that are all products of Pauli $X$ operators. Let us first focus on the sector where $\Omega_{k,2l_0}^Z = 1$ and $U_{k,2l-1}^Z = 1$, denoted by $\tilde{\mathcal{L}}_{\text{bdry},0}$ (the reason for a tilde symbol will be clear below). We denote by $\mathcal{L}$ the Hilbert space of the original $d$-dimensional lattice, and by $\mathcal{L}_G$ the gauge-invariant subspace of it (the symmetric sector for all *local* symmetries). We see that $\tilde{\mathcal{L}}_{\text{bdry},0}$ is isomorphic to the following fictitious space,

$$\tilde{\mathcal{L}}_{\text{fic}} := \mathbb{Z}_2^{n_X + 2n_Z} \text{ symmetric subspace of } \mathcal{L}_G \otimes \mathcal{L}_G, \tag{E.1}$$

where the $\mathbb{Z}_2$ symmetries are generated by $U_k^X \otimes U_k^X$, $U_k^Z \otimes 1$ and $1 \otimes U_k^Z$. We would like to choose the isomorphism from $\tilde{\mathcal{L}}_{\text{bdry},0}$ to the above fictitious space such that the independent stabilizers characterizing $\tilde{\mathcal{L}}_{\text{bdry},0}$ have the most natural correspondence, i.e. $\mathcal{O}_{i,1}^X \mapsto \mathcal{O}_i^X \otimes 1$, $\mathcal{O}_{i,L}^X \mapsto 1 \otimes \mathcal{O}_i^X$, and $U_{k,1}^X \mapsto U_k^X \otimes 1$. Thus, we shall identity the simultaneous eigenstates of the

corresponding stabilizers. One simple consequence is that, *all* the $\mathcal{O}^X$ operators on the two boundary layers, not just the independent ones in Subset B, will satisfy the above mapping rule:

$$\mathcal{O}^X_{i,1} \mapsto \mathcal{O}^X_i \otimes 1, \qquad \mathcal{O}^X_{i,L} \mapsto 1 \otimes \mathcal{O}^X_i. \tag{E.2}$$

However, we have not uniquely determined the isomorphism yet, since the eigenstates have arbitrary phase factors. Thanks to the fact that all the stabilizers are products of Pauli $X$-operators, we can choose orthonormal bases for both $\tilde{\mathcal{L}}_{\text{bdry},0}$ and the fictitious space according to Lemma 3, with the roles of $X$ and $Z$ exchanged. Vectors in these orthonormal bases are automatically eigenstates of the stabilizers and can be identified according to the eigenvalues. As one important consequence of this special choice of bases, the operators $\mathcal{O}^Z_{\alpha,1} Z_{\alpha,2}$ and $Z_{\alpha,L-1} \mathcal{O}^Z_{\alpha,L}$, which all preserve $\tilde{\mathcal{L}}_{\text{bdry},0}$ and can be added to the boundary Hamiltonian, have the simple mapping rule:

$$\mathcal{O}^Z_{\alpha,1} Z_{\alpha,2} \mapsto \mathcal{O}^Z_\alpha \otimes 1, \qquad Z_{\alpha,L-1} \mathcal{O}^Z_{\alpha,L} \mapsto 1 \otimes \mathcal{O}^Z_\alpha. \tag{E.3}$$

We may take the boundary Hamiltonian $H_{\text{bdry}}$ to be a linear combination of the operators $\mathcal{O}^X_{i,1}, \mathcal{O}^X_{i,L}, \mathcal{O}^Z_{\alpha,1} Z_{\alpha,2}$ and $Z_{\alpha,L-1} \mathcal{O}^Z_{\alpha,L}$. From the above discussion, we see that the $\tilde{\mathcal{L}}_{\text{bdry},0}$ block of $H_{\text{bdry}}$, when represented in the fictitious space $\tilde{\mathcal{L}}_{\text{fic}}$, takes the form

$$H^{\text{I}}_{\text{GI}}(J_\alpha, h_i) + H^{\text{II}}_{\text{GI}}(J'_\alpha, h'_i), \tag{E.4}$$

where $H^{\text{I}}_{\text{GI}}$ and $H^{\text{II}}_{\text{GI}}$ act on the two copies of $\mathcal{L}_G$ in Eq. E.1, respectively. We prove in Appendix E.5 that any allowed local term in $H_{\text{bdry}}$ can be generated by the four types of terms considered here and the stabilizers in $H_{\text{bulk}}$. Therefore, our canonical choice of $H_{\text{bdry}}$ is a quite general one.

Other sectors with different eigenvalues of $\Omega^Z_{k,2l_0}$ or $U^Z_{k,2l-1}$ can be analyzed by looking for unitary operators that map them to $\tilde{\mathcal{L}}_{\text{bdry},0}$. Given a set of independent $Z$-type operators $A^Z_1, A^Z_2, \cdots, A^Z_n$ acting on an arbitrary multiple-qubit Hilbert space, there is always a set of $X$-type operators $B^X_1, \cdots, B^X_n$ such that $B^X_k$ anticommutes with $A^Z_k$ but commutes with all the other $A^Z$ operators. It follows that we can always find some $X$-type operators which commute with the bulk Hamiltonian but can independently change the eigenvalues of $\Omega^Z_{k,2l_0}$ and $U^Z_{k,2l-1}$. They generate isomorphisms from all the other sectors of $\mathcal{L}_{\text{bdry}}$ to $\tilde{\mathcal{L}}_{\text{bdry},0}$ which is equivalent to $\tilde{\mathcal{L}}_{\text{fic}}$ as we just shown. These isomorphisms commute with $\mathcal{O}^X_{i,1}$ and $\mathcal{O}^X_{i,L}$, but may anticommute with some $\mathcal{O}^Z_{\alpha,1} Z_{\alpha,2}$ and $Z_{\alpha,L-1} \mathcal{O}^Z_{\alpha,L}$ operators. Therefore, the effective boundary Hamiltonian in each of the other sectors takes the same form as Eq. E.4, but the signs of some GI terms may be flipped.

Let us try to write down the isomorphisms between different sectors more explicitly, which turn out to be illuminating. We start from the $U^Z$ operators acting on the two boundary layers, namely $U^Z_{k,1}$ and $U^Z_{k,L}$. Let $V^X_1, V^X_2, \cdots, V^X_{n_Z}$ be the $X$-type operators acting on $\mathcal{L}$ such that $V^X_k$ anticommutes with $U^Z_k$ but commutes with all the other $Z$-type symmetry generators (local or nonlocal) of the original theory. $V^X_{k,1}$ and $V^X_{k,L}$ commute with the bulk Hamiltonian, and thus can be used to adjust the eigenvalues of $U^Z_{k,1}$ and $U^Z_{k,L}$ within $\mathcal{L}_{\text{bdry}}$, respectively. Each $V^X_{k,1}$ may flip the signs of some $\mathcal{O}^Z$ terms in the $H^{\text{I}}_{\text{GI}}$ part of the effective boundary Hamiltonian. We can, however, apply the unitary operator $V^X_k \otimes 1$ to $\tilde{\mathcal{L}}_{\text{fic}}$ to compensate these sign changes, at the cost of flipping the sign of $U^Z_k \otimes 1$.[13] A similar statement holds for $V^X_{k,L}$. This observation inspires us that there is actually a nicer way of viewing $\mathcal{L}_{\text{bdry}}$: We can alternatively divide $\mathcal{L}_{\text{bdry}}$ into sectors labeled by $\Omega^Z_{k,2l_0}$ and $U^Z_{k,2l-1}$ for all *internal* layers ($3 \leq 2l-1 \leq L-2$). Each new

---

[13] $V^X_k \otimes 1$ is not an automorphism of $\tilde{\mathcal{L}}_{\text{fic}}$, so strictly speaking we shall first embed $\tilde{\mathcal{L}}_{\text{fic}}$ to $\mathcal{L}_G \otimes \mathcal{L}_G$, and then apply $V^X_k \otimes 1$.

sector now includes several old sectors that differ from each other only by the eigenvalues of $U_{k,1}^Z$ and/or $U_{k,L}^Z$. Let $\mathcal{L}_{\mathrm{bdry},0}$ be the new sector where $\Omega_{k,2l_0}^Z = 1$ and $U_{k,2l-1}^Z = 1$ for internal layers. $\mathcal{L}_{\mathrm{bdry},0}$ is isomorphic to a new fictitious space

$$\mathcal{L}_{\mathrm{fic}} := \mathbb{Z}_2^{n_X} \text{ symmetric subspace of } \mathcal{L}_G \otimes \mathcal{L}_G, \tag{E.5}$$

where the $\mathbb{Z}_2$ symmetries are generated by $U_k^X \otimes U_k^X$. The operator mapping rule from $\tilde{\mathcal{L}}_{\mathrm{bdry},0}$ to $\tilde{\mathcal{L}}_{\mathrm{fic}}$ that we established earlier still works here, now from $\mathcal{L}_{\mathrm{bdry}}$ to $\mathcal{L}_{\mathrm{fic}}$. The effective boundary Hamiltonian takes the form of Eq. E.4 as well. Other new sectors are all isomorphic to $\mathcal{L}_{\mathrm{bdry},0}$, and thus to $\mathcal{L}_{\mathrm{fic}}$:

- Denote by $\mathcal{L}'$ the Hilbert space for the $d$-dimensional dual lattice. We may find some $X$-type operators $\Theta_k^X$ acting on $\mathcal{L}'$ such that $\Theta_k^X$ anticommutes with $\Omega_k^Z$ but commutes with all the other $Z$-type symmetry generators (local or nonlocal). It follows that $\prod_{l=1}^{(L-1)/2} \Theta_{k,2l}^X$ is a multilayer operator that commutes with the bulk Hamiltonian and can flip the eigenvalue of $\Omega_{k,2l_0}^Z$. This multilayer operator may flip the signs of some $\mathcal{O}^Z$ terms simultaneously in both $H_{\mathrm{GI}}^{\mathrm{I}}$ and $H_{\mathrm{GI}}^{\mathrm{II}}$.

- Adjusting the values of the internal-layer $U^Z$ operators is more complicated, consisting of three steps: (1) Flip the sign of $U_{k,2l-1}^Z$ using $V_{k,2l-1}^X$ defined earlier, but this may unexpectedly flip the signs of some $Z$-suspension operators as well. (2) Fix the $Z$-suspension operators using string operators of the form $\prod_{l'=1}^{l-1} X_{q,2l'}$ when $l_0 \geq l$ or $\prod_{l'=l}^{(L-1)/2} X_{q,2l'}$ when $l_0 < l$, without affecting the $\Omega_{k,2l_0}^Z$ operators. Notice that the $\Gamma^Z$ operators will not be affected by the above steps either, because $\Gamma_{\sigma,2l_0}^Z$ are unaffected, and the other $\Gamma^Z$ operators can all be generated from the $\Gamma_{\sigma,2l_0}^Z$, $Z$-suspension, and $G^Z$ operators. (3) As an optional step, apply $V_{k,1}^X$ ($V_{k,L}^X$) when $l_0 \geq l$ ($l_0 < l$). With the last step added, the effective boundary Hamiltonian will be invariant under the above operations.

We have seen that altering the eigenvalue of $\Omega_{k,2l_0}^Z$ may change the signs of some $\mathcal{O}^Z$ terms in both the $H_{\mathrm{GI}}^{\mathrm{I}}$ and $H_{\mathrm{GI}}^{\mathrm{II}}$ parts of the effective boundary Hamiltonian. Is it possible to cancel this effect by some additional unitary rotation on $\mathcal{L}_{\mathrm{fic}}$? The answer depends on a certain property of the $\Omega^Z$ operators. Let $\mathcal{G}_Z' = \left\langle \Omega_1^Z, \cdots, \Omega_{m_Z}^Z \right\rangle$ be the group generated by $\Omega^Z$'s in the dual theory. We define a subgroup $\mathcal{G}_{Z,0}' = \{g \in \mathcal{G}_Z' \mid g \text{ is dual to the identity modulo the } G^Z \text{ operators}\}$. We can always redefine the $\Omega^Z$ operators such that for some integer $m_0$, $\mathcal{G}_{Z,0}' = \left\langle \Omega_1^Z, \cdots, \Omega_{m_0}^Z \right\rangle$. We claim and will elaborate below that: *When $k > m_0$ ($k \leq m_0$), it is possible (not possible) to flip the sign of $\Omega_{k,2l_0}^Z$ without affecting the effective boundary Hamiltonian.*

First consider $k \leq m_0$. We write $\Omega_k^Z = \prod_{\alpha \in A} Z_\alpha$ for some subset $A$, then under the Kramers-Wannier operator map, $\Omega_k^Z \mapsto \prod_{\alpha \in A} \mathcal{O}_\alpha^Z = \prod_{r \in R} G_r^Z$ for some subset $R$. In an arbitrary sector of $\mathcal{L}_{\mathrm{bdry}}$,

$$\Omega_{k,2l_0}^Z = \Omega_{k,2}^Z = \prod_{\alpha \in A} (\mathcal{O}_{\alpha,1}^Z Z_{\alpha,2}), \tag{E.6}$$

where we used the fact that $\Omega_{k \leq m_0, 2l_0}^Z$ is related to $\Omega_{k \leq m_0, 2}^Z$ by the multiplication of several $Z$-suspension and $G^Z$ operators. In $\mathcal{L}_{\mathrm{fic}}$, we have

$$1 = \prod_{\alpha \in A} (\mathcal{O}_\alpha^Z \otimes 1). \tag{E.7}$$

Suppose there is an isomorphism from this sector of $\mathcal{L}_{\mathrm{bdry}}$ to $\mathcal{L}_{\mathrm{fic}}$, such that

$$\mathcal{O}_{\alpha,1}^Z Z_{\alpha,2} \mapsto \eta_\alpha \mathcal{O}_\alpha^Z \otimes 1 \quad (\eta_\alpha = \pm 1), \tag{E.8}$$

then we necessarily have $\Omega^Z_{k,2l_0} = \prod_{\alpha \in A} \eta_\alpha$. This means that as we go from one sector to another with a different $\Omega^Z_{k,2l_0}$, some of the $\eta_\alpha$ must change signs!

Next, consider $k > m_0$. We wish to correct all the aforementioned sign changes of the $\mathcal{O}^Z$ terms in the effective boundary Hamiltonian due to the sign flip of $\Omega^Z_{k,2l_0}$. To this end, we restrict the $\mathcal{O}^Z$ operators acting on $\mathcal{L}$ to a subset $\mathcal{O}^Z_{\beta_1}, \mathcal{O}^Z_{\beta_2}, \cdots$ that is independent modulo $G^Z$'s, meaning that each $\mathcal{O}^Z_{\beta_j}$ is not a product of the remaining ones and $G^Z$'s. Then there exist operators $Q^X_{\beta_1}, Q^X_{\beta_2}, \cdots$ such that each $Q^X_{\beta_j}$ anticommutes with $\mathcal{O}^Z_{\beta_j}$ while commutes with all the others in the independent subset and also commutes with all $G^Z$'s. Now, using the operators $Q^X_{\beta_j} \otimes 1$ acting on $\mathcal{L}_{\text{fic}}$, we can freely adjust the signs of the $\mathcal{O}^Z_{\beta_j} \otimes 1$ terms in the effective boundary Hamiltonian. In other words, we can freely adjust the signs of $\eta_{\beta_j}$ whose definition is in Eq. E.8 above. In fact, once we correct all the sign changes of $\eta_{\beta_j}$ due to the sign flip of $\Omega^Z_{k,2l_0}$, the sign changes of all $\eta_\alpha$ are also corrected. To see this, notice that given any relation of the form $\prod_{\alpha \in A} \mathcal{O}^Z_\alpha = 1 \mod G^Z$, $\prod_{\alpha \in A} Z_\alpha$ must be an element of $\mathcal{G}'_{Z,0}$ modulo some $\Gamma^Z$'s, which follows from the definition of $\mathcal{G}'_{Z,0}$ as well as the fact that each $\Gamma^Z$ operator is dual to the product of some $G^Z$'s, and thus $\prod_{\alpha \in A} \eta_\alpha$ is equal to the eigenvalue of this element of $\mathcal{G}'_{Z,0}$ acting on the $2l_0$-th layer. Since each $\mathcal{O}^Z_\alpha$ is a product of some $\mathcal{O}^Z_{\beta_j}$ and $G^Z$ operators, each $\eta_\alpha$ is then a product of some $\eta_{\beta_j}$ and the eigenvalue of some element of $\mathcal{G}'_{Z,0}$ acting on the $2l_0$-th layer. Flipping the sign of $\Omega^Z_{k>m_0,2l_0}$ does not affect any $\Omega^Z_{k',2l_0}$ with $k' \leq m_0$, therefore our statement above about $\eta_\alpha$ is indeed true.

With all these discussions, we propose the following necessary and sufficient condition for anomaly.

**Claim.** *When the standard dual theory is used for constructing the bulk model, the boundary theory is anomalous if and only if either of the following two conditions is satisfied:*

1. *$n_X \geq 1$.*

2. *For some nonempty subset $K \subset \{1, 2, \cdots, m_Z\}$, $\prod_{k \in K} \Omega^Z_k$ is dual to the identity modulo the $G^Z$ operators.*

The first condition guarantees a symmetry charge projection $U^X_k \otimes U^X_k = 1$ acting on the two copies of the GI model in the boundary theory; each copy is allowed to have states with both values of the symmetry charge, but the total charge of the two copies is fixed. When the second condition holds, $\prod_{k \in K} \Omega^Z_k$ is dual to a generalized boundary condition, and a boundary condition projection is applied to the two copies of the GI model in the boundary theory; each copy is allowed to take both values of the boundary condition, but the boundary condition values of the two copies are locked together. When neither condition is satisfied, the boundary theory is a direct sum of *identical* sectors. The Hilbert space of each sector is isomorphic to $\mathcal{L}_G \otimes \mathcal{L}_G$ with the operator mapping rule in Eq. E.2 and E.3. The effective boundary Hamiltonian of each sector takes the form of Eq. E.4, or more generally consists of local terms generated by those in Eq. E.4.

## E.3 General situation

To work with the most general situation, let us recall some notations defined in Appendix D. In general, the nonlocal $Z$-type symmetry group $\mathcal{G}_Z \cong \mathbb{Z}^{n_Z}_2$ of our GI model contains a subgroup $\mathcal{G}_{Z,0} \cong \mathbb{Z}^\nu_2$, whose definition is $\mathcal{G}_{Z,0} = \{g \in \mathcal{G}_Z \mid g = \prod_{\alpha \in A} \mathcal{O}^Z_\alpha \prod_{r \in R} G^Z_r$ for some sets $A$ and $R$ satisfying $\prod_{\alpha \in A} \Delta^Z_\alpha = 1\}$. We can always choose our symmetry generators such that $\mathcal{G}_{Z,0}$ is generated by $U^Z_1, \cdots, U^Z_\nu$. Similarly, the nonlocal $X$-type symmetry group $\mathcal{G}'_X \cong \mathbb{Z}^{m_X}_2$ of the dual theory contains a $\mathbb{Z}^\mu_2$ subgroup, whose definition is $\mathcal{G}'_{X,0} = \{g \in \mathcal{G}'_X \mid g = \prod_{i \in I} \Delta^X_i \prod_{\rho \in R} \Gamma^X_\rho$ for

some sets $I$ and $R$ satisfying $\prod_{i \in I} \mathcal{O}_i^X = 1$}. Without loss of generality, we assume this $\mathbb{Z}_2^\mu$ subgroup is generated by $\Omega_1^X, \cdots, \Omega_\mu^X$.

As in the previous case, we divide the complete and independent set of stabilizers for the full Hilbert space into two subsets:

- Subset A: the $Z$-suspension and $X$-suspension operators, $G_{r,2l-1}^Z$ for all $l$, $G_{s,1}^X$ acting on the first layer, $\Gamma_{\rho,2l}^X$ for all $l$, $\Gamma_{\sigma,2l_0}^Z$ acting on the layer $2l_0$, $U_{1,2l-1}^Z, \cdots, U_{\nu,2l-1}^Z$ for all the *internal* layers ($3 \le 2l-1 \le L-2$), and $\Omega_{1,2l}^X, \cdots, \Omega_{\mu,2l}^X$ for all $l$.

- Subset B: the $2(N - n_Z - \tilde{n}_Z - n_X - \tilde{n}_X)$ number of $\mathcal{O}^X$ operators acting on the two boundary layers, the $n_X$ number of $U_{k,1}^X$, the $m_Z$ number of $\Omega_{k,2l_0}^Z$, the $2n_Z$ number of $U_{k,1}^Z$ and $U_{k,L}^Z$ acting on the boundary layers, the $(n_Z - \nu)(L-3)/2$ number of $U_{k>\nu,2l-1}^Z$ for all the *internal* layers ($3 \le 2l-1 \le L-2$), and the $(m_X - \mu)(L-1)/2$ number of $\Omega_{k>\mu,2l}^X$ for all $l$.

One can check that Subset A is equivalent to the set of stabilizers appearing in the bulk Hamiltonian. In particular, the $Z$-suspension operators contained in Subset A are able to generate all the other $Z$-suspension operators with the help of the included $G^Z$ and $U^Z$ operators. A similar statement applies to the $X$-suspension operators. Also notice that the stabilizers in the bulk Hamiltonian are able to generate Subset A. This result implies that $\mathcal{L}_{\text{bdry}}$ is completely characterized by the stabilizers in Subset B.

**Example.**  Consider the model in Appendix C. We take open boundary condition along the out-of-layer direction, and the number of layers $L$ is an odd integer as explained in Section 5. One may check that $n_Z = m_X = 2$, $\tilde{n}_Z = L_x L_y - 1$, $n_X = m_Z = L_x - 1$, $\tilde{n}_X = 0$, and

$$\nu = \mu = \begin{cases} 1 & (L_x \text{ odd}), \\ 0 & (L_x \text{ even}). \end{cases} \tag{E.9}$$

It follows that

$$\log_2(\dim \mathcal{L}_{\text{bdry}}) = \begin{cases} 2L_x L_y + L & (L_x \text{ odd}), \\ 2L_x L_y + 2L - 2 & (L_x \text{ even}), \end{cases} \tag{E.10}$$

which we have verified numerically.

We can divide $\mathcal{L}_{\text{bdry}}$ into several sectors labeled by the eigenvalues of $\Omega_{k,2l_0}^Z$, $U_{k>\nu,2l-1}^Z$ for all *internal* layers, and $\Omega_{k>\mu,2l}^X$. Denote by $\mathcal{L}_{\text{bdry},0}$ the sector where these operators all equal to 1. Then $\mathcal{L}_{\text{bdry},0}$ is again isomorphic to the fictitious space in Eq. E.5. The operator mapping rule is also similar:

$$\begin{aligned} \mathcal{O}_{i,1}^X &\mapsto \mathcal{O}_i^X \otimes 1, & \mathcal{O}_{i,L}^X &\mapsto 1 \otimes \mathcal{O}_i^X, \\ \mathcal{O}_{\alpha,1}^Z \Delta_{\alpha,2}^Z &\mapsto \mathcal{O}_\alpha^Z \otimes 1, & \Delta_{\alpha,L-1}^Z \mathcal{O}_{\alpha,L}^Z &\mapsto 1 \otimes \mathcal{O}_\alpha^Z. \end{aligned} \tag{E.11}$$

By taking $H_{\text{bdry}}$ as a linear combination of $\mathcal{O}_{i,1}^X$, $\mathcal{O}_{i,L}^X$, $\mathcal{O}_{\alpha,1}^Z \Delta_{\alpha,2}^Z$ and $\Delta_{\alpha,L-1}^Z \mathcal{O}_{\alpha,L}^Z$, the $\mathcal{L}_{\text{bdry},0}$ block of $H_{\text{bdry}}$ may again have the form of Eq. E.4 when represented in $\mathcal{L}_{\text{fic}}$. We prove in Appendix E.5 that any allowed local term in $H_{\text{bdry}}$ can be generated by the four types of terms considered here and the stabilizers in $H_{\text{bulk}}$.

The next task is to establish isomorphisms from other sectors to $\mathcal{L}_{\text{bdry},0}$, and therefore to the fictitious space:

- As before, the eigenvalue of each $\Omega_{k,2l_0}^Z$ can be flipped by the operator $\prod_{l=1}^{(L-1)/2} \Theta_{k,2l}^X$ without affecting the $U^Z$ or $\Omega^X$ operators. This multilayer operator may flip the signs of some $\mathcal{O}^Z$ terms simultaneously in both $H_{\text{GI}}^{\text{I}}$ and $H_{\text{GI}}^{\text{II}}$.

- The $V^X_{k,2l-1}$ operator mentioned previously is able to flip the eigenvalue of $U^Z_{k,2l-1}$ where $3 \leq 2l-1 \leq L-2$ and $k > \nu$, but it may also anticommute with some $Z$-suspension operators. On each internal layer, there are only $N'-m_X-\tilde{m}_X$ number of independent $Z$-suspension operators, corresponding to the same number of independent $\Delta^Z$ operators acting on $\mathcal{L}'$; the remaining ones can be generated with the help of the $G^Z_r$ and $U^Z_{k \leq \nu,2l-1}$ operators. Thus, it suffices to fix the eigenvalues of these independent $Z$-suspension operators. To this end, we first choose an independent subset of the $\Delta^Z$ operators acting on $\mathcal{L}'$: $\Delta^Z_{\alpha_1}, \Delta^Z_{\alpha_2}, \cdots$. Then, there exists a set of operators $E^X_{\alpha_1}, E^X_{\alpha_2}, \cdots$ such that $E^X_{\alpha_j}$ anticommutes with $\Delta^Z_{\alpha_j}$ while commutes with all the other independent $\Delta^Z$ operators. We can use the multilayer operators $\prod^{l-1}_{l'=1} E^X_{\alpha_j,2l}$ ($\prod^{(L-1)/2}_{l'=l} E^X_{\alpha_j,2l}$) when $l_0 \geq l$ ($l_0 < l$) to fix the eigenvalues of the $Z$-suspension operators centered on the $(2l-1)$-th layer without affecting $\Omega^Z_{k,2l_0}$. For a reason explained previously, the $\Gamma^Z$ operators will not be affected by the above steps either. These multilayer operators may anticommute with some $\mathcal{O}^Z_{\alpha,1}\Delta^Z_{\alpha,2}$ or $\Delta^Z_{\alpha,L-1}\mathcal{O}^Z_{\alpha,L}$ operators, thus affecting the effective boundary Hamiltonian, but this effect can be completely canceled by additionally applying $V^X_{k,1}$ or $V^X_{k,L}$. In other words, adjusting the values of the internal-layer $U^Z_{k>\nu,2l-1}$ operators does not affect the effective boundary Hamiltonian.

- The eigenvalues of $\Omega^X_{k>\mu,2l}$ can be adjusted as follows. Let $\Theta^Z_1, \cdots, \Theta^Z_{m_X}$ be $Z$-type operators acting on $\mathcal{L}'$ such that $\Theta^Z_k$ anticommutes with $\Omega^X_k$ while commutes with all the other local or nonlocal $X$-type symmetry generators of the dual theory. $\Theta^Z_{k>\mu,2l}$ is able to flip the sign of $\Omega^X_{k>\mu,2l}$, but it may anticommute with some $X$-suspension operators. Analogous to the $E^X$ operators define above, we can find some $\mathcal{P}^Z_{i_n}$ operators acting on $\mathcal{L}$ such that they can flip the signs of an independent subset of $\mathcal{O}^X$ operators, $\mathcal{O}^X_{i_1}, \mathcal{O}^X_{i_2}, \cdots$, respectively. The multilayer operators $\prod^l_{l'=1} \mathcal{P}^Z_{i_n,2l'-1}$ can be used to fix the eigenvalues of the $X$-suspension operators centered at the $2l$-th layer. The $G^X$ operators will not be affected by the above steps. These multilayer operators may anticommute with some $\mathcal{O}^X_{i,1}$ operators, or equivalently flip the signs of some $\mathcal{O}^X$ operators in the $H^I_{GI}$ part of the effective boundary Hamiltonian.

The sign changes of the $\mathcal{O}^Z$ terms in the effective boundary Hamiltonian due to the sign flip of $\Omega^Z_{k,2l_0}$ can be analyzed in essentially the same way as we did in the previous subsection right above the claim of anomaly condition, with the same conclusion. Thus we will not repeat the discussion again, but just note that in the general situation, the definition of $\mathcal{G}'_{Z,0}$ should be modified to $\mathcal{G}'_{Z,0} = \{g \in \mathcal{G}'_Z \mid g = \prod_{\alpha \in A} \Delta^Z_\alpha$ for some set $A$ such that $\prod_{\alpha \in A} \mathcal{O}^Z_\alpha = 1 \mod G^Z\}$.

We have seen that altering the eigenvalue of $\Omega^X_{k>\mu,2l}$ may change the signs of some $\mathcal{O}^X$ terms in $H^I_{GI}$. Denote the combined operator we described above for flipping the sign of $\Omega^X_{k>\mu,2l}$ by $\Theta^Z_{k,2l} \prod^l_{l'=1} \mathcal{B}^Z_{2l'-1}$ where $\mathcal{B}^Z$ is a product of several $\mathcal{P}^Z$ operators. Applying the unitary operator $\mathcal{B}^Z \otimes 1$ to $\mathcal{L}_{fic}$ is able to cancel all the sign changes of the $\mathcal{O}^X$ terms in $H^I_{GI}$, but this may flip the signs of $U^X_p \otimes U^X_p$ for some $p$, since $\mathcal{B}^Z$ may anticommute with some $U^X_p$ operators.[14] This raises an important question: Is there still some symmetry charge projection, such as $U^X_p \otimes U^X_p$ for some $p$, that applies to all sectors of the boundary theory, and thus can be regarded as a source of anomaly? The answer depends on a certain property of the $U^X$ operators. Let $\mathcal{G}_X = \langle U^X_1, \cdots, U^X_{n_X} \rangle$ be the group generated by the $U^X$'s in the original theory. We define a subgroup $\mathcal{G}_{X,0} = \{g \in \mathcal{G}_X \mid g = \prod_{i \in I} \mathcal{O}^X_i$ for some set $I$ such that $\prod_{i \in I} \Delta^X_i = 1 \mod \Gamma^X\}$. We

---

[14] $\mathcal{B}^Z \otimes 1$ may not be an automorphism of $\mathcal{L}_{fic}$, so strictly speaking we shall first embed $\mathcal{L}_{fic}$ to $\mathcal{L}_G \otimes \mathcal{L}_G$, and then apply $\mathcal{B}^Z \otimes 1$.

can always redefine the $U^X$ operators such that for some integer $n_0$, $\mathcal{G}_{X,0} = \left\langle U_1^X, \cdots, U_{n_0}^X \right\rangle$. We claim and will elaborate below that: $U_p^X \otimes U_p^X = 1$ *holds in the whole boundary theory for* $1 \leq p \leq n_0$ *but not for* $p > n_0$.

First consider $p \leq n_0$. We note that the operator $\Theta_{k,2l}^Z \prod_{l'=1}^{l} \mathcal{B}_{2l'-1}^Z$ mentioned above commutes with $U_{p,1}^X U_{p,L}^X$ that is equal to a product of some $X$-suspension and $\Gamma^X$ operators. It follows that $\mathcal{B}^Z$ commutes with $U_p^X$, and the symmetry charge projection $U_p^X \otimes U_p^X = 1$ for this $p$ is not affected by altering the eigenvalue of $\Omega_{k>\mu,2l}^X$.

Next, consider $p > n_0$. The key is again to understand the dual of $U_p^X$. Each $U^X$ operator can be written as a product of $\mathcal{O}^X$ operators, though there may be multiple ways of doing it, thus we can always define some dual for each $U_p^X$. From the definition of $\mathcal{G}_{X,0}$, we see that the dual operator for each $U_{p>n_0}^X$, up to $\Gamma^X$'s, is a nontrivial product of some $\Omega^X$ operators, which we will denote as $\tilde{\Omega}_p^X$. We claim that any nontrivial product of the $\tilde{\Omega}_p^X$ $(p > n_0)$ operators does not belong to $\mathcal{G}'_{X,0}$ which is defined both in Appendix D and at the beginning of this subsection. Suppose this statement is not true, then there is some nonempty subset $P \subset \{n_0+1, n_0+2, \cdots, n_X\}$ such that the following equations hold.

$$\prod_{p \in P} U_p^X = \prod_{i \in I} \mathcal{O}_i^X \mapsto \prod_{i \in I} \Delta_i^X = \prod_{p \in P} \tilde{\Omega}_p^X \mod \Gamma^X, \tag{E.12}$$

$$\prod_{p \in P} \tilde{\Omega}_p^X = \prod_{i \in J} \Delta_i^X \mod \Gamma^X, \quad \text{s.t.} \quad \prod_{i \in J} \mathcal{O}_i^X = 1. \tag{E.13}$$

It follows that we can alternatively write $\prod_{p \in P} U_p^X = \prod_{i \in I \cup J} \mathcal{O}_i^X$ which is dual to the product of some $\Gamma^X$ operators, contradicting the fact that $\prod_{p \in P} U_p^X$ does not belong to $\mathcal{G}_{X,0}$. Hence, our statement above about the $\tilde{\Omega}_p^X$ operators is indeed true. Consequently, up to a redefinition of the $\Omega^X$ operators, we can simply identify $\tilde{\Omega}_{n_0+1}^X, \tilde{\Omega}_{n_0+2}^X, \cdots$ as $\Omega_{\mu+1}^X, \Omega_{\mu+2}^X, \cdots$, respectively. Now observe that $U_{n_0+j,1}^X U_{n_0+j,L}^X$ is a product of the $X$-suspension operators, the $\Gamma^X$ operators, and $\Omega_{\mu+j,2l}^X$ for all $l$. It follows that the effect of flipping the sign of $\Omega_{\mu+j,2l}^X$ for any $l$ is to flip the sign of $U_{n_0+j}^X \otimes U_{n_0+j}^X$ while leaving the effective boundary Hamiltonian invariant. As a result, if we conbine all sectors of the boundary theory together, $U_p^X \otimes U_p^X$ for $p > n_0$ no longer have definite values.

With all these discussions, we propose the following necessary and sufficient condition for anomaly.

**Claim.** *The boundary theory is anomalous if and only if either of the following two conditions is satisfied:*

1. *For some nonempty subset $K \subset \{1, 2, \cdots, n_X\}$, $\prod_{k \in K} U_k^X$ can be written as a product of $\mathcal{O}^X$ operators such that the product is dual to the identity modulo the $\Gamma^X$ operators.*

2. *For some nonempty subset $K \subset \{1, 2, \cdots, m_Z\}$, $\prod_{k \in K} \Omega_k^Z$ can be written as a product of $\Delta^Z$ operators such that the product is dual to the identity modulo the $G^Z$ operators.*

The first condition guarantees a symmetry charge projection acting on the two copies of the GI model in the boundary theory; each copy is allowed to have states with both values of the symmetry charge, but the total charge of the two copies is fixed. When the second condition holds, $\prod_{k \in K} \Omega_k^Z$ is dual to a generalized boundary condition, and a boundary condition projection is applied to the two copies of the GI model in the boundary theory; each copy is allowed to take both values of the boundary condition, but the boundary condition values of the two copies are locked together. When neither condition is satisfied, the boundary theory is a direct sum of *identical* sectors. The Hilbert space of each sector is isomorphic to $\mathcal{L}_G \otimes \mathcal{L}_G$ with the

operator mapping rule in Eq. E.11. The effective boundary Hamiltonian of each sector takes the form of Eq. E.4, or more generally consists of local terms generated by those in Eq. E.4. Note that each (new) sector here may be the sum of several (old) sectors discussed previously with different values of $U_k^X \otimes U_k^X$.

### E.4 Violating the anomaly conditions

In what cases is the necessary and sufficient condition of anomaly in Claim 1 violated? We find the following result.

**Theorem 4.** *If the anomaly condition is violated, the bulk theory is either trivial, or fracton ordered. Equivalently, if the bulk theory has a topological order (not including fracton order), then the anomaly condition must be statisfied.*

*Proof.* Claim 1 consists of two sufficient conditions of anomaly. They combine to give a necessary condition; when both are violated, the boundary theory is claimed to be nonanomalous.

The simplest situation where the two anomaly conditions are both violated is $n_X + m_Z = 0$. It then follows from Theorem 3 that with periodic boundary condition, the bulk GSD is either trivial or system size dependent, meaning that the bulk model is either trivial or fractonic.

Next, suppose $n_X \geq 1$. The discussion for the case $m_Z \geq 1$ is similar and will not be repeated. Take some arbitrary $U^X$ operator, say $U_a^X$ with $a \in \{1, 2, \cdots, n_X\}$. We can always write $U_a^X = \prod_{i \in I} \mathcal{O}_i^X$ for some index set $I$. We then must have

$$\prod_{i \in I} \Delta_i^X = \prod_{b \in B} \Omega_b^X \mod \Gamma^X, \qquad (E.14)$$

for some nonempty set $B$, otherwise the first anomaly condition would be satisfied. Denote by $\Omega_B^X := \prod_{b \in B} \Omega_b^X$. We have $\Omega_B^X \in \mathcal{G}'_{X,1}$ by definition (see Appendix D). We now claim that $\Omega_B^X \notin \mathcal{G}'_{X,0}$. If this claim were not right, we would have $\Omega_B^X = \prod_{i \in J} \Delta_i^X \mod \Gamma^X$ for some set $J$ (not necessarily the same as $I$) such that $\prod_{i \in J} \mathcal{O}_i^X = 1$. We can then alternatively write $U_a^X = \prod_{i \in I \cup J} \mathcal{O}_i^X$ with the property that $\prod_{i \in I \cup J} \Delta_i^X = 1 \mod \Gamma^X$, contradicting our assumption that the first anomaly condition is violated. We have thus found that $\bar{\mu} - \mu \geq 1$ (see Appendix D). According to Theorem 3, the bulk GSD with periodic boundary condition grows as the system size increases, hence the bulk model has a fracton order. $\square$

It is possible to design a concrete fractonic example that violates both sufficient conditions of anomaly. Consider the following GI model whose qubits live on the links of a 2D square lattice with periodic boundary condition,

$$H = -J \overset{Z}{\underset{}{\circ}} - h \; X \overset{X}{\underset{X}{\circ}} X \;, \qquad (E.15)$$

and consider its standard dual theory which takes exactly the same form. The model has $n_X = m_X = 0$, $n_Z = m_Z = 2$. We can take the $G^Z$'s ($U^Z$'s) to be the same as the $\Gamma^Z$'s ($\Omega^Z$'s). The first anomaly condition about $U^X$ operators is trivially violated. The second anomaly condition is also violated because in this example, the generalized Kramers-Wannier duality *is trivial*. Let us not go into the details, but one can show that with the setup in Section 5, the boundary theory of this model is indeed nonanomalous: The boundary Hilbert space is $2^{L-1}$ copies of $\mathcal{L}_G \otimes \mathcal{L}_G$, and the effective boundary Hamiltonian takes the same form in all sectors.

### E.5 On possible boundary terms

As we defined in the main text, an allowed boundary term is a local operator that commutes with the stabilizers in the bulk Hamiltonian in Section 5 (open boundary condition along the out-of-layer direction). The boundary terms that we have considered so far are $\mathcal{O}_{i,1}^X$, $\mathcal{O}_{i,L}^X$, $\mathcal{O}_{\alpha,1}^Z \Delta_{\alpha,2}^Z$, and $\Delta_{\alpha,L-1}^Z \mathcal{O}_{\alpha,L}^Z$. This set turns out to be complete in the following sense.

**Lemma 7.** *Given any local operator $\mathcal{A}$ that is a product of Pauli operators and commutes with all the stabilizers appearing in the bulk Hamiltonian in Section 5, $\mathcal{A}$ can be locally generated by those bulk stabilizers together with $\mathcal{O}_{i,1}^X$, $\mathcal{O}_{i,L}^X$, $\mathcal{O}_{\alpha,1}^Z \Delta_{\alpha,2}^Z$, and $\Delta_{\alpha,L-1}^Z \mathcal{O}_{\alpha,L}^Z$.*

*Proof.* The proof is not much different from that for Theorem 1. We again write $\mathcal{A} = \mathcal{A}_Z \mathcal{A}_X$. Both $\mathcal{A}_Z$ and $\mathcal{A}_X$ must commute with all the stabilizers in the bulk Hamiltonian, and we will prove that both of them satisfy the claim of the lemma.

Let us start from $\mathcal{A}_Z$. Denote by $l_{\max}$ and $l_{\min}$ the maximal and minimal layer indices of $\mathcal{A}_Z$'s support, respectively. Since $\mathcal{A}_Z$ is supposed to be local, i.e. small compared to the system size, either $l_{\min} \gg 1$ or $l_{\max} \ll L$. These two scenarios are nearly identical, so let us just assume $l_{\max} \ll L$. We can then use the reduction procedure described in the proof of Theorem 1 to reduce $l_{\max}$ until $l_{\max} \leq 2$, if $\mathcal{A}_Z$ is not yet fully reduced to the identity. Now suppose $l_{\max} = 2$, in order to commute with the $\Gamma^X$ operators on the 2nd layer, the top layer of $\mathcal{A}_Z$ must be a local product of some $\Delta^Z$ operators. Thus, we can further reduce $\mathcal{A}_Z$ using the $\mathcal{O}_{\alpha,1}^Z \Delta_{\alpha,2}^Z$ operators so that it no longer has any support on the 2nd layer. If $l_{\max} = 1$, in order to commute with the $X$-suspension operators spanning the 1st, 2nd and 3rd layers, this single-layer $\mathcal{A}_Z$ must be a local product of some $G^Z$ operators. This completes the proof for $\mathcal{A}_Z$.

Next, we consider $\mathcal{A}_X$. We similarly define $l_{\max}$ and $l_{\min}$, and assume $l_{\max} \ll L$ without loss of generality. Using the reduction procedure in the proof of Theorem 1, we can reduce $l_{\max}$ all the way to 1, if $\mathcal{A}_X$ is not yet fully reduced to the identity operator. If $l_{\max} = 1$, in order to commute with the $G^Z$ operators, this single-layer $\mathcal{A}_X$ must be a local product of the $\mathcal{O}_{i,1}^X$ operators. We have thus completed the proof for $\mathcal{A}_X$. $\qquad\square$

Some minor technical comments: (1) The above result still holds without assuming $\mathcal{A}$ to be a product of Pauli operators. We can expand $\mathcal{A}$ as a superposition of the linearly independent tensor product operators of $I, X, Y, Z$. Since Pauli operators either commute or anticommute, $S \mathcal{A} S^{-1} = \mathcal{A}$ for a bulk stabilizer $S$ implies that any tensor product operator entering the expansion of $\mathcal{A}$ with a nonzero coefficient must also commute with $S$. (2) If $\mathcal{A}$ is a local operator that preserves $\mathcal{L}_{\text{bdry}}$ but does not commute with the bulk Hamiltonian, its effect on $\mathcal{L}_{\text{bdry}}$ is equivalent to some local operator that commutes with the bulk stabilizers, so it is unnecessary to consider such an operators as a boundary term candidate. To see this, notice that this operator $\mathcal{A}$ overlaps with at most finite number of bulk stabilizers since it is local. We can then repeatedly "symmetrize" the operator by $\mathcal{A} \mapsto (\mathcal{A} + S \mathcal{A} S^{-1})/2$ for each overlapping bulk stabilizer $S$. The resulting new operator has the same action on $\mathcal{L}_{\text{bdry}}$, commutes with all the bulk stabilizers, and has a linear size exceeding the old one by at most an O(1) amount.

## F The polynomial representation and topological orders

We derive the properties of the bulk model in (54) through the polynomial representation introduced in [58], see also [65] for pedagogical purpose. The Hamiltonian local terms are represented by a stabilizer map, which is a matrix with elements in $\mathbb{F}_2[x, x^{-1}, y, y^{-1}]$, the

Laurent polynomials whose coefficients are in $\mathbb{F}_2$,

$$S = \begin{pmatrix} 1 & 1+y & 0 & 0 \\ 1+\bar{x} & 0 & 0 & 0 \\ 1+\bar{y} & 0 & 0 & 0 \\ 0 & 1+\bar{x} & 0 & 0 \\ 0 & 0 & 1+x & 0 \\ 0 & 0 & 1 & 1+y \\ 0 & 0 & 0 & 1+x \\ 0 & 0 & 1+\bar{y} & 0 \end{pmatrix}, \tag{F.1}$$

where $\bar{x} \equiv x^{-1}$ and $\bar{y} \equiv y^{-1}$. The model has a robust GSD. This is determined from that $\ker \epsilon_S = \operatorname{im} S$, where $\epsilon_S$ is the excitation map. [65] The Hamiltonian in the stabilizer formalism with such a property is also called an exact code.

We compute the GSD on a square lattice with $L_x \times L_y$ sites and periodic boundary conditions. Treating the $\hat{y}$ as the layer-indexed direction, we also assume $L_y$ to be even. Since the number of types of stabilizers in the Hamiltonian is the same as the number of qubits in a unit cell, the GSD can be computed from

$$\log_2 D = \dim_{\mathbb{F}_2} \operatorname{coker} S^{\dagger}. \tag{F.2}$$

The block-diagonal structure in $S^{\dagger}$ allows us to reduce the evaluation further [108, 109] to

$$\log_2 D = \dim_{\mathbb{F}_2} \operatorname{coker}(S^Z)^{\dagger} + \dim_{\mathbb{F}_2} \operatorname{coker}(S^X)^{\dagger} \tag{F.3}$$

$$= \dim_{\mathbb{F}_2} \frac{\mathbb{F}_2[x,y]}{I((S^Z)^{\dagger}) \oplus \mathbf{b}_L} + \dim_{\mathbb{F}_2} \frac{\mathbb{F}_2[x,y]}{I((S^X)^{\dagger}) \oplus \mathbf{b}_L}. \tag{F.4}$$

where $S^Z$ and $S^X$ are sub-matrices of the block matrix $S = [S^Z, 0; 0, S^X]$, $I(\sigma)$ is the ideal of $\sigma$, and $\mathbf{b}_L$ is the ideal generated by the polynomials that declare boundary conditions.

We find that for the periodic boundary conditions represented by $\mathbf{b}_L = \langle x^{L_x} - 1, y^{L_y} - 1 \rangle$ with even $L_y$, the associated ideals represented in a Groebner basis [65] are

$$I\left((S^Z)^{\dagger}\right) \oplus \mathbf{b}_L = I\left((S^X)^{\dagger}\right) \oplus \mathbf{b}_L = \langle 1+y^2, 1+x \rangle. \tag{F.5}$$

It follows that

$$\log_2 D = 2 \dim_{\mathbb{F}_2} \mathbb{F}_2^2 = 4. \tag{F.6}$$

The degeneracy is the same as two copies of toric code.

Similarly, we can also obtain the GSD for the $\mathbb{Z}_2 \times \mathbb{Z}_2$ model given in (48) from the polynomial representation. The stabilizer map for this model is

$$S_{\mathbb{Z}_2 \times \mathbb{Z}_2} = \begin{pmatrix} 1+x+x^{-1} & 0 \\ y+y^{-1} & 0 \\ 0 & y+y^{-1} \\ 0 & 1+x+x^{-1} \end{pmatrix}. \tag{F.7}$$

The model is locally topological ordered, implying the absence of symmetry breaking order. This is determined by that $S_{\mathbb{Z}_2 \times \mathbb{Z}_2}$ satisfy the codimension condition,[15] $\operatorname{codim} I(S_C) \geq 2$, where $S_C$ is the stabilizer map of the classical spin model, which becomes the quantum model after

---

[15] We recall that the codimension of an ideal of polynomials $f_i$ is the codimension of the algebraic variety defined by the system of polynomial equations $f_i = 0$. For instance, in two-dimensional space, the codimension of $\langle (x-1)(y-1), x-1 \rangle$ is 1, and the codimension of $\langle x-1, y-1 \rangle$ is 2.

gauging, and $I(S_C)$ is the ideal of $S_C$. In the current case, $S_C$ is the map $(1 + x + x^{-1}, y + y^{-1})$, and $\operatorname{codim} I(S_C) = 2$.

The GSD on a torus can be evaluated by

$$\log_2 D = 2 \dim_{\mathbb{F}_2} \frac{\mathbb{F}_2[x, y]}{\langle 1 + x + x^2, 1 + y^2, x^{3L_x} - 1, y^{L_y} - 1 \rangle} \,. \tag{F.8}$$

Since,

$$\frac{\mathbb{F}_2[x, y]}{\langle 1 + x + x^2, 1 + y^2, x^{3L_x} - 1, y^{L_y} - 1 \rangle} \cong \mathbb{F}_2^2 \,, \tag{F.9}$$

we obtain that

$$\log_2 D = 4 \,. \tag{F.10}$$

## G  The variant bulk construction

We explain the variation from the basic bulk construction, which allows us to produce the pure topologically ordered model from the SPT model in (52).

Towards constructing the bulk model, we note that (52) does not satisfy the assumptions in Section 2. The price is that the GSD in the bulk model we would obtain from the basic construction is not robust. Part of the degeneracy originates from symmetry breaking orders. Nevertheless, let us give a minimal variation of the construction. This is enough to provide us a pure topologically ordered bulk.

Firstly, we note that the GI model we begin with has the following property. We group the operators $\{O_\alpha^Z\}$ and $\{O_i^X\}$ according to their commutation relations.

1. $\mathcal{A}_1 = \{O_\alpha^Z, O_i^X\}$,

2. $\mathcal{A}_2 = \{O_{\alpha'}^Z, O_{i'}^X\}$.

The two sets have the property that all operators in each set commute with each other; for any $Z$-type ($X$-type) operator in one set, there are $X$-type ($Z$-type) of operators in the other set that anti-commutes with it. By design, $O_0^Z \in \mathcal{A}_1$.

Now we give the modified rule in constructing the bulk model. For local operators in $\mathcal{A}_1$, their corresponding three-layer local operators are $\Delta_{\alpha,l-1}^Z O_{\alpha,l}^Z \Delta_{\alpha,l+1}^Z$ and $\Delta_{i,l-1}^X O_{i,l}^X \Delta_{i,l+1}^X$ in the bulk model and center at odd layers *i.e.*,$l \in 2\mathbb{Z} + 1$. On the other hand, for local operators in $\mathcal{A}_2$, their corresponding three-layer local opertors $O_{\alpha,l-1}^Z \Delta_{\alpha,l}^Z O_{\alpha,l+1}^Z$ and $O_{i,l-1}^X \Delta_{i,l}^X O_{i,l+1}^X$ in the bulk model center at even layers, *i.e.*,$l \in 2\mathbb{Z}$. This rule of determining the layer index of local terms supported on three-layers is the only modification in the variant construction.

Explicitly, the bulk Hamiltonian coming from the variant construction is the following,

$$\begin{aligned}
H_{\text{bulk}}^{\text{II}} = &-\sum_l \left( \sum_{\alpha \in \mathcal{A}_1} \Delta_{\alpha,2l}^Z O_{\alpha,2l+1}^Z \Delta_{\alpha,2l+2}^Z + \sum_{i \in \mathcal{A}_1} \Delta_{i,2l}^X O_{i,2l+1}^X \Delta_{i,2l+2}^X \right) \\
&- \sum_l \left( \sum_{\alpha \in \mathcal{A}_2} O_{\alpha,2l-1}^Z \Delta_{\alpha,2l}^Z O_{\alpha,2l+1}^Z + \sum_{i \in \mathcal{A}_2} O_{i,2l-1}^X \Delta_{i,2l}^X O_{i,2l+1}^X \right) \\
&- \sum_{r,l} G_{r,2l+1}^Z - \sum_{s,l} G_{s,2l+1}^X - \sum_{\rho,l} \Gamma_{\rho,2l}^X - \sum_{\sigma,l} \Gamma_{\sigma,2l}^Z \,.
\end{aligned} \tag{G.1}$$

In the example of the GI model capturing the $\mathbb{Z}_2 \times \mathbb{Z}_2$ SPT phase, the two sets of local operators are

$$\begin{aligned}
\mathcal{A}_1 &= \{Z_{2i} Z_{2i+1} Z_{2i+2}, X_{2i-1} X_{2i} X_{2i+1}\} \,, \\
\mathcal{A}_2 &= \{X_{2i}, Z_{2i+1}\} \,.
\end{aligned} \tag{G.2}$$

**Theorem 5.** *A sufficient condition for the bulk model to have a robust ground state subspace is that the GI model has the following properties:*

- *$\mathcal{A}_2$ forms a CSLO.*

- *The dual of $\mathcal{A}_1$, denoted as $\mathcal{A}'_1$, forms a CSLO.*

- *Neither the GI model nor the dual model has local symmetries.*

We can see that the $\mathbb{Z}_2 \times \mathbb{Z}_2$ model has these properties. Particularly, in this example, the dual of $\mathcal{A}_1$ is $\mathcal{A}'_1 = \{Z_{2i+1}, X_{2i}\}$ is a CSLO.

Now let us prove the above theorem. To prove that the bulk model has a robust ground state subspace is the same as to prove that the terms in the bulk Hamiltonian form a CSLO.

Suppose there is a local operator that commutes with all terms in the Hamiltonian of the bulk model, we will show it must be either an identity operator or a product of Hamiltonian local terms.

The operator is local in the sense that its support on any layer is finite, independent of total system size along any direction. We begin with considering that the support of the local operator has a single connected component.

First, it cannot be a local operator that is non-trivial only within a single layer. Such an operator, if it existed on an odd layer, would commute with all operators in $\mathcal{A}_1$ and $\mathcal{A}_2$, thus it would be a local symmetry operator of $H_{GI}$. However, there is no local symmetry in the GI model, as required in the properties. Similarly, if the operator is within an even layer, it would be a local symmetry of the dual model, and this violates the required properties. In the end, a local symmetry operator within a single layer for the bulk model does not exist.

Second, it cannot be a local operator within two adjacent layers. Suppose there exists such an operator, and let us call it $A$. Without losing generality, let us suppose its support is on the $z$-th layer (an odd layer) and the $z+1$-th layer (an even layer). $A$ can thus be decomposed as $A_o A_e$, with $A_o$ ($A_e$) on the odd (even) layer. $A$ commutes with all terms in the Hamiltonian of the bulk model. In particular, $A$ commutes with all terms $O_{k,z-2}\Delta_{k,z-1}O_{k,z}$, for any operator $O_k$ in $\mathcal{A}_2$. $A$ at most share supports with these operators on the $z$-th layer. This means $A_o$ commutes with $\mathcal{A}_2$. Through similar steps, one can show that $A_e$ commutes with $\mathcal{A}'_1$, the dual of $\mathcal{A}_1$. Next, $A$ also commutes with $O_{k,z}\Delta_{k,z+1}O_{k,z+2}$, for any operator $O_k$ in $\mathcal{A}_2$. Because $A_o$ on the $z$-th layer commutes with all $O_k \in \mathcal{A}_2$, $A_e$ on the $z+1$-th layer must commute with all $\Delta_k \in \mathcal{A}'_2$. Thus, $A_e$ commutes with $\mathcal{A}'_1 \cup \mathcal{A}'_2$, which is the set of all local operators in the dual of the GI model. Since we have required that the dual model has no local symmetries, $A_e$ is an identity operator. Through a similar step, we can show that $A_o$ must commute with $\mathcal{A}_1$. Combined with the derivation several steps above, this means that $A_o$ commutes with all Hamiltonian local terms in the GI model. And as we require the GI model has no local symmetries, $A_o$ is at most an identity operator. In conclusion, $A = A_o A_e$ if commutes with all Hamiltonian local terms of the bulk model, must be an identity operator.

As the final case, we consider that the local operator $A$ has a support from the $z_{\min}$-th layer to the $z_{\max}$-th layer. Both $z_{\min}$ and $z_{\max}$ are finite, and $z_{\max} - z_{\min} \geq 3$. In this case, we can run the same steps as in the proof of Theorem 1. That is, by multiplying Hamiltonian local terms, we can reduce the support of $A$ to be within at most two adjacent layers. At this end, we can use the results above to show that $A$ after the reduction, must be an identity operator. Thus, the operator $A$ we begin with, is a product of Hamiltonian local terms.

Finally, we consider that the operator has multiple disconnected components. In this case, we take the operator as a single component operator, which are identity operators on some layers. Then through the steps above, we can conclude that the operator must be either an identity operator, or a product of Hamiltonian local terms. This completes our proof.

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
