# Peer review of "Towards Non-Invertible Anomalies from Generalized Ising Models"

_SciPost Physics, doi:SciPost Phys. 15, 150 (2023)_

## Round 1 · Referee Report · Anonymous (Referee 1) · 2023-3-12

Report

This manuscript introduces a general construction of a $(d+1)$-dimensional solvable bulk model based on a $d$-dimensional generalized Ising (GI) model. The boundary of the $(d+1)$-dimensional model naturally realizes the GI model with extra global constraints (from global symmetry charge projection and generalized boundary conditions). Hence, the bulk theory serves as a characterization of the non-invertible anomaly on the boundary GI model with global constraints. Various examples are provided in this manuscript.

I think this work is interesting and should be published. I hope the authors can address the following questions before the publication of the manuscript.

  1. According to the definition of this paper, the non-invertible anomaly of a $d$-dimensional theory is manifested by the "partition function vector" of this theory. Can the authors comment on how the exactly solvable $d+1$-dimensional bulk theory provides a natural explanation of this partition function vector?

  2. This is a question related to the previous one. The construction of the exactly solvable bulk model uses a generalized version of the Kramers-Wannier duality. In what sense is this bulk construction the natural one that captures the non-invertible anomaly in the $d$-dimensional GI model with global constraints?

  3. The authors claim that "When the bulk model has fracton orders, it can have an anomaly-free boundary, such that when discrete global symmetries appear on the boundary, the boundary Hilbert space includes all charged sectors, rather than only the symmetric sector." on page 3. Can the authors clarify if having all charged sectors appear on the boundary is a sufficient and necessary condition for the absence of the non-invertible anomaly discussed in this paper?

  4. Can this construction capture invertible anomalies as well?

  5. The authors provide an interesting example where the bulk model is the X-cube fracton model in 3 dimensions. Out of curiosity, is it possible to find an example where a type-II fracton model appears as the bulk theory?

  • validity: high
  • significance: good
  • originality: high
  • clarity: top
  • formatting: perfect
  • grammar: -

Author:  Shang Liu  on 2023-08-09  [id 3887]

(in reply to Report 1 on 2023-03-12)
Category:
answer to question

We would like to thank the referee for carefully reading our long paper and for the positive comments. We apologize for the rather delayed reply; we were waiting until both referee reports were ready.

These are all excellent questions, and our responses are given below:

  1. To fully characterize the noninvertible anomaly, we would need to study the boundary theories with all possible topological excitations in the bulk. The corresponding partition functions form a vector, and we need to analyze the modular transformation property of this partition function vector. In this work, for simplicity, we have restricted ourselves only to the vacuum sector, i.e. no topological excitations in the bulk. This is why we use the title "Towards Non-Invertible Anomalies..."; future works are needed to study the boundary theories with more general excitations in the bulk.

  2. Thank you for the interesting question. In the case of pure topological order (no fracton order), the bulk theory is conjectured to be uniquely determined by the boundary theory. Hence, when our construction produces a pure topologically ordered bulk theory, we can probably call it the natural one. In particular, in our construction, we use the symmetries of the quantum model to determine an operator algebra (which is generated by the Hamiltonian local terms and local symmetries operators), it is conjectured that each operator algebra corresponds to a unique bulk topological order. Our construction is an explicit way to determine the bulk topological order. The generalized Kramers-Wannier duality turns out to be an effective step to generate commuting Hamiltonian local terms in the bulk from the operator algebra.

    On the other hand, in the case of fracton orders, multiple bulk theories may be generated from a single boundary theory, and it is not clear whether any of them is more natural than the others.

  3. Having all charged sectors appearing on the boundary is a necessary condition for no anomaly, but it is not sufficient. In our paper, we have shown that noninvertible anomaly can manifest in two ways: constraints on symmetry charge sectors, and constraints on boundary conditions. Absence of anomaly requires the absence of both types of constraints. Here in Page 3, we did not explicitly mention the boundary condition constraints because the notion is a bit hard to clearly explain without explicit examples.

  4. In this paper, we restrict ourselves to generalized Ising models with onsite symmetries. If we would like to also characterize 't Hooft anomalies of 0-form symmetries, then we need to consider symmetry actions which are not onsite, e.g. finite-depth circuit symmetries. This is an interesting future direction.

    Our formalism is also not able to characterize invertible gravitational anomalies, such as the integer quantum Hall edge. This is because stabilizer models are unable to describe chiral theories to begin with. A possible future project is to first find a field-theoretic description of our construction protocol, which might be able to include chiral models.

    Another (perhaps simpler) possibility is to consider noninvertible anomalies of higher form symmetries. For example, as shown in Yoshida's paper 1508.03468, there is a (2+1)D one-form $Z_2\times Z_2$ SPT, which has a stabilizer bulk Hamitonian, as well as an onsite boundary symmetry action. This particular model can be realized with a period-three layer structure (instead of period-two as the models in our paper), so it does not directly fit into our formalism, but some generalized version of our construction may be able to include this example.

  5. It actually seems possible to generate Haah's code as the bulk theory using the variant construction introduced in Appendix G. Let us briefly explain the construction using the attached figure. We take the GI model lattice (original lattice) to be a 2d square lattice with two qubits on each site; see panel (a) of the figure. The dual model lattice is the same except for a relative translation in the $x$ direction by half lattice constant. Now, using the notation in Appendix G, we need to specify two sets of operators $\mathcal A_1$ and $\mathcal A_2$ which define the GI model. These two sets of operators have been illustrated in panel (b) of the figure. The dual model is defined by two other sets of operators $\mathcal A_1'$ and $\mathcal A_2'$ which are dual to $\mathcal A_1$ and $\mathcal A_2$, respectively. We take $\mathcal A_1'$ ($\mathcal A_2'$) to be of the same form as $\mathcal A_2$ ($\mathcal A_1$). The precise duality map between operators is fixed by the following information: Each dual pair of operators are of the same type ($X$-type or $Z$-type), and centered at the same location. Now if we construct the bulk model, we find two different stabilizers (up to translations) as shown in panel (c), where the red arrows indicate the layering direction $z$. These stabilizers are nothing but those in Haah's code.

Attachment:

ReplyFigure.pdf

---

## Round 1 · Referee Report · Anonymous (Referee 2) · 2023-5-30

Strengths

1- The bulk-boundary correspondence is an important issue in the study of non-invertible topological phases. 2- The paper provides a clear and explicit construction for the bulk-boundary correspondence, which is of significance.

Report

The manuscript proposes a bulk-boundary correspondence for non-invertible topological phases, focusing on the study of generalized Ising (GI) models with Z2 symmetries. The authors provide explicit constructions of (d+1)-dimensional bulk models that can match the global constraints of the GI models, including the generation of fracton models.

Overall, the paper makes a valuable contribution to the field by presenting a new perspective on the bulk-boundary correspondence in non-invertible topological phases. The paper is suitable for publication in Scipost Physics. Addressing the below suggestions would improve the clarity and readability of the manuscript.

Requested changes

1- To enhance the clarity of the paper, I recommend that the authors provide more explicit formulas regarding \Omega^Z of the generalized Ising models in section V.A. This would assist readers in understanding the anomaly matching and Claim 1.

2- Additionally, it would be beneficial for the authors to clarify the meaning of "a GI model can terminate on the boundary of a bulk model" directly.

3- Furthermore, there are numerous symbols in the manuscript that can make it challenging to follow certain equations. For instance, in Eq. (6), it would be helpful if the authors refer to the appropriate places where the symbols G's and Gamma's are defined. By doing so, readers can easily track the meaning and interpretation of these symbols.

  • validity: high
  • significance: high
  • originality: high
  • clarity: high
  • formatting: excellent
  • grammar: -

Author:  Shang Liu  on 2023-08-09  [id 3886]

(in reply to Report 2 on 2023-05-30)
Category:
answer to question

We thank the referee for carefully reading our long paper and for the positive comments. Our responses to the requested changes are listed below.

  1. As far as we understand, the referee is referring to the beginning part of Section V (instead of Section V.A) where Claim 1 is stated. We have now added a more precise discussion about the definition of $\Omega^Z$ after it is first mentioned in this section.

  2. We have added a clarification about the meaning of ``GI model terminates on the boundary'' at the first paragraph of Section V.

  3. Indeed, there are many different symbols in this paper. To assist the readers, we have added a table at the beginning of Section IV which summarizes our notations. A reference to this table has also been added right above the expression of the bulk Hamiltonian (previous Eq.6, current Eq.7). In the previous version of our manuscript, we did not explicitly write down the general Hamiltonian of the dual model, and this has now been fixed as well.

We hope with all these changes, our manuscript is more clarified, and easy to read.

---

## Round 2 · Referee Report · Anonymous (Referee 2) · 2023-8-24

Report

I'm pleased to observe that the authors have effectively addressed my earlier concerns. The added explanations have enhanced the paper's clarity. Furthermore, the inclusion of the general Hamiltonian of the dual model addresses a previous omission, enhancing the manuscript's comprehensiveness. Considering these changes, I recommend the manuscript for publication.

---

## Round 2 · Author Response

Dear Editor,

Thank you for sending us the referee reports, and sorry for the delayed reply.

We have addressed all questions from both referees, and made necessary changes to the manuscript accordingly. We believe our manuscript is now ready for publication.

Best regards,
The Authors

---

## Round 2 · List of Changes

1. We added a more precise discussion about the definition of $\Omega^Z$ after it is first mentioned in Section V.

  2. We have added a clarification about the meaning of "GI model terminates on the boundary" at the first paragraph of Section V.

  3. There are many different symbols in this paper. To assist the readers, we have added a table at the beginning of Section IV which summarizes our notations. A reference to this table has also been added right above the expression of the bulk Hamiltonian (previous Eq.6, current Eq.7).

  4. In the previous version of our manuscript, we did not explicitly write down the general Hamiltonian of the dual model, and this has now been fixed.

---

## Editorial Decision

published